# Understanding Overparameterization in Generative Adversarial Networks

**Yogesh Balaji**[1,*]**Mohammadmahdi Sajedi**[2*]**, Neha Mukund Kalibhat**[1]**, Mucong Ding**[1]**,
Dominik Stöger**[2]**, Mahdi Soltanolkotabi**[2]**, Soheil Feizi**[1]
[1] University of Maryland, College Park, MD
[2] University of Southern California, Los Angeles, CA

## ABSTRACT

A broad class of *unsupervised* deep learning methods such as Generative Adversarial Networks (GANs) involve training of overparameterized models where the number of parameters of the model exceeds a certain threshold. Indeed, most successful GANs used in practice are trained using overparameterized generator and discriminator networks, both in terms of depth and width. A large body of work in *supervised* learning have shown the importance of model overparameterization in the convergence of the gradient descent (GD) to globally optimal solutions. In contrast, the unsupervised setting and GANs in particular involve non-convex concave mini-max optimization problems that are often trained using Gradient Descent/Ascent (GDA). The role and benefits of model overparameterization in the convergence of GDA to a global saddle point in non-convex concave problems is far less understood. In this work, we present a comprehensive analysis of the importance of model overparameterization in GANs both theoretically and empirically. We theoretically show that in an overparameterized GAN model with a 1-layer neural network generator and a linear discriminator, GDA converges to a global saddle point of the underlying non-convex concave min-max problem. To the best of our knowledge, this is the first result for global convergence of GDA in such settings. Our theory is based on a more general result that holds for a broader class of nonlinear generators and discriminators that obey certain assumptions (including deeper generators and random feature discriminators). Our theory utilizes and builds upon a novel connection with the convergence analysis of linear time-varying dynamical systems which may have broader implications for understanding the convergence behavior of GDA for non-convex concave problems involving overparameterized models. We also empirically study the role of model overparameterization in GANs using several large-scale experiments on CIFAR-10 and Celeb-A datasets. Our experiments show that overparameterization improves the quality of generated samples across various model architectures and datasets. Remarkably, we observe that overparameterization leads to faster and more stable convergence behavior of GDA across the board.

## 1 INTRODUCTION

In recent years, we have witnessed tremendous progress in deep generative modeling with some state-of-the-art models capable of generating photo-realistic images of objects and scenes (Brock et al., 2019; Karras et al., 2019; Clark et al., 2019). Three prominent classes of deep generative models include GANs (Goodfellow et al., 2014), VAEs (Kingma & Welling, 2014) and normalizing flows (Dinh et al., 2017). Of these, GANs remain a popular choice for data synthesis especially in the image domain. GANs are based on a two player *min-max* game between a generator network that generates samples from a distribution, and a critic (discriminator) network that discriminates real distribution from the generated one. The networks are optimized using Gradient Descent/Ascent (GDA) to reach a saddle-point of the min-max optimization problem.

---

[*]First two authors contributed equally. Correspondence to *yogesh@cs.umd.edu, sajedi@usc.edu*

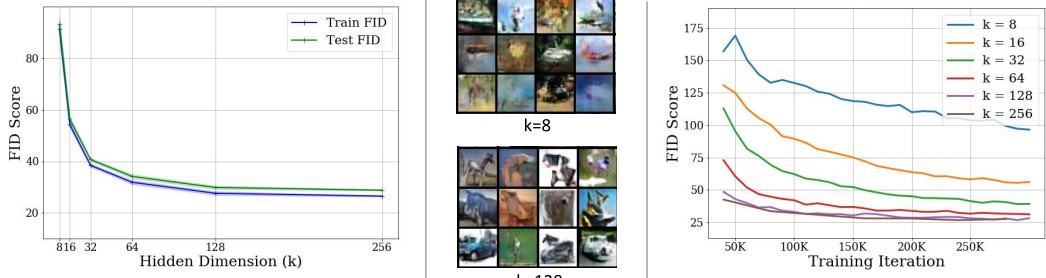

Figure 1: **Overparameterization in GANs.** We train DCGAN models by varying the size of the hidden dimension $k$ (larger the $k$, more overparameterized the models are, see Fig. 8 for details). Overparameterized GANs enjoy improved training and test FID scores *(the left panel)*, generate high-quality samples *(the middle panel)* and have fast and stable convergence *(the right panel)*.

One of the key factors that has contributed to the successful training of GANs is model *overparameterization*, defined based on the model parameters count. By increasing the complexity of discriminator and generator networks, both in depth and width, recent papers show that GANs can achieve photo-realistic image and video synthesis (Brock et al., 2019; Clark et al., 2019; Karras et al., 2019). While these works empirically demonstrate some benefits of *overparameterization*, there is lack of a rigorous study explaining this phenomena. In this work, we attempt to provide a comprehensive understanding of the role of *overparameterization* in GANs, both theoretically and empirically. We note that while *overparameterization* is a key factor in training successful GANs, other factors such as generator and discriminator architectures, regularization functions and model hyperparameters have to be taken into account as well to improve the performance of GANs.

Recently, there has been a large body of work in *supervised* learning (e.g. regression or classification problems) studying the importance of model overparameterization in gradient descent (GD)'s convergence to globally optimal solutions (Soltanolkotabi et al., 2018; Allen-Zhu et al., 2019; Du et al., 2019; Oymak & Soltanolkotabi, 2019; Zou & Gu, 2019; Oymak et al., 2019). A key observation in these works is that, under some conditions, overparameterized models experience *lazy training* (Chizat et al., 2019) where optimal model parameters computed by GD remain close to a randomly initialized model. Thus, using a linear approximation of the model in the parameter space, one can show the global convergence of GD in such minimization problems.

In contrast, training GANs often involves solving a non-convex concave *min-max* optimization problem that fundamentally differs from a single minimization problem of classification/regression. The key question is whether overparameterized GANs also experience lazy training in the sense that overparameterized generator and discriminator networks remain sufficiently close to their initializations. This may then lead to a general theory of global convergence of GDA for such overparameterized non-convex concave min-max problems.

In this paper we first theoretically study the role of overparameterization for a GAN model with a 1-hidden layer generator and a linear discriminator. We study two optimization procedures to solve this problem: (i) using a conventional training procedure in GANs based on GDA in which generator and discriminator networks perform simultaneous steps of gradient descent to optimize their respective models, (ii) using GD to optimize generator's parameters for the optimal discriminator. The latter case corresponds to taking a sufficiently large number of gradient ascent steps to update discriminator's parameters for each GD step of the generator. In both cases, our results show that in an overparameterized regime, the GAN optimization converges to a global solution. To the best of our knowledge, this is the first result showing the global convergence of GDA in such settings. While in our results we focus on one-hidden layer generators and linear discriminators, our theory is based on analyzing a general class of min-max optimization problems which can be used to study a much broader class of generators and discriminators potentially including deep generators and deep random feature-based discriminators. A key component of our analysis is a novel connection to exponential stability of non-symmetric time varying dynamical systems in control theory which may have broader implications for theoretical analysis of GAN's training. Ideas from control theory have

also been used for understanding and improving training dynamics of GANs in (Xu et al., 2019; An et al., 2018).

Having analyzed overparameterized GANs for relatively simple models, we next provide a comprehensive empirical study of this problem for practical GANs such as DCGAN (Radford et al., 2016) and ResNet GAN (Gulrajani et al., 2017) trained on CIFAR-10 and Celeb-A datasets. For example, the benefit of overparamterization in training DCGANs on CIFAR-10 is illustrated in Figure 1. We have three key observations: (i) as the model becomes more overparameterized (e.g. using wider networks), the *training* FID scores that measure the training error, decrease. This phenomenon has been observed in other studies as well (Brock et al., 2019). (ii) overparameterization does not hurt the *test* FID scores (i.e. the generalization gap remains small). This improved test-time performance can also be seen qualitatively in the center panel of Figure 1, where overparameterized models produce samples of improved quality. (iii) Remarkably, overparameterized GANs, with a lot of parameters to optimize over, have significantly improved convergence behavior of GDA, both in terms of rate and stability, compared to small GAN models (see the right panel of Figure 1).

In summary, in this paper

- We provide the first theoretical guarantee of simultaneous GDA's global convergence for an overparameterized GAN with one-hidden neural network generator and a linear discriminator (Theorem 2.1).

- By establishing connections with linear time-varying dynamical systems, we provide a theoretical framework to analyze simultaneous GDA's global convergence for a general overparameterized GAN (including deeper generators and random feature discriminators), under some general conditions (Theorems 2.3 and A.4).

- We provide a comprehensive empirical study of the role of model overparameterization in GANs using several large-scale experiments on CIFAR-10 and Celeb-A datasets. We observe overparameterization improves GANs' training error, generalization error, sample qualities as well as the convergence rate and stability of GDA.

## 2 THEORETICAL RESULTS

### 2.1 PROBLEM FORMULATION

Given $n$ data points of the form $\boldsymbol{x}_1, \boldsymbol{x}_2, \ldots, \boldsymbol{x}_n \in \mathbb{R}^m$, the goal of GAN's training is to find a generator that can mimic sampling from the same distribution as the training data. More specifically, the goal is to find a generator mapping $\mathcal{G}_{\boldsymbol{\theta}}(z) : \mathbb{R}^d \to \mathbb{R}^m$, parameterized by $\boldsymbol{\theta} \in \mathbb{R}^p$, so that $\mathcal{G}_{\boldsymbol{\theta}}(\boldsymbol{z}_1), \mathcal{G}_{\boldsymbol{\theta}}(\boldsymbol{z}_2), \ldots, \mathcal{G}_{\boldsymbol{\theta}}(\boldsymbol{z}_n)$ with $\boldsymbol{z}_1, \boldsymbol{z}_2, \ldots, \boldsymbol{z}_n$ generated i.i.d. according to $\mathcal{N}(\boldsymbol{0}, \boldsymbol{I}_d)$ has a similar empirical distribution to $\boldsymbol{x}_1, \boldsymbol{x}_2, \ldots, \boldsymbol{x}_n$[1]. To measure the discrepancy between the data points and the GAN outputs, one typically uses a discriminator mapping $\mathcal{D}_{\widetilde{\boldsymbol{\theta}}} : \mathbb{R}^m \to \mathbb{R}$ parameterized with $\widetilde{\theta} \in \mathbb{R}^{\widetilde{p}}$. The overall training approach takes the form of the following min-max optimization problem which minimizes the worst-case discrepancy detected by the discriminator

$$\min_{\boldsymbol{\theta}} \max_{\widetilde{\boldsymbol{\theta}}} \quad \frac{1}{n} \sum_{i=1}^{n} \mathcal{D}_{\widetilde{\boldsymbol{\theta}}}(\boldsymbol{x}_i) - \frac{1}{n} \sum_{i=1}^{n} \mathcal{D}_{\widetilde{\boldsymbol{\theta}}}(\mathcal{G}_{\boldsymbol{\theta}}(\boldsymbol{z}_i)) + \mathcal{R}(\widetilde{\boldsymbol{\theta}}). \tag{1}$$

Here, $\mathcal{R}(\widetilde{\boldsymbol{\theta}})$ is a regularizer that typically ensures the discriminator is Lipschitz. This formulation mimics the popular Wasserstein GAN (Arjovsky et al., 2017) (or, IPM GAN) formulations. This optimization problem is typically solved by running Gradient Descent Ascent (GDA) on the minimization/maximization variables.

The generator and discriminator mappings $\mathcal{G}$ and $\mathcal{D}$ used in practice are often deep neural networks. Thus, the min-max optimization problem above is highly nonlinear and non-convex concave. Saddle point optimization is a classical and fundamental problem in game theory (Von Neumann & Morgenstern, 1953) and control (Gutman, 1979). However, most of the classical results apply to the

---

[1]In general, the number of observed and generated samples can be different. However, in practical GAN implementations, batch sizes of observed and generated samples are usually the same. Thus, for simplicity, we make this assumption in our setup.

convex-concave case (Arrow et al., 1958) while the saddle point optimization of GANs is often *non convex-concave*. If GDA converges to the global (local) saddle points, we say it is globally (locally) stable. For a general min-max optimization, however, GDA can be trapped in a loop or even diverge. Except in some special cases (e.g. (Feizi et al., 2018) for a quadratic GAN formulation or (Lei et al., 2019) for the under-parametrized setup when the generator is a one-layer network), GDA is not globally stable for GANs in general (Nagarajan & Kolter, 2017; Mescheder et al., 2018; Adolphs et al., 2019; Mescheder et al., 2017; Daskalakis et al., 2020).

None of these works, however, study the role of model overparameterization in the global/local convergence (stability) of GDA. In particular, it has been empirically observed (as we also demonstrate in this paper) that when the generator/discriminator contain a large number of parameters (i.e. are sufficiently overparameterized) GDA does indeed find (near) globally optimal solutions. In this section we wish to demystify this phenomenon from a theoretical perspective.

## 2.2 Definition of Model Overparameterization

In this paper, we use *overparameterization* in the context of model parameters count. Informally speaking, overparameterized models have large number of parameters, that is we assume that the number of model parameters is sufficiently large. In specific problem setups of Section 2, we precisely compute thresholds where the number of model parameters should exceed in order to observe nice convergence properties of GDA. Note that the definition of *overparameterization* based on model parameters count is related, but distinct from the complexity of the hypothesis class. For instance, in our empirical studies, when we say we *overparameterize* a neural network, we fix the number of layers in the neural network and increase the hidden dimensions. Our definition does not include the case where the number of layers also increases, which forms a different hypothesis class.

## 2.3 Results for one-hidden layer generators and random discriminators

In this section, we discuss our main results on the convergence of gradient based algorithms when training GANs in the overparameterized regime. We focus on the case where the generator takes the form of a single hidden-layer ReLU network with $d$ inputs, $k$ hidden units, and $m$ outputs. Specifically, $\mathcal{G}(z) = V \cdot \text{ReLU}(Wz)$ with $W \in \mathbb{R}^{k \times d}$ and $V \in \mathbb{R}^{m \times k}$ denoting the input-to-hidden and hidden-to-output weights. We also consider a linear discriminator of the form $\mathcal{D}(x) = d^T x$ with an $\ell_2$ regularizer on the weights i.e. $\mathcal{R}(d) = -\|d\|_{\ell_2}^2/2$. The overall min-max optimization problem (equation 1) takes the form

$$\min_{W \in \mathbb{R}^{k \times d}} \max_{d \in \mathbb{R}^m} \quad \mathcal{L}(W, d) := \langle d, \frac{1}{n} \sum_{i=1}^n (x_i - V\text{ReLU}(Wz_i)) \rangle - \frac{\|d\|_{\ell_2}^2}{2}. \tag{2}$$

Note that we initialize $V$ at random and keep it fixed throughout the training. The common approach to solve the above optimization problem is to run a Gradient Descent Ascent (GDA) algorithm. At iteration $t$, GDA takes the form

$$\begin{cases} d_{t+1} = & d_t + \mu \nabla_d \mathcal{L}(W_t, d_t) \\ W_{t+1} = & W_t - \eta \nabla_W \mathcal{L}(W_t, d_t) \end{cases} \tag{3}$$

Next, we establish the global convergence of GDA for an overparameterized model. Note that a global saddle point $(W^*, d^*)$ is defined as

$$\mathcal{L}(W^*, d) \leq \mathcal{L}(W^*, d^*) \leq \mathcal{L}(W, d^*)$$

for all feasible $W$ and $d$. If these inequalities hold in a local neighborhood, $(W^*, d^*)$ is called a local saddle point.

**Theorem 2.1** *Let $x_1, x_2, \ldots, x_n \in \mathbb{R}^m$ be $n$ training data with their mean defined as $\bar{x} :=$ $\frac{1}{n} \sum_{i=1}^n x_i$. Consider the GAN model with a linear discriminator of the form $\mathcal{D}(x) = d^T x$ parameterized by $d \in \mathbb{R}^m$ and a one hidden layer neural network generator of the form $\mathcal{G}(z) = V\phi(Wz)$ parameterized by $W \in \mathbb{R}^{k \times d}$ with $V \in \mathbb{R}^{m \times k}$ a fixed matrix generated at random with i.i.d. $\mathcal{N}(0, \sigma_v^2)$ entries. Also assume the input data to the generator $\{z_i\}_{i=1}^n$ are generated i.i.d. according to $\sim \mathcal{N}(0, \sigma_z^2 I_d)$. Furthermore, assume the generator weights at initialization $W_0 \in \mathbb{R}^{k \times d}$*

*are generated i.i.d. according to $\mathcal{N}(0, \sigma_w^2)$. Furthermore, assume the standard deviations above obey $\sigma_v \sigma_w \sigma_z \geq \|\bar{x}\|_{\ell_2} / (md^{\frac{5}{2}} \log d^{\frac{3}{2}})$. Then, as long as*

$$k \geq C \cdot md^4 \log(d)^3$$

*with $C$ a fixed constant, running GDA updates per equation 3 starting from the random $W_0$ above and $d_0 = \mathbf{0}^2$ with step-sizes obeying $0 < \mu \leq 1$ and $\eta = \bar{\eta} \frac{\mu}{324 \cdot k \cdot \frac{d + \frac{n-1}{\pi}}{n} \cdot \sigma_v^2 \cdot \sigma_z^2}$, with $\bar{\eta} \leq 1$, satisfies*

$$\left\| \frac{1}{n} \sum_{i=1}^{n} V ReLU(W_\tau z_i) - \bar{x} \right\|_{\ell_2} \leq 5 \left( 1 - 10^{-5} \cdot \bar{\eta}\mu \right)^\tau \left\| \frac{1}{n} \sum_{i=1}^{n} V ReLU(W_0 z_i) - \bar{x} \right\|_{\ell_2}. \quad (4)$$

*This holds with probability at least $1 - (n+5) e^{-\frac{m}{1500}} - 5k \cdot e^{-c_1 \cdot n} - (2k+2) e^{-\frac{d}{216}} - ne^{-c_2 \cdot md^3 \log(d)^2}$ where $c_1$, $c_2$ are fixed numerical constants.*

To better understand the implications of the above theorem, note that the objective of equation 2 can be simplified by solving the inner maximization in a closed form so that the min-max problem in equation 2 is equivalent to the following single minimization problem:

$$\min_{W} \ \mathcal{L}(W) := \frac{1}{2} \left\| \frac{1}{n} \sum_{i=1}^{n} V ReLU(W z_i) - \bar{x} \right\|_{\ell_2}^2, \quad (5)$$

which has a global optimum of zero. As a result equation 4 in Theorem 2.1 guarantees that running simultaneous GDA updates achieves the global optimum. This holds as long as the generator network is sufficiently overparameterized in the sense that the number of hidden nodes is polynomially large in its output dimension $m$ and input dimension $d$. Interestingly, the rate of convergence guaranteed by this result is geometric, guaranteeing fast GDA convergence to the global optima. To the extent of our knowledge, this is the first result that establishes the global convergence of simultaneous GDA for an overparameterized GAN model.

While the result proved above shows the global convergence of GDA for a GAN with 1-hidden layer generator and a linear discriminator, for a general GAN model, local saddle points may not even exist and GDA may converge to approximate local saddle points (Berard et al., 2020; Farnia & Ozdaglar, 2020). For a general min-max problem, (Daskalakis et al., 2020) has recently shown that *approximate* local saddle points exist under some general conditions on the lipschitzness of the objective function. Understanding GDA dynamics for a general GAN remains an important open problem. Our result in Theorem 2.1 is a first and important step towards that.

We acknowledge that the considered GAN formulation of equation 2 is very simpler than GANs used in practice. Specially, since the discriminator is linear, this GAN can be viewed as a moment-matching GAN (Li et al., 2017) pushing first moments of input and generative distributions towards each other. Alternatively, this GAN formulation can be viewed as one instance of the Sliced Wasserstein GAN (Deshpande et al., 2018). Although the maximization on discriminator's parameters is concave, the minimization over the generator's parameters is still non-convex due to the use of a neural-net generator. Thus, the overall optimization problem is a non-trivial non-convex concave min-max problem. From that perspective, our result in Theorem 2.1 *partially* explains the role of model overparameterization in GDA's convergence for GANs.

Given the closed form equation 5, one may wonder what would happen if we run gradient descent on this minimization objective directly. That is running gradient descent updates of the form $W_{\tau+1} = W_\tau - \eta \nabla \mathcal{L}(W_\tau)$ with $\mathcal{L}(W)$ given by equation 5. This is equivalent to GDA but instead of running one gradient ascent iteration for the maximization iteration we run infinitely many. Interestingly, in some successful GAN implementations (Gulrajani et al., 2017), often more updates on the discriminator's parameters are run per generator's updates. This is the subject of the next result.

**Theorem 2.2** *Consider the setup of Theorem 2.1. Then as long as*

$$k \geq C \cdot md^4 \log(d)^3$$

---

[2]The zero initialization of $d$ is merely done for simplicity. A similar result can be derived for an arbitrary initialization of the discriminator's parameters with minor modifications. See Theorem 2.3 for such a result.

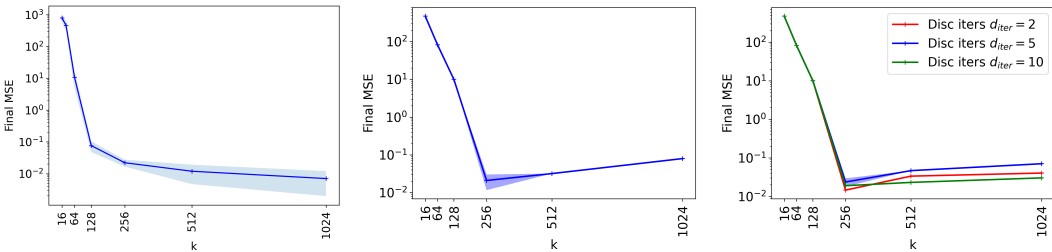

(a) Discriminator trained to optimality

(b) Gradient Descent Ascent

(c) $d_{iter}$ steps of discriminator update per generator iteration

Figure 2: **Convergence plot** a GAN model with linear discriminator and 1-hidden layer generator as the hidden dimension ($k$) increases. *Final mse* is the mse loss between true data mean and the mean of generated distribution. Over-parameterized models show improved convergence

with $C$ a fixed numerical constant, running GD updates of the form $\boldsymbol{W}_{\tau+1} = \boldsymbol{W}_\tau - \eta \nabla \mathcal{L}(\boldsymbol{W}_\tau)$ on the loss given in equation 5 with step-size $\eta = \frac{2\bar{\eta}}{243k \cdot \frac{d + \frac{n-1}{\pi}}{n} \cdot \sigma_v^2 \cdot \sigma_z^2}$, with $\bar{\eta} \leq 1$, satisfies

$$\left\| \frac{1}{n} \sum_{i=1}^n \boldsymbol{V} ReLU\left(\boldsymbol{W}_\tau \boldsymbol{z}_i\right) - \bar{\boldsymbol{x}} \right\|_{\ell_2} \leq \left(1 - 4 \times 10^{-6} \cdot \bar{\eta}\right)^\tau \left\| \frac{1}{n} \sum_{i=1}^n \boldsymbol{V} ReLU\left(\boldsymbol{W}_0 \boldsymbol{z}_i\right) - \bar{\boldsymbol{x}} \right\|_{\ell_2}. \quad (6)$$

This holds with probability at least $1 - (n+5) e^{-\frac{m}{1500}} - 5k \cdot e^{-c_1 \cdot n} - (2k+2) e^{-\frac{d}{216}} - ne^{-c_2 \cdot md^3 \log(d)^2}$ with $c_1$, $c_2$ fixed numerical constants.

This theorem states that if we solve the max part of equation 2 in closed form and run GD on the loss function per equation 5 with enough overparameterization, the loss will decrease at a geometric rate to zero. This result holds again when the model is sufficiently overparameterized. The proof of Theorem 2.2 relies on a result from (Oymak & Soltanolkotabi, 2020), which was developed in the framework of supervised learning. Also note that the amount of overparameterization required in both Theorems 2.1 and 2.2 is the same.

## 2.4 CAN THE ANALYSIS BE EXTENDED TO MORE GENERAL GANS?

In the previous section, we focused on the implications of our results for one-hidden layer generator and linear discriminator. However, as it will become clear in the proofs, our theoretical results are based on analyzing the convergence behavior of GDA on a more general min-max problem of the form

$$\min_{\boldsymbol{\theta} \in \mathbb{R}^p} \max_{\boldsymbol{d} \in \mathbb{R}^m} h(\boldsymbol{\theta}, \boldsymbol{d}) := \langle \boldsymbol{d}, f(\boldsymbol{\theta}) - \boldsymbol{y} \rangle - \frac{\|\boldsymbol{d}\|_{\ell_2}^2}{2}, \quad (7)$$

where $f : \mathbb{R}^p \to \mathbb{R}^m$ denotes a general nonlinear mapping.

**Theorem 2.3 (Informal version of Theorem A.4)** *Consider a general nonlinear mapping $f : \mathbb{R}^p \to \mathbb{R}^m$ with the singular values of its Jacobian mapping around initialization obeying certain assumptions (most notably $\sigma_{\min}(\mathcal{J}(\boldsymbol{\theta}_0)) \geq \alpha$). Then, running GDA iterations of the form*

$$\begin{cases} \boldsymbol{d}_{t+1} = \boldsymbol{d}_t + \mu \nabla_{\boldsymbol{d}} h(\boldsymbol{\theta}_t, \boldsymbol{d}_t) \\ \boldsymbol{\theta}_{t+1} = \boldsymbol{\theta}_t - \eta \nabla_{\boldsymbol{\theta}} h(\boldsymbol{\theta}_t, \boldsymbol{d}_t) \end{cases} \quad (8)$$

*with sufficiently small step sizes $\eta$ and $\mu$ obeys*

$$\|f(\boldsymbol{\theta}_t) - \boldsymbol{y}\|_{\ell_2} \leq \gamma \left(1 - \frac{\eta \alpha^2}{2}\right)^t \sqrt{\|f(\boldsymbol{\theta}_0) - \boldsymbol{y}\|_{\ell_2}^2 + \|\boldsymbol{d}_0\|_{\ell_2}^2}.$$

Note that similar to the previous sections one can solve the maximization problem in equation 7 in closed form so that equation 7 is equivalent to the following minimization problem

$$\min_{\boldsymbol{\theta} \in \mathbb{R}^p} \mathcal{L}(\boldsymbol{\theta}) := \frac{1}{2} \|f(\boldsymbol{\theta}) - \boldsymbol{y}\|_{\ell_2}^2, \quad (9)$$

with global optima equal to zero. Theorem 2.3 ensures that GDA converges with a fast geometric rate to this global optima. This holds as soon as the model $f(\boldsymbol{\theta})$ is sufficiently overparameterized which is quantitatively captured via the minimum singular value assumption on the Jacobian at initialization $(\sigma_{\min}(\mathcal{J}(\boldsymbol{\theta}_0)) \geq \alpha$ which can only hold when $m \leq p$). This general result can thus be used to provide theoretical guarantees for a much more general class of generators and discriminators. To be more specific, consider a deep GAN model where the generator $\mathcal{G}_{\boldsymbol{\theta}}$ is a deep neural network with parameters $\boldsymbol{\theta}$ and the discriminator is a deep random feature model of the form $\mathcal{D}_{\boldsymbol{d}}(\boldsymbol{x}) = \boldsymbol{d}^T \psi(\boldsymbol{x})$ parameterized with $d$ and $\psi : \mathbb{R}^d \to \mathbb{R}^m$ a deep neural network with random weights. Then the min-max training optimization problem equation 1 with a regularizer $\mathcal{R}(\boldsymbol{d}) = -\|\boldsymbol{d}\|_{\ell_2}^2 /2$ is a special instance of equation 7 with

$$f(\boldsymbol{\theta}) := \frac{1}{n} \sum_{i=1}^{n} \psi(\mathcal{G}_{\boldsymbol{\theta}}(\boldsymbol{z}_i)) \quad \text{and} \quad \boldsymbol{y} := \frac{1}{n} \sum_{i=1}^{n} \psi(\boldsymbol{x}_i)$$

Therefore, the above result can in principle be used to rigorously analyze global convergence of GDA for an overparameterized GAN problem with a deep generator and a deep random feature discriminator model. However, characterizing the precise amount of overparameterization required for such a result to hold requires a precise analysis of the minimum singular value of the Jacobian of $f(\boldsymbol{\theta})$ at initialization as well as other singular value related conditions stated in Theorem A.4. We defer such a precise analysis to future works.

**Numerical Validations:** Next, we numerically study the convergence of GAN model considered in Theorems 2.1 and 2.2 where the discriminator is a linear network while the generator is a one hidden layer neural net. In our experiments, we generate $\boldsymbol{x}_i$'s from an $m$-dimension Gaussian distribution with mean $\mu$ and an identity covariance matrix. The mean vector $\mu$ is randomly generated. We train two variants of GAN models using (1) GDA (as considered in Thm 2.1) and (2) GD on generator while solving the discriminator to optimality (as considered in Thm 2.2).

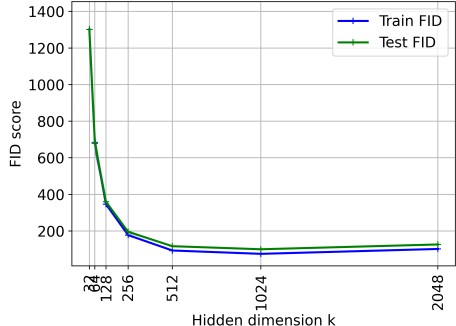

Figure 3: **MLP Overparameterization on MNIST.**

In Fig. 2, we plot the converged loss values of GAN models trained using both techniques (1) and (2) as the hidden dimension $k$ of the generator is varied. The MSE loss between the true data mean and the data mean of generated samples is used as our evaluation metric. As this MSE loss approaches 0, the model converges to the global saddle point. We observe that overparameterized GAN models show improved convergence behavior than the narrower models. Additionally, the MSE loss converges to 0 for larger values of $k$ which shows that with sufficient overparamterization, GDA converges to a global saddle point.

## 3 EXPERIMENTS

In this section, we demonstrate benefits of overparamterization in large GAN models. In particular, we train GANs on two benchmark datasets: CIFAR-10 ($32 \times 32$ resolution) and Celeb-A ($64 \times 64$ resolution). We use two commonly used GAN architectures: DCGAN and Resnet-based GAN. For both of these architectures, we train several models, each with a different number of filters in each layer, denoted by $k$. For simplicity, we refer to $k$ as the hidden dimension. Appendix Fig. 8 illustrates the architectures used in our experiments. Networks with large $k$ are more overparameterized.

We use the same value of $k$ for both generator and discriminator networks. This is in line with the design choice made in most recent GAN models (Radford et al., 2016; Brock et al., 2019), where the size of generator and discriminator models are roughly maintained the same. We train each model till convergence and evaluate the performance of converged models using FID scores. FID scores measure the Frechet distance between feature distributions of real and generated data

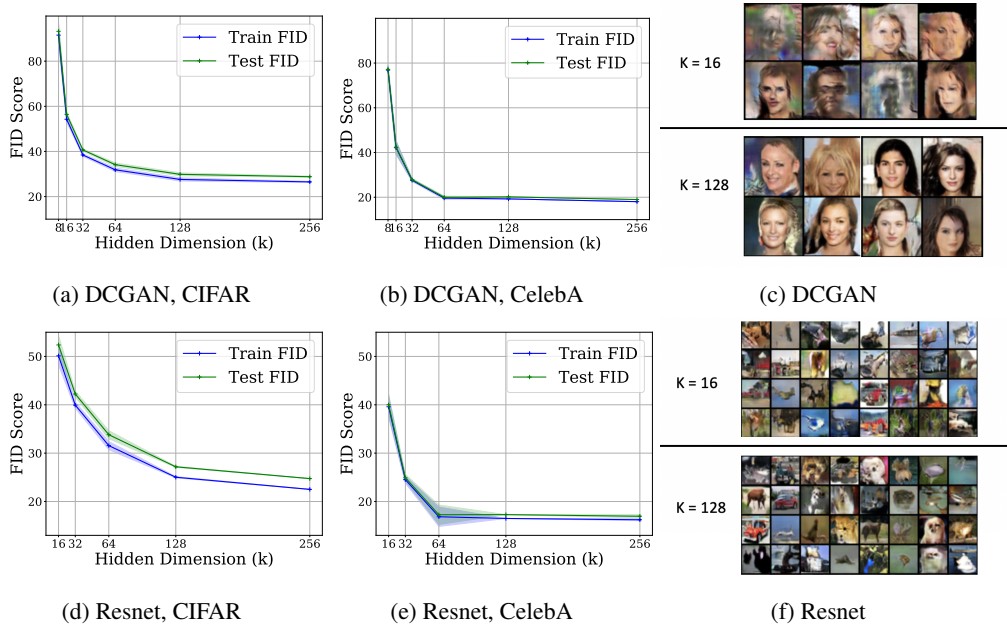

Figure 4: **Overparamterization Results:** We plot the FID scores (lower, better) of DCGAN and Resnet DCGAN as the hidden dimension $k$ is varied. Results on CIFAR-10 and Celeb-A are shown on the plots on the left and right panels, respectively. Overparameterization gives better FID scores.

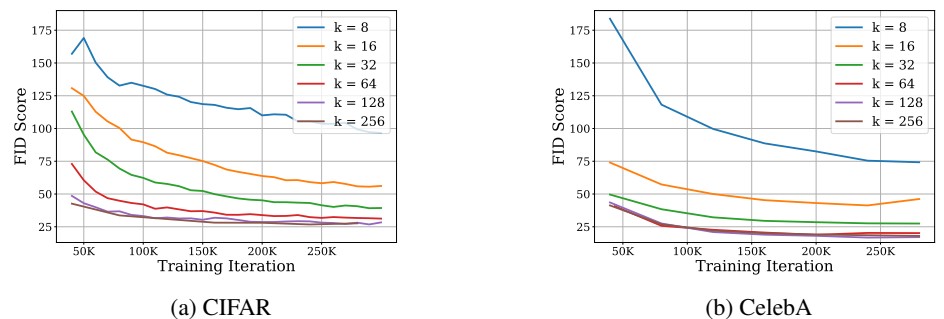

Figure 5: **DCGAN Training Results:** We plot the FID scores across training iterations of DCGAN on CIFAR-10 and Celeb-A for different values of hidden dimension $k$. Remarkably, we observe that over-parameterization improves the rate of convergence of GDA and its stability in training.

distributions (Heusel et al., 2017). A small FID score indicates high-quality synthesized samples. Each experiment is conducted for 5 runs, and mean and the variance of FID scores are reported.

**Overparameterization yields better generative models:** In Fig. 4, we show the plot of FID scores as the hidden dimension ($k$) is varied for DCGAN and Resnet GAN models. We observe a clear trend where the FID scores are high (i.e. poor) for small values of $k$, while they improve as models become more overparameterized. Also, the FID scores saturate beyond $k = 64$ for DCGAN models, and $k = 128$ for Resnet GAN models. Interestingly, these are the standard values used in the existing model architecures (Radford et al., 2016; Gulrajani et al., 2017).This trend is also consistent on MLP GANs trained on MNIST dataset (Fig. 3). We however notice that FID score in MLP GANs increase marginally as $k$ increases from 1024 to 2048. This is potentially due to an increased generalization gap in this regime where it offsets potential benefits of over-parameterization

**Overparameterization leads to improved convergence of GDA:** In Fig. 5, we show the plot of FID scores over training iterations for different values for $k$. We observe that models with larger

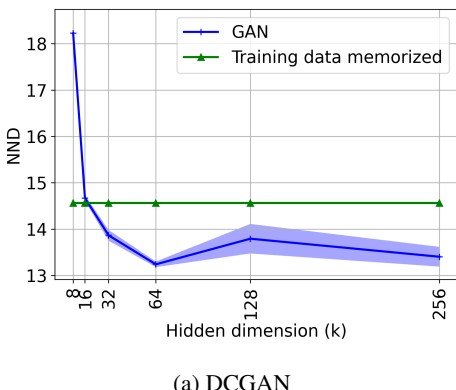 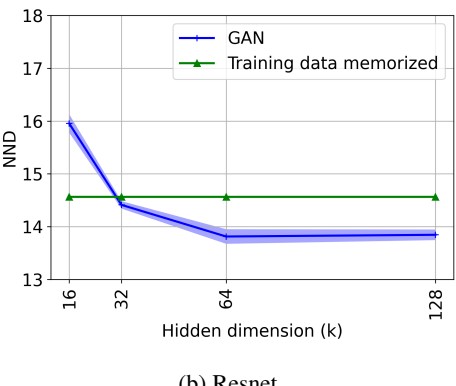

(a) DCGAN                                        (b) Resnet

Figure 6: **Generalization in GANs:** We plot the **NND scores** as the hidden dimension $k$ is varied for DCGAN (shown in (a)) and Resnet (shown in (b)) models.

values of $k$ converge faster and demonstrate a more stable behavior. This agrees with our theoretical results that overparameterized models have a fast rate of convergence.

**Generalization gap in GANs:** To study the generalization gap, we compute the FID scores by using (1) the training-set of real data, which we call *FID train*, and (2) a held-out validation set of real data, which we call *FID test*. In Fig. 4, a plot of FID train *(in blue)* and FID test *(in green)* are shown as the hidden dimension $k$ is varied. We observe that FID test values are consistently higher than the the FID train values. Their gap does not increase with increasing overparameterization.

However, as explained in (Gulrajani et al., 2019), the FID score has the issue of assigning low values to memorized samples. To alleviate the issue, (Gulrajani et al., 2019; Arora et al., 2017) proposed Neural Net Divergence (NND) to measure generalization in GANs. In Fig. 6, we plot NND scores by varying the hidden dimensions in DCGAN and Resnet GAN trained on CIFAR-10 dataset. We observe that increasing the value of $k$ decreases the NND score. Interestingly, the NND score of memorized samples are higher than most of the GAN models. This indicates that overparameterized models have not been memorizing training samples and produce better generative models.

## 4 CONCLUSION

In this paper, we perform a systematic study of the importance of overparameterization in training GANs. We first analyze a GAN model with one-hidden layer generator and a linear discriminator optimized using Gradient Descent Ascent (GDA). Under this setup, we prove that with sufficient overparameterization, GDA converges to a global saddle point. Additionally, our result demonstrate that overparameterized models have a fast rate of convergence. We then validate our theoretical findings through extensive experiments on DCGAN and Resnet models trained on CIFAR-10 and Celeb-A datasets. We observe overparameterized models to perform well both in terms of the rate of convergence and the quality of generated samples.

## 5 ACKNOWLEDGEMENT

M. Sajedi would like to thank Sarah Dean for introducing (Rugh, 1996). This project was supported in part by NSF CAREER AWARD 1942230, HR00112090132, HR001119S0026, NIST 60NANB20D134 and Simons Fellowship on "Foundations of Deep Learning." M. Soltanolkotabi is supported by the Packard Fellowship in Science and Engineering, a Sloan Research Fellowship in Mathematics, an NSF-CAREER under award 1846369, the Air Force Office of Scientific Research Young Investigator Program (AFOSR-YIP) under award FA9550-18-1-0078, DARPA Learning with Less Labels (LwLL) and FastNICS programs, and NSF-CIF awards 1813877 and 2008443.

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

# Appendix

## A    PROOFS

In this section, we prove Theorems 2.1 and 2.2. First, we provide some notations we use throughout the remainder of the paper in Section A.1. Before proving these specialized results for one hidden layer generators and linear discriminators (Theorems 2.1 and 2.2), we state and prove a more general result (formal version of Theorem 2.3) on the convergence of GDA on a general class of min-max problems in Section A.3. Then we state a few preliminary calculations in Section A.4. Next, we state some key lemmas in Section A.5 and defer their proofs to Appendix B. Finally, we prove Theorems 2.1 and 2.2 in Sections A.6 and A.7, respectively.

### A.1    NOTATION

We will use $C$, $c$, $c_1$, etc. to denote positive absolute constants, whose value may change throughout the paper and from line to line. We use $\phi(z) = \text{ReLU}(z) = \max(0, z)$ and its (generalized) derivative $\phi'(z) = \mathbf{1}_{\{z \geq 0\}}$ with $\mathbf{1}$ being the indicator function. $\sigma_{min}(\boldsymbol{X})$ and $\sigma_{max}(\boldsymbol{X}) = \|\boldsymbol{X}\|$ denote the minimum and maximum singular values of matrix $\boldsymbol{X}$. For two arbitrary matrices $\boldsymbol{A}$ and $\boldsymbol{B}$, $\boldsymbol{A} \otimes \boldsymbol{B}$ denotes their kronecker product. The spectral radius of a matrix $\boldsymbol{A} \in \mathbb{C}^{n \times n}$ is defined as $\rho(\boldsymbol{A}) = \max\{|\lambda_1|, \ldots, |\lambda_n|\}$, where $\lambda_i$'s are the eigenvalues of $\boldsymbol{A}$. Throughout the proof we shall assume $\phi := ReLU$ to avoid unnecessarily long expressions.

### A.2    PROOF SKETCH OF THE MAIN RESULTS

In this section, we provide a brief overview of our proofs. We focus on the main result in this manuscript, which is about the convergence of GDA (Theorem 2.1). To do this we study the converge of GDA on the more general min-max problem of the form (see Theorem A.4 for a formal statement)

$$\min_{\boldsymbol{\theta} \in \mathbb{R}^n} \max_{\boldsymbol{d} \in \mathbb{R}^m} h(\boldsymbol{\theta}, \boldsymbol{d}) := \langle \boldsymbol{d}, f(\boldsymbol{\theta}) - \boldsymbol{y} \rangle - \frac{\|\boldsymbol{d}\|_{\ell_2}^2}{2}. \tag{10}$$

In this case the GDA iterates take the form

$$\begin{cases} \boldsymbol{d}_{t+1} = (1 - \mu)\boldsymbol{d}_t + \mu(f(\boldsymbol{\theta}_t) - \boldsymbol{y}) \\ \boldsymbol{\theta}_{t+1} = \boldsymbol{\theta}_t - \eta \mathcal{J}^T(\boldsymbol{\theta}_t)\boldsymbol{d}_t \end{cases}. \tag{11}$$

Our proof for global convergence of GDA on this min-max loss consists of the following steps.

**Step 1: Recasting the GDA updates as a linear time-varying system**
In the first step we carry out a series of algebraic manipulations to recast the GDA updates (equation 11) into the following form

$$\begin{bmatrix} \boldsymbol{r}_{t+1} \\ \boldsymbol{d}_{t+1} \end{bmatrix} = \boldsymbol{A}_t \begin{bmatrix} \boldsymbol{r}_t \\ \boldsymbol{d}_t \end{bmatrix},$$

where $\boldsymbol{r}_t = f(\boldsymbol{\theta_t}) - \boldsymbol{y}$ denotes the residuum and $\boldsymbol{A}_t$ denotes a properly defined transition matrix.

**Step 2: Approximation by a linear time-invariant system**
Next, to analyze the behavior of the time-varying dynamical system above we approximate it by the following time-invariant linear dynamical system

$$\begin{bmatrix} \widetilde{\boldsymbol{r}}_{t+1} \\ \widetilde{\boldsymbol{d}}_{t+1} \end{bmatrix} = \begin{bmatrix} \boldsymbol{I} & -\eta \mathcal{J}^T(\boldsymbol{\theta}_0)\mathcal{J}(\boldsymbol{\theta}_0) \\ \mu \boldsymbol{I} & (1 - \mu)\boldsymbol{I} \end{bmatrix} \begin{bmatrix} \widetilde{\boldsymbol{r}}_t \\ \widetilde{\boldsymbol{d}}_t \end{bmatrix},$$

where $\boldsymbol{\theta}_0$ denotes the initialization. The validity of this approximation is ensured by our assumptions on the Jacobian of the function $f$, which, among others, guarantee that it does not change too much in a sufficiently large neighborhood around the initialization and that the smallest singular value of $\mathcal{J}(\boldsymbol{\theta}_0)$ is bounded from below.

**Step 3: Analysis of time-invariant linear dynamical system**
To analyze the time-invariant dynamical system above, we utilize and refine intricate arguments

from the control theory literature involving the spectral radius of the fixed transition matrix above to obtain

$$\left\| \begin{bmatrix} \widetilde{r}_t \\ \widetilde{d}_t \end{bmatrix} \right\|_{\ell_2} \lesssim \left( 1 - \eta \alpha^2 \right)^t \left\| \begin{bmatrix} \widetilde{r}_0 \\ \widetilde{d}_0 \end{bmatrix} \right\|_{\ell_2}.$$

**Step 4: Completing the proof via a perturbation argument**

In the last step of our proof we show that the two sequences $\begin{bmatrix} r_t \\ d_t \end{bmatrix}$ and $\begin{bmatrix} \widetilde{r}_t \\ \widetilde{d}_t \end{bmatrix}$ will remain close to each other. This is based on a novel perturbation argument. The latter combined with Step 3 allows us to conclude

$$\left\| \begin{bmatrix} r_t \\ d_t \end{bmatrix} \right\|_{\ell_2} \lesssim \left( 1 - \frac{\eta \alpha^2}{2} \right)^t \left\| \begin{bmatrix} r_0 \\ d_0 \end{bmatrix} \right\|_{\ell_2},$$

which finishes the global convergence of GDA on equation 10 and hence the proof of Theorem A.4.

In order to deduce Theorem 2.1 from Theorem A.4, we need to check that the Jacobian at the initialization is bounded from below at the origin and that it does not change too quickly in a large enough neighborhood. In order to prove that we will leverage recent ideas from the deep learning theory literature revolving around the neural tangent kernel. This allows us to guarantee that this conditions are indeed met, if the neural network is sufficiently wide and the initialization is chosen large enough.

The second main result of this manuscript, Theorem 2.2, can be deduced more directly from recent results on overparameterized learning (see Oymak & Soltanolkatabi (2020)). Hence, we have deferred its proof to Section A.7.

### A.3 ANALYSIS OF GDA: A CONTROL THEORY PERSPECTIVE

In this section we will focus on solving a general min-max optimization problem of the form

$$\min_{\theta \in \mathbb{R}^n} \max_{d \in \mathbb{R}^m} h(\theta, d) := \langle d, f(\theta) - y \rangle - \frac{\|d\|_{\ell_2}^2}{2}, \tag{12}$$

where $f : \mathbb{R}^n \to \mathbb{R}^m$ is a general nonlinear mapping. In particular, we focus on analyzing the convergence behavior of Gradient Descent/Ascent (GDA) on the above loss, starting from initial estimates $\theta_0$ and $d_0$. In this case the GDA updates take the following form

$$\begin{cases} d_{t+1} = (1 - \mu) d_t + \mu (f(\theta_t) - y) \\ \theta_{t+1} = \theta_t - \eta \mathcal{J}^T(\theta_t) d_t \end{cases}. \tag{13}$$

We note that solving the inner maximization problem in equation 12 would yield

$$\min_{\theta \in \mathbb{R}^n} \frac{1}{2} \|f(\theta) - y\|_{\ell_2}^2. \tag{14}$$

In this section, our goal is to show that when running the GDA updates of equation 13, the norm of the residual vector defined as $r_t := f(\theta_t) - y$ goes to zero and hence we reach a global optimum of equation 14 (and in turn equation 12).

Our proof will build on ideas from control theory and dynamical systems literature. For that, we are first going to rewrite the equations 13 in a more convenient way. We define the average Jacobian along the path connecting two points $x, y \in \mathbb{R}^n$ as

$$\mathcal{J}(y, x) = \int_0^1 \mathcal{J}(x + \alpha(y - x)) \, d\alpha,$$

where $\mathcal{J}(\theta) \in \mathbb{R}^{m \times n}$ is the Jacobian associated with the nonlinear mapping $f$. Next, from the fundamental theorem of calculus it follows that

$$\begin{aligned} r_{t+1} = f(\theta_{t+1}) - y &= f\left(\theta_t - \eta \mathcal{J}_t^T d_t\right) - y \\ &= f(\theta_t) - \eta \mathcal{J}_{t+1,t} \mathcal{J}_t^T d_t - y \\ &= r_t - \eta \mathcal{J}_{t+1,t} \mathcal{J}_t^T d_t, \end{aligned} \tag{15}$$

where we used the shorthands $\mathcal{J}_t := \mathcal{J}(\boldsymbol{\theta}_t)$ and $\mathcal{J}_{t+1,t} := \mathcal{J}(\boldsymbol{\theta}_{t+1}, \boldsymbol{\theta}_t)$ for exposition purposes. Next, we combine the updates $\boldsymbol{r}_t$ and $\boldsymbol{d}_t$ into a state vector of the form $\boldsymbol{z}_t := \begin{bmatrix} \boldsymbol{r}_t \\ \boldsymbol{d}_t \end{bmatrix} \in \mathbb{R}^{2m}$. Using this notation the relationship between the state vectors from one iteration to the next takes the form

$$\boldsymbol{z}_{t+1} = \underbrace{\begin{bmatrix} \boldsymbol{I} & -\eta\mathcal{J}_{t+1,t}\mathcal{J}_t^T \\ \mu\boldsymbol{I} & (1-\mu)\,\boldsymbol{I} \end{bmatrix}}_{=:\boldsymbol{A}_t} \boldsymbol{z}_t, \;\; t \geq 0, \tag{16}$$

which resembles a time-varying linear dynamical system with transition matrix $\boldsymbol{A}_t$. Now note that to show convergence of $\boldsymbol{r}_t$ to zero it suffices to show convergence of $\boldsymbol{z}_t$ to zero. To do this we utilize the following notion of uniform exponential stability, which will be crucial in analyzing the solutions of equation 16. (See Rugh (1996) for a comprehensive overview on stability notions in discrete state equations.)

**Definition 1** *A linear state equation of the form $\boldsymbol{z}_{t+1} = \boldsymbol{A}_t\boldsymbol{z}_t$ is called uniformly exponentially stable if for every $t \geq 0$ we have $\|\boldsymbol{z}_t\|_{\ell_2} \leq \gamma\lambda^t \|\boldsymbol{z}_0\|_{\ell_2}$, where $\gamma \geq 1$ is a finite constant and $0 \leq \lambda < 1$.*

Using the above definition to show the convergence of the state vector $\boldsymbol{z}_t$ to zero at a geometric rate it suffices to show the state equation 16 is exponentially stable.[3] For that, we are first going to analyze a state equation which results from linearizing the nonlinear function $f(\boldsymbol{\theta})$ around the initialization $\boldsymbol{\theta}_0$. In the next step, we are going to show that the behavior of these two problems are similar, provided we stay close to initialization (which we are also going to prove). Specifically, we consider the linearized problem

$$\min_{\widetilde{\boldsymbol{\theta}} \in \mathbb{R}^n} \max_{\widetilde{\boldsymbol{d}} \in \mathbb{R}^m} h_{\text{lin}}\left(\widetilde{\boldsymbol{\theta}}, \widetilde{\boldsymbol{d}}\right) := \left\langle \widetilde{\boldsymbol{d}}, f(\boldsymbol{\theta}_0) + \mathcal{J}_0\left(\widetilde{\boldsymbol{\theta}} - \boldsymbol{\theta}_0\right) - \boldsymbol{y} \right\rangle - \frac{\left\|\widetilde{\boldsymbol{d}}\right\|_{\ell_2}^2}{2}. \tag{17}$$

We first analyze GDA on this linearized problem starting from the same initialization as the original problem, i.e. $\widetilde{\boldsymbol{\theta}}_0 = \boldsymbol{\theta}_0$ and $\widetilde{\boldsymbol{d}}_0 = \boldsymbol{d}_0$. The gradient descent update for $\widetilde{\boldsymbol{\theta}}_t$ takes the form

$$\widetilde{\boldsymbol{\theta}}_{t+1} = \widetilde{\boldsymbol{\theta}}_t - \eta\mathcal{J}_0^T\widetilde{\boldsymbol{d}}_t, \tag{18}$$

and the gradient ascent update for $\widetilde{\boldsymbol{d}}_t$ takes the form

$$\begin{aligned} \widetilde{\boldsymbol{d}}_{t+1} &= \widetilde{\boldsymbol{d}}_t + \mu\left(f(\boldsymbol{\theta}_0) + \mathcal{J}_0\left(\widetilde{\boldsymbol{\theta}}_t - \boldsymbol{\theta}_0\right) - \boldsymbol{y} - \widetilde{\boldsymbol{d}}_t\right) \\ &= (1-\mu)\,\widetilde{\boldsymbol{d}}_t + \mu\widetilde{\boldsymbol{r}}_t, \end{aligned} \tag{19}$$

where we used the linear residual defined as $\widetilde{\boldsymbol{r}}_t = f(\boldsymbol{\theta}_0) + \mathcal{J}_0\left(\widetilde{\boldsymbol{\theta}}_t - \boldsymbol{\theta}_0\right) - \boldsymbol{y}$. Moreover, the residual from one iterate to the next can be written as follows

$$\begin{aligned} \widetilde{\boldsymbol{r}}_{t+1} &= f(\boldsymbol{\theta}_0) + \mathcal{J}_0\left(\widetilde{\boldsymbol{\theta}}_{t+1} - \boldsymbol{\theta}_0\right) - \boldsymbol{y} \\ &= f(\boldsymbol{\theta}_0) + \mathcal{J}_0\left(\widetilde{\boldsymbol{\theta}}_t - \eta\mathcal{J}_0^T\widetilde{\boldsymbol{d}}_t - \boldsymbol{\theta}_0\right) - \boldsymbol{y} \\ &= \widetilde{\boldsymbol{r}}_t - \eta\mathcal{J}_0\mathcal{J}_0^T\widetilde{\boldsymbol{d}}_t. \end{aligned} \tag{20}$$

Again, we define a new vector $\widetilde{\boldsymbol{z}}_t = \begin{bmatrix} \widetilde{\boldsymbol{r}}_t \\ \widetilde{\boldsymbol{d}}_t \end{bmatrix} \in \mathbb{R}^{2m}$ and by putting together equations 19 and 20 we arrive at

$$\widetilde{\boldsymbol{z}}_{t+1} = \begin{bmatrix} \boldsymbol{I} & -\eta\mathcal{J}_0\mathcal{J}_0^T \\ \mu\boldsymbol{I} & (1-\mu)\,\boldsymbol{I} \end{bmatrix} \widetilde{\boldsymbol{z}}_t = \boldsymbol{A}\widetilde{\boldsymbol{z}}_t, \;\; t \geq 0, \tag{21}$$

---

[3]We note that technically the dynamical system equation 16 is not linear. However, we still use exponential stability with some abuse of notation to refer to the property that $\|\boldsymbol{z}_t\|_{\ell_2} \leq \gamma\lambda^t \|\boldsymbol{z}_0\|_{\ell_2}$ holds. As we will see in the forth-coming paragraphs, our formal analysis is via a novel perturbation analysis of a linear dynamical system and therefore keeping this terminology is justified.

which is of the form of a linear time-invariant state equation. As a first step in our proof, we are going to show that the linearized state equations are uniformly exponentially stable. First, recall the following well-known lemma, which characterizes uniform exponential stability in terms of the eigenvalues of $\boldsymbol{A}$.

**Lemma A.1** *(Rugh, 1996, Theorem 22.11) A linear state equation of the form $\widetilde{z}_{t+1} = \boldsymbol{A}\widetilde{z}_t$ with $\boldsymbol{A}$ a fixed matrix is uniformly exponentially stable if and only if all eigenvalues of $\boldsymbol{A}$ have magnitudes strictly less than one, i.e. $\rho\left(\boldsymbol{A}\right) < 1$. In this case, it holds for all $t \geq 0$ and all $\boldsymbol{z}$ that*

$$\|\boldsymbol{A}^t \boldsymbol{z}\| \leq \gamma \rho\left(\boldsymbol{A}\right)^t \|\boldsymbol{z}\|,$$

*where $\gamma \geq 1$ is an absolute constant, which only depends on $\boldsymbol{A}$.*

In the next lemma, we prove that under suitable assumptions on $\mathcal{J}_0$ and the step sizes $\mu$ and $\eta$ the state equations 21 indeed fulfill this condition.

**Lemma A.2** *Assume that $\alpha \leq \sigma_{min}\left(\mathcal{J}_0\right) \leq \sigma_{max}\left(\mathcal{J}_0\right) \leq \beta$ and consider the matrix $\boldsymbol{A} = \begin{bmatrix} \boldsymbol{I} & -\eta\mathcal{J}_0\mathcal{J}_0^T \\ \mu\boldsymbol{I} & (1-\mu)\boldsymbol{I} \end{bmatrix}$. Suppose that $\frac{\mu}{\eta} \geq 4\beta^2$. Then it holds that $\rho\left(\boldsymbol{A}\right) \leq 1 - \eta\alpha^2$.*

**Proof** Suppose that $\lambda$ is an eigenvalue of $\boldsymbol{A}$. Hence, there is an eigenvector $[\boldsymbol{x}, \boldsymbol{y}]^T \neq \boldsymbol{0}$ such that

$$\begin{bmatrix} \boldsymbol{I} & -\eta\mathcal{J}_0\mathcal{J}_0^T \\ \mu\boldsymbol{I} & (1-\mu)\boldsymbol{I} \end{bmatrix} \begin{bmatrix} \boldsymbol{x} \\ \boldsymbol{y} \end{bmatrix} = \lambda \begin{bmatrix} \boldsymbol{x} \\ \boldsymbol{y} \end{bmatrix}$$

holds. By a direct calculation we observe that this yields the equation

$$\eta\mathcal{J}_0\mathcal{J}_0^T \boldsymbol{x} = \left(\frac{-\left(1-\lambda\right)^2}{\mu} + (1-\lambda)\right)\boldsymbol{x}.$$

In particular, $\boldsymbol{x}$ must be an eigenvector of $\mathcal{J}_0\mathcal{J}_0^T$. Denoting the corresponding eigenvalue with $s$, we obtain the identity

$$\frac{\left(1-\lambda\right)^2}{\mu} - (1-\lambda) + \eta s = 0.$$

Hence, we must have

$$\lambda \in \left\{ 1 - \frac{\mu}{2} + \sqrt{\frac{\mu^2}{4} - \mu\eta s};\ 1 - \frac{\mu}{2} - \sqrt{\frac{\mu^2}{4} - \mu\eta s} \right\}.$$

Note that the square root is indeed well-defined, since

$$\frac{\mu^2}{4} - \mu\eta s \geq \mu\eta\beta^2 - \mu\eta s \geq 0,$$

where in the first inequality we used the assumption $\frac{\mu}{\eta} \geq 4\beta^2$ and in the second line we used that $s \leq \beta^2$, which is a consequence of our assumption on the singular values of $\mathcal{J}_0$. Hence, it follows by the reverse triangle inequality that

$$|\lambda| - \left(1 - \frac{\mu}{2}\right) \leq \left|\lambda - \left(1 - \frac{\mu}{2}\right)\right| = \sqrt{\left(\frac{\mu}{2}\right)^2 - \mu\eta s} < \frac{\mu}{2} - \eta s \leq \frac{\mu}{2} - \eta\alpha^2,$$

where the second inequality is valid as $\frac{\mu}{2} - \eta s \geq 0$ is implied by $\frac{\mu}{2} \geq 2\eta\beta^2 > \eta s$. In the last inequality we used the fact that $\alpha^2 \leq s$, which is a consequence of our assumption on the singular values of $\mathcal{J}_0$. By rearranging terms, we obtain that $|\lambda| < 1 - \eta\alpha^2$. Since $\lambda$ was an arbitrary eigenvalue of $\boldsymbol{A}$, the result follows. ∎

Since the last lemma shows that under suitable conditions it holds that $\rho\left(\boldsymbol{A}\right) < 1$, Lemma A.3 yields uniform exponential stability of our state equations. However, this will not be sufficient for our purposes. The reason is that Lemma A.3 does not specify the constant $\gamma$ and that in order to deal with the time-varying dynamical system we will need a precise estimate. The next lemma shows that for the state equations 21 we have, under suitable assumptions, $\gamma \leq 5$.

**Lemma A.3** *Consider the linear, time invariant system of equations*

$$\widetilde{z}_{t+1} = \begin{bmatrix} I & -\eta \mathcal{J}_0 \mathcal{J}_0^T \\ \mu I & (1-\mu) I \end{bmatrix} \widetilde{z}_t = A \widetilde{z}_t, \quad t \geq 0.$$

*Furthermore, assume that $\alpha \leq \sigma_{min}(\mathcal{J}_0) \leq \sigma_{max}(\mathcal{J}_0) \leq \beta$ and suppose that the condition $\frac{\mu}{\eta} \geq 8\beta^2$ is satisfied. Then there is a constant $\gamma \leq 5$ such that for all $t \geq 0$ it holds that*

$$\|\widetilde{z}_t\|_{\ell_2} \leq \gamma \left(1 - \eta \alpha^2\right)^t \|\widetilde{z}_0\|_{\ell_2}.$$

**Proof** Denote the SVD decomposition of $\mathcal{J}_0$ by $W \Sigma V^T$ and note that

$$\begin{bmatrix} I & -\eta \mathcal{J}_0 \mathcal{J}_0^T \\ \mu I & (1-\mu) I \end{bmatrix} = \begin{bmatrix} W & 0 \\ 0 & W \end{bmatrix} \begin{bmatrix} I & -\eta \Sigma \Sigma^T \\ \mu I & (1-\mu) I \end{bmatrix} \begin{bmatrix} W^T & 0 \\ 0 & W^T. \end{bmatrix}$$

This means we can write

$$\begin{bmatrix} I & -\eta \mathcal{J}_0 \mathcal{J}_0^T \\ \mu I & (1-\mu) I \end{bmatrix} = \begin{bmatrix} W & 0 \\ 0 & W \end{bmatrix} P \begin{bmatrix} C_1 & & 0 \\ & \ddots & \\ 0 & & C_m \end{bmatrix} P^T \begin{bmatrix} W^T & 0 \\ 0 & W^T \end{bmatrix},$$

where $P$ is a permutation matrix and the matrices $C_i$ are of the form $C_i = \begin{bmatrix} 1 & -\eta \sigma_i^2 \\ \mu & (1-\mu) \end{bmatrix}$, for $1 \leq i \leq m$, where the $\sigma_i$'s denote the singular values of $\mathcal{J}_0$. Using this decomposition we can deduce

$$\|\widetilde{z}_t\|_{\ell_2} = \|A^t \widetilde{z}_0\|_{\ell_2} \leq \|A^t\| \|\widetilde{z}_0\|_{\ell_2} = \left( \max_{1 \leq i \leq m} \|C_i^t\| \right) \|\widetilde{z}_0\|_{\ell_2}.$$

Now suppose that $V_i D_i V_i^{-1}$ is the eigenvalue decomposition of $C_i$, where the columns of $V_i$ contain the eigenvectors and $D_i$ is a diagonal matrix consisting of the eigenvalues. (Note that it follows from our assumptions on $\mu$ and $\eta$ that the matrix $C_i$ is diagonalizable.) We have

$$\|C_i^t\| = \|V_i D_i^t V_i^{-1}\| \leq \|V_i\| \|D_i^t\| \|V_i^{-1}\| = \kappa_i \cdot \rho(C_i)^t,$$

where we defined $\kappa_i := \|V_i\| \|V_i^{-1}\|$. From Lemma A.2 we know that the assumption $\frac{\mu}{\eta} \geq 4\beta^2$ results in $\rho(A) \leq 1 - \eta \alpha^2$. Therefore, defining $\gamma := \max_{1 \leq i \leq m} \kappa_i$ and noting $\rho(A) = \max_{1 \leq i \leq m} \rho(C_i)$, we obtain that

$$\|\widetilde{z}_t\|_{\ell_2} \leq \left( \max_{1 \leq i \leq m} \|C_i^t\| \right) \|\widetilde{z}_0\|_{\ell_2} \leq \gamma \left(1 - \eta \alpha^2\right)^t \|\widetilde{z}_0\|_{\ell_2}.$$

In order to finish the proof we need to show that $\gamma \leq 5$. For that, note that calculating the eigenvectors of $C_i$ directly reveals that we can represent this matrix as

$$V_i = \begin{bmatrix} \frac{1+\sqrt{1-4\frac{\eta \sigma_i^2}{\mu}}}{2} & \frac{1-\sqrt{1-4\frac{\eta \sigma_i^2}{\mu}}}{2} \\ 1 & 1 \end{bmatrix}.$$

Since $\|V_i\| = \sqrt{\lambda_{\max}(V_i V_i^T)}$ and $\|V_i^{-1}\| = \sqrt{\lambda_{\min}(V_i V_i^T)}$, we calculate $V_i V_i^T$, which yields

$$V_i V_i^T = \begin{bmatrix} 1 - 2\frac{\eta \sigma_i^2}{\mu} & 1 \\ 1 & 2 \end{bmatrix}.$$

This representation allows us to directly calculate the two eigenvalues of $V_i V_i^T$, which shows that

$$\kappa_i = \sqrt{\frac{\lambda_{\max}\left(V_i V_i^T\right)}{\lambda_{\min}\left(V_i V_i^T\right)}}$$

$$= \sqrt{\frac{3 - 2\frac{\eta\sigma_i^2}{\mu} + \sqrt{\left(1 + 2\frac{\eta\sigma_i^2}{\mu}\right)^2 + 4}}{3 - 2\frac{\eta\sigma_i^2}{\mu} - \sqrt{\left(1 + 2\frac{\eta\sigma_i^2}{\mu}\right)^2 + 4}}}$$

$$= \frac{3 - 2\frac{\eta\sigma_i^2}{\mu} + \sqrt{\left(1 + 2\frac{\eta\sigma_i^2}{\mu}\right)^2 + 4}}{2\sqrt{1 - 4\frac{\eta\sigma_i^2}{\mu}}}$$

$$\leq \frac{6}{2\sqrt{1 - 4\frac{\eta\sigma_i^2}{\mu}}}$$

$$< 5,$$

where the last inequality holds because of $\frac{\eta\sigma_i^2}{\mu} \leq \frac{\eta\beta^2}{\mu} \leq \frac{1}{8}$. Since $\gamma = \max\limits_{1 \leq i \leq m} \kappa_i$, this finishes the proof. ∎

Now that we have shown that the linearized iterates converge to the global optima we turn our attention to showing that the nonlinear iterates 16 are close to its linear counterpart 21. For that, we make the following assumptions.

**Assumption 1:** The singular values of the Jacobian at initialization are bounded from below

$$\sigma_{\min}\left(\mathcal{J}\left(\boldsymbol{\theta}_0\right)\right) \geq \alpha$$

for some positive constants $\alpha$ and $\beta$.

**Assumption 2:** In a neighborhood with radius $R$ around the initialization, the Jacobian mapping associated with $f$ obeys

$$\|\mathcal{J}\left(\boldsymbol{\theta}\right)\| \leq \beta$$

for all $\boldsymbol{\theta} \in \mathcal{B}_R\left(\boldsymbol{\theta}_0\right)$, where $\mathcal{B}_R\left(\boldsymbol{\theta}_0\right) := \{\boldsymbol{\theta} \in \mathbb{R}^p : \|\boldsymbol{\theta} - \boldsymbol{\theta}_0\|_{\ell_2} \leq R\}$.

**Assumption 3:** In a neighborhood with radius $R$ around the initialization, the spectral norm of the Jacobian varies no more than $\epsilon$ in the sense that

$$\|\mathcal{J}\left(\boldsymbol{\theta}\right) - \mathcal{J}\left(\boldsymbol{\theta}_0\right)\| \leq \epsilon$$

for all $\boldsymbol{\theta} \in \mathcal{B}_R\left(\boldsymbol{\theta}_0\right)$.

With these assumptions in place, we are ready to state the main theorem.

**Theorem A.4** *Consider the GDA updates for the min-max optimization problem 12*

$$\begin{bmatrix} \boldsymbol{d}_{t+1} \\ \boldsymbol{\theta}_{t+1} \end{bmatrix} = \begin{bmatrix} \boldsymbol{d}_t + \mu\nabla_{\boldsymbol{d}}h(\boldsymbol{\theta}_t, \boldsymbol{d}_t) \\ \boldsymbol{\theta}_t - \eta\nabla_{\boldsymbol{\theta}}h(\boldsymbol{\theta}_t, \boldsymbol{d}_t) \end{bmatrix} \tag{22}$$

*and consider the GDA updates of the linearized problem 21*

$$\begin{bmatrix} \widetilde{\boldsymbol{d}}_{t+1} \\ \widetilde{\boldsymbol{\theta}}_{t+1} \end{bmatrix} = \begin{bmatrix} \widetilde{\boldsymbol{d}}_t + \mu\nabla_{\boldsymbol{d}}h_{lin}(\widetilde{\boldsymbol{\theta}}_t, \widetilde{\boldsymbol{d}}_t) \\ \widetilde{\boldsymbol{\theta}}_t - \eta\nabla_{\boldsymbol{\theta}}h_{lin}(\widetilde{\boldsymbol{\theta}}_t, \widetilde{\boldsymbol{d}}_t) \end{bmatrix}. \tag{23}$$

*Set $\boldsymbol{z}_t := \begin{bmatrix} \boldsymbol{r}_t \\ \boldsymbol{d}_t \end{bmatrix}$ and $\widetilde{\boldsymbol{z}}_t := \begin{bmatrix} \widetilde{\boldsymbol{r}}_t \\ \widetilde{\boldsymbol{d}}_t \end{bmatrix}$, where $\boldsymbol{r}_t := f\left(\boldsymbol{\theta}_t\right) - \boldsymbol{y}$ and $\widetilde{\boldsymbol{r}}_t = f\left(\boldsymbol{\theta}_0\right) + \mathcal{J}_0\left(\widetilde{\boldsymbol{\theta}}_t - \boldsymbol{\theta}_0\right) - \boldsymbol{y}$ denote the residuals.*
*Assume that the step sizes of the gradient descent ascent updates satisfy $\frac{\mu}{\eta} \geq 8\beta^2$ as well as $0 <$*

$\mu \leq 1$. *Moreover, assume that the assumptions 1-3 hold for the Jacobian $\mathcal{J}(\boldsymbol{\theta})$ of $f(\boldsymbol{\theta})$ around the initialization $\boldsymbol{\theta}_0 \in \mathbb{R}^n$ with parameters $\alpha$, $\beta$, $\epsilon$, and*

$$R := 2\gamma \frac{\beta^2}{\alpha^2} \left\| \begin{bmatrix} \mathcal{J}_0^\dagger & 0 \\ 0 & \mathcal{J}_0^\dagger \end{bmatrix} \boldsymbol{z}_0 \right\|_{\ell_2} + \frac{18\epsilon\beta^2\gamma^2}{\alpha^4} \|\boldsymbol{z}_0\|_{\ell_2}, \tag{24}$$

*which satisfy $4\gamma\beta\epsilon \leq \alpha^2$. Here, $1 \leq \gamma \leq 5$ is a constant, which only depends on $\mu$, $\eta$, and $\mathcal{J}_0$. By $\mathcal{J}_0^\dagger$ we denote the pseudo-inverse of the Jacobian at initialization $\mathcal{J}_0$. Then, assuming the same initialization $\boldsymbol{\theta}_0 = \widetilde{\boldsymbol{\theta}}_0$, $\boldsymbol{d}_0 = \widetilde{\boldsymbol{d}}_0$ (and, hence, $\boldsymbol{z}_0 = \widetilde{\boldsymbol{z}}_0$), the following holds for all iterations $t \geq 0$.*

- $\|\boldsymbol{z}_t\|_{\ell_2}$ *converges to $0$ with a geometric rate, i.e.*

$$\|\boldsymbol{z}_t\|_{\ell_2} \leq \gamma \left(1 - \frac{\eta\alpha^2}{2}\right)^t \|\boldsymbol{z}_0\|_{\ell_2}. \tag{25}$$

- *The trajectories of $\boldsymbol{z}_t$ and $\widetilde{\boldsymbol{z}}_t$ stay close to each other and converge to the same limit, i.e.*

$$\|\boldsymbol{z}_t - \widetilde{\boldsymbol{z}}_t\|_{\ell_2} \leq 2\eta\gamma^2\beta\epsilon \cdot t \left(1 - \frac{\eta\alpha^2}{2}\right)^{t-1} \|\boldsymbol{z}_0\|_{\ell_2}$$
$$\leq \frac{4\gamma^2\beta\epsilon}{e\left(15\ln\frac{16}{15}\right)\alpha^2} \|\boldsymbol{z}_0\|_{\ell_2}. \tag{26}$$

- *The parameters of the original and linearized problems stay close to each other, i.e.*

$$\left\|\widetilde{\boldsymbol{\theta}}_t - \boldsymbol{\theta}_t\right\|_{\ell_2} \leq \frac{9\epsilon\beta^2\gamma^2}{\alpha^4} \|\boldsymbol{z}_0\|_{\ell_2}, \tag{27}$$

- *The parameters of the original problem stay close to the initialization, i.e.*

$$\|\boldsymbol{\theta}_t - \boldsymbol{\theta}_0\|_{\ell_2} \leq \frac{R}{2}. \tag{28}$$

Theorem A.4 will be the main ingredient in the proof of Theorem 2.1. However, as discussed in Section 2.4 we believe that this meta theorem can be used to deal with a much richer class of generators and discriminators.

### A.3.1 PROOF OF THEOREM A.4

We will prove the statements in the theorem by induction. The base case for $\tau = 0$ is trivial. Now assume that the equations equation 25 to equation 28 hold for $\tau = 0, \ldots, t-1$. Our goal is to show that they hold for iteration $t$ as well.

**Part I:** First, we are going to show that $\boldsymbol{\theta}_t \in \mathcal{B}_R(\boldsymbol{\theta}_0)$. Note that by the triangle inequality and the induction assumption we have that

$$\|\boldsymbol{\theta}_t - \boldsymbol{\theta}_0\|_{\ell_2} \leq \|\boldsymbol{\theta}_t - \boldsymbol{\theta}_{t-1}\|_{\ell_2} + \|\boldsymbol{\theta}_{t-1} - \boldsymbol{\theta}_0\|_{\ell_2}$$
$$\leq \|\boldsymbol{\theta}_t - \boldsymbol{\theta}_{t-1}\|_{\ell_2} + \frac{R}{2}.$$

Hence, in order to prove the claim it remains to show that $\|\boldsymbol{\theta}_t - \boldsymbol{\theta}_{t-1}\|_{\ell_2} \leq \frac{R}{2}$. For that, we compute

$$\frac{1}{\eta} \|\boldsymbol{\theta}_t - \boldsymbol{\theta}_{t-1}\|_{\ell_2} = \left\|\mathcal{J}^T(\boldsymbol{\theta}_{t-1}) \boldsymbol{d}_{t-1}\right\|_{\ell_2}$$

$$\leq \left\|\mathcal{J}^T(\boldsymbol{\theta}_{t-1}) \widetilde{\boldsymbol{d}}_{t-1}\right\|_{\ell_2} + \left\|\mathcal{J}^T(\boldsymbol{\theta}_{t-1})\right\| \left\|\boldsymbol{d}_{t-1} - \widetilde{\boldsymbol{d}}_{t-1}\right\|_{\ell_2}$$

$$\leq \left\|\mathcal{J}_0^T \widetilde{\boldsymbol{d}}_{t-1}\right\|_{\ell_2} + \left\|\mathcal{J}(\boldsymbol{\theta}_{t-1}) - \mathcal{J}_0\right\| \left\|\widetilde{\boldsymbol{d}}_{t-1}\right\|_{\ell_2} + \left\|\mathcal{J}^T(\boldsymbol{\theta}_{t-1})\right\| \left\|\boldsymbol{d}_{t-1} - \widetilde{\boldsymbol{d}}_{t-1}\right\|_{\ell_2}$$

$$\overset{(i)}{\leq} \gamma \left\| \begin{bmatrix} \mathcal{J}_0^T & 0 \\ 0 & \mathcal{J}_0^T \end{bmatrix} \boldsymbol{z}_0 \right\|_{\ell_2} + \epsilon \cdot \gamma \|\boldsymbol{z}_0\|_{\ell_2} + \frac{4\beta^2\epsilon\gamma^2}{e\left(15\ln\frac{16}{15}\right)\alpha^2} \|\boldsymbol{z}_0\|_{\ell_2}$$

$$\overset{(ii)}{\leq} \gamma\beta^2 \left\| \begin{bmatrix} \mathcal{J}_0^\dagger & 0 \\ 0 & \mathcal{J}_0^\dagger \end{bmatrix} \boldsymbol{z}_0 \right\|_{\ell_2} + \frac{3\beta^2\epsilon\gamma^2}{\alpha^2} \|\boldsymbol{z}_0\|_{\ell_2},$$

where $\gamma \leq 5$ is a constant. Let us verify the last two inequalities. Inequality $(ii)$ holds because $1 \leq \gamma, 1 \leq \frac{\beta^2}{\alpha^2}$, and

$$
\begin{aligned}
\left\| \begin{bmatrix} \mathcal{J}_0^T & 0 \\ 0 & \mathcal{J}_0^T \end{bmatrix} \boldsymbol{z}_0 \right\|_{\ell_2} &= \left\| \begin{bmatrix} \boldsymbol{V}\boldsymbol{\Sigma}^T\boldsymbol{W}^T & 0 \\ 0 & \boldsymbol{V}\boldsymbol{\Sigma}^T\boldsymbol{W}^T \end{bmatrix} \boldsymbol{z}_0 \right\|_{\ell_2} \\
&= \sqrt{\sum_{i=1}^n \sigma_i^2 \left( \langle \boldsymbol{w}_i, \boldsymbol{r}_0 \rangle^2 + \langle \boldsymbol{w}_i, \boldsymbol{d}_0 \rangle^2 \right)} \\
&\leq \beta^2 \sqrt{\sum_{i=1}^n \frac{1}{\sigma_i^2} \left( \langle \boldsymbol{w}_i, \boldsymbol{r}_0 \rangle^2 + \langle \boldsymbol{w}_i, \boldsymbol{d}_0 \rangle^2 \right)} = \beta^2 \left\| \begin{bmatrix} \mathcal{J}_0^\dagger & 0 \\ 0 & \mathcal{J}_0^\dagger \end{bmatrix} \boldsymbol{z}_0 \right\|_{\ell_2}. \quad (29)
\end{aligned}
$$

Also $(i)$ follows from assumptions 1-3, $\left\| \boldsymbol{d}_{t-1} - \widetilde{\boldsymbol{d}}_{t-1} \right\|_{\ell_2} \leq \| \boldsymbol{z}_{t-1} - \widetilde{\boldsymbol{z}}_{t-1} \|_{\ell_2}$ together with induction assumption equation 26, $\left\| \widetilde{\boldsymbol{d}}_{t-1} \right\|_{\ell_2} \leq \| \widetilde{\boldsymbol{z}}_{t-1} \|_{\ell_2} \leq \| \boldsymbol{z}_0 \|_{\ell_2}$, and

$$
\begin{aligned}
\left\| \mathcal{J}_0^T \widetilde{\boldsymbol{d}}_{t-1} \right\|_{\ell_2} &\leq \left\| \begin{bmatrix} \mathcal{J}_0^T \widetilde{\boldsymbol{r}}_{t-1} \\ \mathcal{J}_0^T \widetilde{\boldsymbol{d}}_{t-1} \end{bmatrix} \right\|_{\ell_2} \\
&= \left\| \begin{bmatrix} \boldsymbol{I} & -\eta \mathcal{J}_0^T \mathcal{J}_0 \\ \mu \boldsymbol{I} & (1-\mu)\boldsymbol{I} \end{bmatrix} \begin{bmatrix} \mathcal{J}_0^T \widetilde{\boldsymbol{r}}_{t-2} \\ \mathcal{J}_0^T \widetilde{\boldsymbol{d}}_{t-2} \end{bmatrix} \right\|_{\ell_2} \\
&= \left\| \begin{bmatrix} \boldsymbol{I} & -\eta \mathcal{J}_0^T \mathcal{J}_0 \\ \mu \boldsymbol{I} & (1-\mu)\boldsymbol{I} \end{bmatrix}^{t-1} \begin{bmatrix} \mathcal{J}_0^T \widetilde{\boldsymbol{r}}_0 \\ \mathcal{J}_0^T \widetilde{\boldsymbol{d}}_0 \end{bmatrix} \right\|_{\ell_2} \leq \gamma \left(1 - \eta\alpha^2\right)^{t-1} \left\| \begin{bmatrix} \mathcal{J}_0^T & 0 \\ 0 & \mathcal{J}_0^T \end{bmatrix} \boldsymbol{z}_0 \right\|_{\ell_2},
\end{aligned}
$$
(30)

where in the last inequality we applied Lemma A.3. Finally, by using $\eta \leq \frac{1}{8\beta^2}$ we arrive at

$$
\begin{aligned}
\| \boldsymbol{\theta}_t - \boldsymbol{\theta}_{t-1} \|_{\ell_2} &\leq \gamma \eta \beta^2 \left\| \begin{bmatrix} \mathcal{J}_0^\dagger & 0 \\ 0 & \mathcal{J}_0^\dagger \end{bmatrix} \boldsymbol{z}_0 \right\|_{\ell_2} + \frac{3\eta\beta^2\epsilon\gamma^2}{\alpha^2} \| \boldsymbol{z}_0 \|_{\ell_2} \\
&\leq \frac{\gamma}{8} \left\| \begin{bmatrix} \mathcal{J}_0^\dagger & 0 \\ 0 & \mathcal{J}_0^\dagger \end{bmatrix} \boldsymbol{z}_0 \right\|_{\ell_2} + \frac{3\epsilon\gamma^2}{8\alpha^2} \| \boldsymbol{z}_0 \|_{\ell_2} \\
&\leq \frac{R}{2},
\end{aligned}
$$

where the last line is directly due to inequality (24), $\gamma \leq 5$, and $\alpha \leq \beta$. Hence, we have established $\boldsymbol{\theta}_t \in \mathcal{B}_R(\boldsymbol{\theta}_0)$.

**Part II:** In Lemma A.3 we showed that the time invariant system of state equations $\widetilde{\boldsymbol{z}}_{t+1} = \boldsymbol{A}\widetilde{\boldsymbol{z}}_t$ is uniformly exponentially stable, i.e. $\| \widetilde{\boldsymbol{z}}_t \|_{\ell_2}$ goes down to zero exponentially fast. Now by using the assumption that the Jacobian remains close to the Jacobian at the initialization $\mathcal{J}_0$, we aim to show the exponential stability of the time variant system of the state equations 16. For that, we compute

$$
\begin{aligned}
\boldsymbol{z}_t = \boldsymbol{A}_{t-1}\boldsymbol{z}_{t-1} &= \begin{bmatrix} \boldsymbol{I} & -\eta \mathcal{J}_{t,t-1}\mathcal{J}_{t-1}^T \\ \mu \boldsymbol{I} & (1-\mu)\boldsymbol{I} \end{bmatrix} \boldsymbol{z}_{t-1} \\
&= \begin{bmatrix} \boldsymbol{I} & -\eta \mathcal{J}_0 \mathcal{J}_0^T \\ \mu \boldsymbol{I} & (1-\mu)\boldsymbol{I} \end{bmatrix} \boldsymbol{z}_{t-1} + \begin{bmatrix} \eta\left(\mathcal{J}_0\mathcal{J}_0^T - \mathcal{J}_{t,t-1}\mathcal{J}_{t-1}^T\right)\boldsymbol{d}_{t-1} \\ \boldsymbol{0} \end{bmatrix} \\
&=: \boldsymbol{A}\boldsymbol{z}_{t-1} + \boldsymbol{\Delta}_{t-1}.
\end{aligned}
$$

Now set $\lambda := 1 - \eta\alpha^2$. By induction, we obtain the relation $z_t = A^t z_0 + \sum_{i=0}^{t-1} A^{t-1-i}\Delta_i$. Hence,

$$
\begin{aligned}
\|z_t\|_{\ell_2} &= \left\| A^t z_0 + \sum_{i=0}^{t-1} A^{t-1-i}\Delta_i \right\|_{\ell_2} \\
&\leq \|A^t z_0\|_{\ell_2} + \sum_{i=0}^{t-1} \|A^{t-1-i}\Delta_i\|_{\ell_2} \\
&\leq \gamma\lambda^t \|z_0\|_{\ell_2} + \sum_{i=0}^{t-1} \gamma\lambda^{t-1-i} \left\| \eta \left( \mathcal{J}_0\mathcal{J}_0^T - \mathcal{J}_{i+1,i}\mathcal{J}_i^T \right) \right\| \|d_i\|_{\ell_2} \\
&\leq \gamma\lambda^t \|z_0\|_{\ell_2} + \sum_{i=0}^{t-1} \eta\gamma\lambda^{t-1-i} (2\beta\epsilon) \|z_i\|_{\ell_2}.
\end{aligned}
$$
(31)

The second inequality holds because of Lemma A.3. The last inequality holds because by combining our assumptions 1 to 3 with $\theta_t \in \mathcal{B}_R(\theta_0)$ and the induction assumption 28 for $0 \leq i \leq t-1$, we have that

$$
\begin{aligned}
\|\mathcal{J}_0\mathcal{J}_0^T - \mathcal{J}_{i+1,i}\mathcal{J}_i^T\| &= \|\mathcal{J}_0\mathcal{J}_0^T - \mathcal{J}_0\mathcal{J}_i^T + \mathcal{J}_0\mathcal{J}_i^T - \mathcal{J}_{i+1,i}\mathcal{J}_i^T\| \\
&\leq \|\mathcal{J}_0\| \|\mathcal{J}_0 - \mathcal{J}_i\| + \|\mathcal{J}_0 - \mathcal{J}_{i+1,i}\| \|\mathcal{J}_i\| \\
&\leq \beta \|\mathcal{J}_0 - \mathcal{J}_i\| + \beta \|\mathcal{J}_0 - \mathcal{J}_{i+1,i}\| \\
&\leq 2\beta\epsilon.
\end{aligned}
$$
(32)

In order to deal with inequality 31, we will rely on the following lemma.

**Lemma A.5** *(Rugh, 1996, Lemma 24.5) Consider two real sequences $p(t)$ and $\phi(t)$, where $p(t) \geq 0$ for all $t \geq 0$ and*

$$
\phi(t) \leq \begin{cases} \psi, & \text{if } t = 0 \\ \psi + \eta \sum_{i=0}^{t-1} p(i)\phi(i), & \text{if } t \geq 1 \end{cases}
$$

*where $\eta$ and $\psi$ are constants with $\eta \geq 0$. Then for all $t \geq 1$ we have*

$$
\phi(t) \leq \psi \prod_{i=0}^{t-1} (1 + \eta \cdot p(i)).
$$

Now we define $\phi_t = \frac{\|z_t\|_{\ell_2}}{\lambda^t}$ and rewrite inequality 31 as

$$
\phi_t \leq \gamma\phi_0 + \sum_{i=0}^{t-1} \frac{2\eta\gamma\beta\epsilon}{\lambda}\phi_i.
$$

Hence, Lemma A.5 yields that

$$
\begin{aligned}
\phi_t &\leq \gamma\phi_0 \prod_{i=0}^{t-1} \left( 1 + \frac{2\eta\gamma\beta\epsilon}{\lambda} \right) \\
&= \gamma\phi_0 \left( 1 + \frac{2\eta\gamma\beta\epsilon}{\lambda} \right)^t \\
&\overset{(i)}{\leq} \gamma\phi_0 \left( 1 + \frac{\eta\alpha^2}{2\lambda} \right)^t \\
&\overset{(ii)}{=} \gamma\phi_0 \left( \frac{1 - \frac{\eta\alpha^2}{2}}{1 - \eta\alpha^2} \right)^t,
\end{aligned}
$$

where $(i)$ follows from $4\gamma\beta\epsilon \le \alpha^2$ and $(ii)$ holds by inserting $\lambda = 1 - \eta\alpha^2$. Inserting the definition of $\phi_0$ and $\phi_t$ we obtain that

$$\|\boldsymbol{z}_t\|_{\ell_2} \le \gamma \left(1 - \frac{\eta\alpha^2}{2}\right)^t \|\boldsymbol{z}_0\|_{\ell_2}.$$

This completes the proof of Part II.

**Part III:** In this part, our aim is to show that the error vector $\boldsymbol{e}_t := \boldsymbol{z}_t - \widetilde{\boldsymbol{z}}_t$ obeys inequality 26. First, note that

$$
\begin{aligned}
\boldsymbol{e}_t &= \boldsymbol{z}_t - \widetilde{\boldsymbol{z}}_t \\
&\overset{(*)}{=} (\boldsymbol{A}\boldsymbol{z}_{t-1} + \boldsymbol{\Delta}_{t-1}) - \boldsymbol{A}\widetilde{\boldsymbol{z}}_{t-1} \\
&= \boldsymbol{A}\boldsymbol{e}_{t-1} + \boldsymbol{\Delta}_{t-1},
\end{aligned}
$$

where in $(*)$ we used the same notation as in Part II for $\boldsymbol{\Delta}_{t-1}$. Using a recursive argument as well as $\boldsymbol{e}_0 = 0$ we obtain that

$$
\begin{aligned}
\|\boldsymbol{e}_t\|_{\ell_2} &= \left\|\sum_{i=0}^{t-1} \boldsymbol{A}^{t-1-i} \boldsymbol{\Delta}_i\right\|_{\ell_2} \\
&\le \sum_{i=0}^{t-1} \eta\gamma \left(1 - \eta\alpha^2\right)^{t-1-i} \|\boldsymbol{\Delta}_i\|_{\ell_2} \\
&\overset{(i)}{\le} \sum_{i=0}^{t-1} \eta\gamma \left(1 - \eta\alpha^2\right)^{t-1-i} \|\mathcal{J}_0\mathcal{J}_0^T - \mathcal{J}_{i+1,i}\mathcal{J}_i^T\| \|\boldsymbol{d}_i\|_{\ell_2} \\
&\overset{(ii)}{\le} \sum_{i=0}^{t-1} 2\eta\beta\epsilon\gamma \left(1 - \eta\alpha^2\right)^{t-1-i} \|\boldsymbol{z}_i\|_{\ell_2}.
\end{aligned}
$$

The first inequality follows from the triangle inequality and Lemma A.3. Inequality $(i)$ follows from the definition of $\boldsymbol{\Delta}_i$. Inequality $(ii)$ follows from inequality 32. Setting $c := 2\eta\beta\epsilon$ we continue

$$
\begin{aligned}
\|\boldsymbol{e}_t\|_{\ell_2} &\le \sum_{i=0}^{t-1} c\gamma \left(1 - \eta\alpha^2\right)^{t-i-1} \|\boldsymbol{z}_i\|_{\ell_2} \\
&\overset{(iii)}{\le} \sum_{i=0}^{t-1} c\gamma^2 \left(1 - \eta\alpha^2\right)^{t-i-1} \left(1 - \frac{\eta\alpha^2}{2}\right)^i \|\boldsymbol{z}_0\|_{\ell_2} \\
&\overset{(iv)}{\le} \sum_{i=0}^{t-1} c\gamma^2 \left(1 - \frac{\eta\alpha^2}{2}\right)^{t-1} \|\boldsymbol{z}_0\|_{\ell_2} \\
&= 2\eta\gamma^2\beta\epsilon \cdot t \left(1 - \frac{\eta\alpha^2}{2}\right)^{t-1} \|\boldsymbol{z}_0\|_{\ell_2}.
\end{aligned}
$$

Here $(iii)$ holds because of our induction hypothesis 25 and $(iv)$ follows simply from $1 - \eta\alpha^2 \le 1 - \frac{\eta\alpha^2}{2}$. This shows the first part of equation 26 for iteration t. Finally, to derive the second part of equation 26 we observe that for all $t \ge 0$ and $0 < x \le \frac{1}{16}$ we have $t(1-x)^{t-1} \le \frac{1}{e\left(15\ln\frac{16}{15}\right)x}$. Since $\frac{\eta\alpha^2}{2} \le \frac{\mu\alpha^2}{16\beta^2} \le \frac{1}{16}$ we can use this estimate, which yields

$$
\begin{aligned}
\|\boldsymbol{e}_t\|_{\ell_2} &\le 2\eta\gamma^2\beta\epsilon \cdot t \left(1 - \frac{\eta\alpha^2}{2}\right)^{t-1} \|\boldsymbol{z}_0\|_{\ell_2} \\
&\le \frac{4\gamma^2\beta\epsilon}{e\left(15\ln\frac{16}{15}\right)\alpha^2} \|\boldsymbol{z}_0\|_{\ell_2}.
\end{aligned}
$$

Hence, we have shown equation 26.

**Part IV:** In this part, we aim to show that the parameters of the original and linearized problems are close. For that, we compute that

$$
\frac{1}{\eta}\left\|\boldsymbol{\theta}_t - \widetilde{\boldsymbol{\theta}}_t\right\|_{\ell_2} = \left\|\sum_{i=0}^{t-1}\nabla_{\boldsymbol{\theta}}h\left(\boldsymbol{\theta}_i, \boldsymbol{d}_i\right) - \nabla_{\boldsymbol{\theta}}h_{\lin}\left(\boldsymbol{\theta}_i, \boldsymbol{d}_i\right)\right\|_{\ell_2}
$$

$$
= \left\|\sum_{i=0}^{t-1}\mathcal{J}^T\left(\boldsymbol{\theta}_i\right)\boldsymbol{d}_i - \mathcal{J}_0^T\widetilde{\boldsymbol{d}}_i\right\|_{\ell_2}
$$

$$
\leq \sum_{i=0}^{t-1}\left\|\left(\mathcal{J}^T\left(\boldsymbol{\theta}_i\right) - \mathcal{J}_0^T\right)\widetilde{\boldsymbol{d}}_i\right\|_{\ell_2} + \sum_{i=0}^{t-1}\left\|\mathcal{J}^T\left(\boldsymbol{\theta}_i\right)\left(\boldsymbol{d}_i - \widetilde{\boldsymbol{d}}_i\right)\right\|_{\ell_2}
$$

$$
\overset{(i)}{\leq} \sum_{i=0}^{t-1}\epsilon\left\|\widetilde{\boldsymbol{z}}_i\right\|_{\ell_2} + \beta\sum_{i=0}^{t-1}\left\|\boldsymbol{e}_i\right\|_{\ell_2}
$$

$$
\overset{(ii)}{\leq} \gamma\epsilon\sum_{i=0}^{t-1}\left(1 - \eta\alpha^2\right)^i\left\|\boldsymbol{z}_0\right\|_{\ell_2} + 2\eta\gamma^2\beta^2\epsilon\sum_{i=0}^{t-1}i\left(1 - \frac{\eta\alpha^2}{2}\right)^{i-1}\left\|\boldsymbol{z}_0\right\|_{\ell_2}.
$$

Here $(i)$ follows from assumptions 2 and 3, and $(ii)$ holds because of Lemma A.3 and our induction hypothesis 26. Hence, using the formula $\sum_{i=0}^{t}ix^i = \frac{x\left(1 + tx^{t+1} - (t+1)x^t\right)}{(x-1)^2}$ we obtain that

$$
\frac{1}{\eta}\left\|\boldsymbol{\theta}_t - \widetilde{\boldsymbol{\theta}}_t\right\|_{\ell_2} \leq \gamma\epsilon\left\|\boldsymbol{z}_0\right\|_{\ell_2}\left(\frac{1 - \left(1 - \eta\alpha^2\right)^t}{\eta\alpha^2} + 2\eta\beta^2\gamma\frac{1 - t\left(1 - \frac{\eta\alpha^2}{2}\right)^{t-1} + (t-1)\left(1 - \frac{\eta\alpha^2}{2}\right)^t}{\left(\frac{\eta\alpha^2}{2}\right)^2}\right)
$$

$$
\leq \gamma\epsilon\left\|\boldsymbol{z}_0\right\|_{\ell_2}\left(\frac{1}{\eta\alpha^2} + 2\eta\beta^2\gamma\frac{1}{\left(\frac{\eta\alpha^2}{2}\right)^2}\right)
$$

$$
\overset{(iii)}{\leq} \gamma\epsilon\left\|\boldsymbol{z}_0\right\|_{\ell_2}\left(\frac{\beta^2\gamma}{\eta\alpha^4} + \frac{8\beta^2\gamma}{\eta\alpha^4}\right)
$$

$$
= \frac{9\epsilon\beta^2\gamma^2}{\eta\alpha^4}\left\|\boldsymbol{z}_0\right\|_{\ell_2},
$$

where $(iii)$ holds due to $1 \leq \gamma$ and $1 \leq \frac{\beta^2}{\alpha^2}$. Hence, we have established inequality 27 for iteration $t$.

**Part V:** In this part, we are going to prove equation 28 for iteration $t$. First, it follows from the triangle inequality that

$$
\left\|\boldsymbol{\theta}_t - \boldsymbol{\theta}_0\right\|_{\ell_2} \leq \left\|\widetilde{\boldsymbol{\theta}}_t - \boldsymbol{\theta}_0\right\|_{\ell_2} + \left\|\boldsymbol{\theta}_t - \widetilde{\boldsymbol{\theta}}_t\right\|_{\ell_2}
$$

$$
\leq \left\|\widetilde{\boldsymbol{\theta}}_t - \boldsymbol{\theta}_0\right\|_{\ell_2} + \frac{9\epsilon\beta^2\gamma^2}{\alpha^4}\left\|\boldsymbol{z}_0\right\|_{\ell_2},
$$

where in the second inequality we have used Part IV. Now we bound $\left\|\widetilde{\boldsymbol{\theta}}_t - \boldsymbol{\theta}_0\right\|_{\ell_2}$ from above as follows

$$
\begin{aligned}
\left\|\widetilde{\boldsymbol{\theta}}_t - \boldsymbol{\theta}_0\right\|_{\ell_2} &= \eta \left\|\sum_{i=0}^{t-1} \mathcal{J}_0^T \widetilde{\boldsymbol{d}}_i\right\|_{\ell_2} \\
&\leq \eta \sum_{i=0}^{t-1} \left\|\mathcal{J}_0^T \widetilde{\boldsymbol{d}}_i\right\|_{\ell_2} \\
&\overset{(i)}{\leq} \eta\gamma \sum_{i=0}^{t-1} \left(1 - \eta\alpha^2\right)^i \left\|\begin{bmatrix} \mathcal{J}_0^T & 0 \\ 0 & \mathcal{J}_0^T \end{bmatrix} \boldsymbol{z}_0\right\|_{\ell_2} \\
&= \eta\gamma \frac{1 - \left(1 - \eta\alpha^2\right)^t}{\eta\alpha^2} \left\|\begin{bmatrix} \mathcal{J}_0^T & 0 \\ 0 & \mathcal{J}_0^T \end{bmatrix} \boldsymbol{z}_0\right\|_{\ell_2} \\
&\overset{(ii)}{\leq} \gamma \frac{\beta^2}{\alpha^2} \left\|\begin{bmatrix} \mathcal{J}_0^\dagger & 0 \\ 0 & \mathcal{J}_0^\dagger \end{bmatrix} \boldsymbol{z}_0\right\|_{\ell_2},
\end{aligned}
$$

where $(i)$ holds by equation 30 and $(ii)$ holds by equation 29. Hence, it follows from the definition of $R$ (equation 24) that

$$
\begin{aligned}
\|\boldsymbol{\theta}_t - \boldsymbol{\theta}_0\|_{\ell_2} &\leq \gamma \frac{\beta^2}{\alpha^2} \left\|\begin{bmatrix} \mathcal{J}_0^\dagger & 0 \\ 0 & \mathcal{J}_0^\dagger \end{bmatrix} \boldsymbol{z}_0\right\|_{\ell_2} + \frac{9\epsilon\beta^2\gamma^2}{\alpha^4} \|\boldsymbol{z}_0\|_{\ell_2} \\
&= \frac{R}{2}.
\end{aligned}
$$

This completes the proof.

## A.4 PRELIMINARIES FOR PROOFS OF RESULTS WITH ONE-HIDDEN LAYER GENERATOR AND LINEAR DISCRIMINATOR

In this section, we gather some preliminary results that will be useful in proving the main results i.e. Theorems 2.1 and 2.2. We begin by noting that Theorem 2.1 is an instance of Theorem A.4 with $f(\boldsymbol{W}) = \frac{1}{n} \sum_{i=1}^n \boldsymbol{V} \cdot \phi(\boldsymbol{W}\boldsymbol{z}_i)$. We thus begin this section by noting that $f(\boldsymbol{W})$ can be rewritten as follows

$$
f(\boldsymbol{W}) = \boldsymbol{V} \cdot \begin{bmatrix} \frac{1}{n} \sum_{i=1}^n \phi\left(\boldsymbol{w}_1^T \boldsymbol{z}_i\right) \\ \cdot \\ \cdot \\ \cdot \\ \frac{1}{n} \sum_{i=1}^n \phi\left(\boldsymbol{w}_k^T \boldsymbol{z}_i\right) \end{bmatrix}.
$$

Furthermore, the Jacobian of this mapping $f(\boldsymbol{W})$ takes the form

$$
\mathcal{J}(\boldsymbol{W}) = \frac{1}{n} \sum_{i=1}^n \left(\boldsymbol{V} \cdot \operatorname{diag}\left(\phi'\left(\boldsymbol{W}\boldsymbol{z}_i\right)\right)\right) \otimes \boldsymbol{z}_i^T. \tag{33}
$$

To characterize the spectral properties of this Jacobian it will be convenient to write down the expression for $\mathcal{J}(\boldsymbol{W})\mathcal{J}(\boldsymbol{W})^T$ which has a compact form

$$
\begin{aligned}
\mathcal{J}(\boldsymbol{W})\mathcal{J}(\boldsymbol{W})^T &\overset{(i)}{=} \frac{1}{n^2} \sum_{i,j=1}^n \left(\left(\boldsymbol{V} \cdot \operatorname{diag}\left(\phi'\left(\boldsymbol{W}\boldsymbol{z}_i\right)\right)\right) \otimes \boldsymbol{z}_i^T\right)\left(\operatorname{diag}\left(\phi'\left(\boldsymbol{W}\boldsymbol{z}_j\right)\right)\boldsymbol{V}^T \otimes \boldsymbol{z}_j\right) \\
&\overset{(ii)}{=} \frac{1}{n^2} \sum_{i,j=1}^n \left(\boldsymbol{V} \operatorname{diag}\left(\phi'\left(\boldsymbol{W}\boldsymbol{z}_i\right)\right) \operatorname{diag}\left(\phi'\left(\boldsymbol{W}\boldsymbol{z}_j\right)\right)\boldsymbol{V}^T\right) \otimes \left(\boldsymbol{z}_i^T \boldsymbol{z}_j\right) \\
&= \frac{1}{n^2} \boldsymbol{V} \operatorname*{diag}_{\ell=1,\ldots,k} \left(\left\|\sum_{i=1}^n \boldsymbol{z}_i \phi'\left(\boldsymbol{w}_\ell^T \boldsymbol{z}_i\right)\right\|_{\ell_2}^2\right) \boldsymbol{V}^T \\
&= \frac{1}{n^2} \boldsymbol{V} \cdot \boldsymbol{D}^2 \cdot \boldsymbol{V}^T,
\end{aligned}
$$

where $D$ is a diagonal matrix with entries

$$D_{\ell\ell} = \left\| \sum_{i=1}^n z_i \phi'(w_\ell^T z_i) \right\|_{\ell_2} = \left\| Z^T \phi'(Z w_\ell) \right\|_{\ell_2}, \tag{34}$$

and $Z \in \mathbb{R}^{n \times d}$ contains the $z_i$'s in its rows. Note that we used simple properties of kronecker product in (i) and (ii), namely $(A \otimes B)^T = A^T \otimes B^T$ and $(A \otimes B)(C \otimes D) = (AC) \otimes (BD)$. The next lemma establishes concentration of the diagonal entries of matrix $D^2$ around their mean, which will be used in the future lemmas regarding the spectrum of the Jacobian mapping. The proof is deferred to Appendix B.1.

**Lemma A.6** *Suppose $w \in \mathbb{R}^d$ is a fixed vector, $z_1, z_2, \cdots, z_n \in \mathbb{R}^d$ are distributed as $\mathcal{N}(0, \sigma_z^2 I_d)$ and constitute the rows of $Z \in \mathbb{R}^{n \times d}$. Then for any $0 \le \delta \le \frac{3}{2}$ the random variable $D = \left\| Z^T \phi'(Z w) \right\|_{\ell_2}$ satisfies*

$$(1 - \delta) \mathbb{E}(D^2) \le D^2 \le (1 + \delta) \mathbb{E}(D^2)$$

*with probability at least $1 - 2\left(e^{-\frac{n\delta^2}{18}} + e^{-\frac{d\delta^2}{54}} + e^{-c_1 n\delta}\right)$ where $c_1$ is a positive constant. Moreover we have*

$$\mathbb{E}(D^2) = \sigma_z^2 \left( \frac{nd}{2} + \frac{n(n-1)}{2\pi} \right).$$

*Furthermore, using the above equation we have*

$$\mathbb{E}\left[ \mathcal{J}(W) \mathcal{J}(W)^T \right] = \frac{\sigma_z^2 \left( d + \frac{n-1}{\pi} \right)}{2n} V V^T.$$

## A.5 Lemmas regarding the initial misfit and the spectrum of the Jacobian

In this section, we state some lemmas regarding the spectrum of the Jacobian mapping and the initial misfit, and defer their proofs to Appendix B. First, we state a result on the minimum singular value of the Jacobian mapping at initialization.

**Lemma A.7** *(Minimum singular value of the Jacobian at initialization) Consider our GAN model with a linear discriminator and the generator being a one hidden layer neural network of the form $z \rightarrowtail V \phi(W z)$, where we have n independent data points $z_1, z_2, \cdots, z_n \in \mathbb{R}^d$ distributed as $\mathcal{N}(0, \sigma_z^2 I_d)$ and aggregated as the rows of a matrix $Z \in \mathbb{R}^{n \times d}$, and $V \in \mathbb{R}^{m \times k}$ has i.i.d $\mathcal{N}(0, \sigma_v^2)$ entries. We also assume that $W_0 \in \mathbb{R}^{k \times d}$ has i.i.d $\mathcal{N}(0, \sigma_w^2)$ entries and all entries of $W_0$, $V$, and $Z$ are independent. Then the Jacobian matrix at the initialization point obeys*

$$\sigma_{min}(\mathcal{J}(W_0)) \ge \left( \sqrt{(1-\delta)^2 k - (1+\delta)^2} - \sqrt{m}(1+\eta)(1+\delta) \right) \sigma_v \sigma_z \sqrt{\frac{d + \frac{n-1}{\pi}}{2n}}, \quad 0 \le \delta \le \frac{3}{2}$$

*with probability at least $1 - 3e^{-\frac{\eta^2 m}{8}} - 2k \cdot \left( e^{-\frac{n\delta^2}{18}} + e^{-\frac{d\delta^2}{54}} + e^{-c_1 n\delta} \right)$, where $c_1$ is a positive constant.*

Next lemma helps us bound the spectral norm of the Jacobian at initialization, which will be used later to derive upper bounds on Jacobian at every point near initialization.

**Lemma A.8** *(spectral norm of the Jacobian at initialization) Following the setup of previous lemma, the operator norm of the Jacobian matrix at initialization point $W_0 \in \mathbb{R}^{k \times d}$ satisfies*

$$\|\mathcal{J}(W_0)\| \le (1 + \delta) \sigma_v \sigma_z \left( \sqrt{k} + 2\sqrt{m} \right) \sqrt{\frac{d + \frac{n-1}{\pi}}{2n}}, \quad 0 \le \delta \le \frac{3}{2}$$

*with probability at least $1 - e^{-\frac{m}{2}} - k \cdot \left( e^{-\frac{n\delta^2}{18}} + e^{-\frac{d\delta^2}{54}} + e^{-c_1 n\delta} \right)$, with $c_1$ a positive constant.*

The next lemma is adapted from Van Veen et al. (2018) and allows us to bound the variations in the Jacobian matrix around initialization.

**Lemma A.9** (*single-sample Jacobian perturbation*) *Let* $V \in \mathbb{R}^{m \times k}$ *be a matrix with i.i.d.* $\mathcal{N}(0, \sigma_v^2)$ *entries,* $W \in \mathbb{R}^{k \times d}$, *and define the Jacobian mapping* $\mathcal{J}(W; z) = (V \, diag \, (\phi'(Wz))) \otimes z^T$. *Then, by taking* $W_0$ *to be a random matrix with i.i.d.* $\mathcal{N}(0, \sigma_w^2)$ *entries, we have*

$$\|\mathcal{J}(W; z) - \mathcal{J}(W_0; z)\| \leq \sigma_v \|z\|_{\ell_2} \left( 2\sqrt{m} + \sqrt{6 \left( \frac{2kR}{\sigma_w} \right)^{\frac{2}{3}} log \left( \frac{k}{3 \left( \frac{2kR}{\sigma_w} \right)^{\frac{2}{3}}} \right)} \right)$$

*for all* $W \in \mathbb{R}^{k \times d}$ *obeying* $\|W - W_0\| \leq R$ *with probability at least* $1 - e^{-\frac{m}{2}} - e^{-\frac{\left( \frac{2kR}{\sigma_w} \right)^{\frac{2}{3}}}{6}}$.

Our final key lemma bounds the initial misfit $f(W_0) - y := \frac{1}{n} \sum_{i=1}^n V \phi(W_0 z_i) - \bar{x}$.

**Lemma A.10** (*Initial misfit*) *Consider our GAN model with a linear discriminator and the generator being a one hidden layer neural network of the form* $z \rightarrowtail V\phi(Wz)$, *where we have n independent data points* $z_1, z_2, \cdots, z_n \in \mathbb{R}^d$ *distributed as* $\mathcal{N}(0, \sigma_z^2 I_d)$ *and aggregated as the rows of a matrix* $Z \in \mathbb{R}^{n \times d}$, *and* $V \in \mathbb{R}^{m \times k}$ *has i.i.d* $\mathcal{N}(0, \sigma_v^2)$ *entries. We also assume that the initial* $W_0 \in \mathbb{R}^{k \times d}$ *has i.i.d* $\mathcal{N}(0, \sigma_w^2)$ *entries. Then the following event*

$$\left\| \frac{1}{n} \sum_{i=1}^n V \phi(W_0 z_i) - \bar{x} \right\|_{\ell_2} \leq (1 + \delta) \frac{1}{\sqrt{2\pi}} \sigma_v \sigma_w \sigma_z \sqrt{kdm} + \|\bar{x}\|_{\ell_2}, \quad 0 \leq \delta \leq 3$$

*holds with probability at least* $1 - \left( k \cdot e^{-c_2 n (\delta/27)^2} + e^{-\frac{(\delta/9)^2 m}{2}} + e^{-\frac{(\delta/3)^2 kd}{2}} \right)$, *with* $c_2$ *a fixed constant.*

## A.6 PROOF OF THEOREM 2.1

In this section, we prove Theorem 2.1 by using our general meta Theorem A.4. To do this we need to check that Assumptions 1-3 are satisfied with high probability. Specifically, in our case the parameter $\theta$ is the matrix $W$ and the non-linear mapping $f$ is given by $f(W) = \frac{1}{n} \sum_{i=1}^n V \phi(Wz_i)$. We note that in our result $d_0 = 0$ and thus $\|z_0\|_{\ell_2} = \|r_0\|_{\ell_2}$, which simplifies our analysis.

To prove Assumption 1 note that by setting $\delta = \frac{1}{2}$ and $\eta = \frac{1}{3}$ in Lemma A.7, we have

$$\sigma_{\min}(\mathcal{J}(W_0)) \geq \sigma_v \sigma_z \left( \frac{1}{2} \sqrt{k - 9} - 2\sqrt{m} \right) \sqrt{\frac{d + \frac{n-1}{\pi}}{2n}}$$

$$=: \alpha.$$

This holds with probability at least $1 - 3e^{-\frac{m}{72}} - 4k \cdot e^{-c \cdot n} - 2k \cdot e^{-\frac{d}{216}}$, concluding the proof of Assumption 1. Next, by setting $\delta = \frac{1}{2}$ in Lemma A.8 we have

$$\|\mathcal{J}(W_0)\| \leq \zeta := \frac{3}{2} \sigma_v \sigma_z \left( \sqrt{k} + 2\sqrt{m} \right) \sqrt{\frac{d + \frac{n-1}{\pi}}{2n}}$$

with probability at least $1 - e^{-\frac{m}{2}} - 2k \cdot e^{-c \cdot n} - k \cdot e^{-\frac{d}{216}}$. Now to bound spectral norm of Jacobian at $W$ where $\|W - W_0\| \leq R$ (the value of R is defined in the proof of assumption 3 below), we use triangle inequality to get

$$\|\mathcal{J}(W)\| \leq \|\mathcal{J}(W_0)\| + \|\mathcal{J}(W) - \mathcal{J}(W_0)\|.$$

This last inequality together with assumption 3, which we will prove below, yields

$$\|\mathcal{J}(W)\| \leq \|\mathcal{J}(W_0)\| + \epsilon \leq \|\mathcal{J}(W_0)\| + \frac{\alpha^2}{4\gamma\beta} \leq \|\mathcal{J}(W_0)\| + \frac{\|\mathcal{J}(W_0)\|^2}{4\beta}.$$

Therefore by choosing $\beta = 2\zeta$ we arrive at

$$
\begin{aligned}
\|\mathcal{J}(\boldsymbol{W})\| &\leq \|\mathcal{J}(\boldsymbol{W}_0)\| + \frac{\|\mathcal{J}(\boldsymbol{W}_0)\|^2}{4\beta} \\
&= \|\mathcal{J}(\boldsymbol{W}_0)\| + \frac{\|\mathcal{J}(\boldsymbol{W}_0)\|^2}{8\zeta} \\
&\leq \|\mathcal{J}(\boldsymbol{W}_0)\| + \frac{\|\mathcal{J}(\boldsymbol{W}_0)\|^2}{8\|\mathcal{J}(\boldsymbol{W}_0)\|} \\
&\leq 2\|\mathcal{J}(\boldsymbol{W}_0)\| \\
&\leq 2\zeta = \beta,
\end{aligned}
$$

establishing that assumption 2 holds with

$$
\beta = 3\sigma_v \sigma_z \left(\sqrt{k} + 2\sqrt{m}\right) \sqrt{\frac{d + \frac{n-1}{\pi}}{2n}}
$$

with probability at least $1 - e^{-\frac{m}{2}} - 2k \cdot e^{-c \cdot n} - k \cdot e^{-\frac{d}{216}}$.

Finally to show that Assumption 3 holds, we use the single-sample Jacobian perturbation result of Lemma A.9 combined with the triangle inequality to conclude that

$$
\begin{aligned}
\|\mathcal{J}(\boldsymbol{W}) - \mathcal{J}(\boldsymbol{W}_0)\| &= \left\| \frac{1}{n} \left( \sum_{i=1}^n \mathcal{J}(\boldsymbol{W}; z_i) - \mathcal{J}(\boldsymbol{W}_0; z_i) \right) \right\| \\
&\leq \frac{1}{n} \sum_{i=1}^n \|\mathcal{J}(\boldsymbol{W}; z_i) - \mathcal{J}(\boldsymbol{W}_0; z_i)\| \\
&\leq \frac{\sigma_v}{n} \left( \sum_{i=1}^n \|z_i\|_{\ell_2} \right) \left( 2\sqrt{m} + \sqrt{6 \left(\frac{2kR}{\sigma_w}\right)^{\frac{2}{3}} \log\left(\frac{k}{3\left(\frac{2kR}{\sigma_w}\right)^{\frac{2}{3}}}\right)} \right) \\
&\overset{(i)}{\leq} \sigma_v \frac{\|\boldsymbol{Z}\|_F}{\sqrt{n}} \left( 2\sqrt{m} + \sqrt{6 \left(\frac{2kR}{\sigma_w}\right)^{\frac{2}{3}} \log\left(\frac{k}{3\left(\frac{2kR}{\sigma_w}\right)^{\frac{2}{3}}}\right)} \right) \\
&\overset{(ii)}{\leq} \frac{5}{4}\sigma_v \sigma_z \sqrt{d} \left( 2\sqrt{m} + \sqrt{6 \left(\frac{2kR}{\sigma_w}\right)^{\frac{2}{3}} \log\left(\frac{k}{3\left(\frac{2kR}{\sigma_w}\right)^{\frac{2}{3}}}\right)} \right),
\end{aligned} \tag{35}
$$

where $(i)$ holds by Cauchy–Schwarz inequality, and $(ii)$ holds because for a Gaussian matrix $\boldsymbol{Z} \in \mathbb{R}^{n \times d}$ with $\mathcal{N}(0, \sigma_z^2)$ entries the following holds

$$
\mathbb{P}\left( \|\boldsymbol{Z}\|_F \leq \frac{5}{4}\sigma_z \sqrt{nd} \right) \geq \mathbb{P}\left( \|\boldsymbol{Z}\|_F^2 \leq \frac{3}{2}\sigma_z^2 nd \right) \geq 1 - e^{-\frac{nd}{24}}.
$$

Now we set $\epsilon = \frac{\alpha^2}{4\gamma\beta}$ and show that Assumption 3 holds with this choice of $\epsilon$ and with radius $\widetilde{R}$, whose value will be defined later in the proof. First, note that

$$
\begin{aligned}
\epsilon &= \frac{\alpha^2}{4\gamma\beta} \\
&= \frac{\sigma_v^2 \sigma_z^2 \left(\frac{1}{2}\sqrt{k-9} - 2\sqrt{m}\right)^2 \left(\frac{d + \frac{n-1}{\pi}}{2n}\right)}{12\gamma\sigma_v\sigma_z \left(\sqrt{k} + 2\sqrt{m}\right)\sqrt{\frac{d + \frac{n-1}{\pi}}{2n}}} \\
&\stackrel{(i)}{\geq} \frac{\sigma_v\sigma_z \left(\frac{1}{8}\sqrt{k}\right)^2 \cdot \sqrt{\frac{1}{4\pi}}}{60\left(3\sqrt{k}\right)} \\
&\geq \frac{\sigma_v\sigma_z\sqrt{k}}{42000},
\end{aligned}
$$

where $(i)$ holds by assuming $k \geq C \cdot m$ with $C$ being a large positive constant. Combining the last inequality with equation 35, we observe that a sufficient condition for assumption 3 to hold is

$$
\frac{5}{4}\sigma_v\sigma_z\sqrt{d}\left(2\sqrt{m} + \sqrt{6\left(\frac{2kR}{\sigma_w}\right)^{\frac{2}{3}}\log\left(\frac{k}{3\left(\frac{2kR}{\sigma_w}\right)^{\frac{2}{3}}}\right)}\right) \leq \frac{\sigma_v\sigma_z\sqrt{k}}{42000},
$$

which is equivalent to

$$
105000\sqrt{md} + 52500 \cdot \sqrt{d} \cdot \sqrt{6\left(\frac{2kR}{\sigma_w}\right)^{\frac{2}{3}}\log\left(\frac{k}{3\left(\frac{2kR}{\sigma_w}\right)^{\frac{2}{3}}}\right)} \leq \sqrt{k}.
$$

Now the first term in the L.H.S. is upper bounded by $\frac{1}{2}\sqrt{k}$ if $k \geq (210000)^2\,md$, and for the second term we need

$$
105000 \cdot \sqrt{d} \cdot \sqrt{6\left(\frac{2kR}{\sigma_w}\right)^{\frac{2}{3}}\log\left(\frac{k}{3\left(\frac{2kR}{\sigma_w}\right)^{\frac{2}{3}}}\right)} \leq \sqrt{k},
$$

which by defining $x = 3d\left(\frac{2R}{\sigma_w\sqrt{k}}\right)^{\frac{2}{3}}$ is equivalent to

$$
x\log\frac{d}{x} \leq \frac{1}{2 \cdot 105000^2}.
$$

This last inequality holds for $x \leq \frac{c}{\log d}$ with $c < 1$ a sufficiently small positive constant, which translates into

$$
R \leq c\frac{\sigma_w\sqrt{k}}{(d\log d)^{\frac{3}{2}}}. \tag{36}
$$

So far we have shown that Assumption 3 holds with $\epsilon = \frac{\alpha^2}{4\gamma\beta}$ and with radius $\widetilde{R}$ defined as $\widetilde{R} := c\frac{\sigma_w\sqrt{k}}{(d\log d)^{\frac{3}{2}}}$, and we conclude that it holds for any radius $R$ less than $\widetilde{R}$ as well. Now we work with

the definition of $R$ in equation 24 to show that $R \leq \widetilde{R}$:

$$
\begin{aligned}
\frac{R}{2} &= \gamma \frac{\beta^2}{\alpha^2} \left\| \begin{bmatrix} \mathcal{J}_0^\dagger & 0 \\ 0 & \mathcal{J}_0^\dagger \end{bmatrix} \boldsymbol{z}_0 \right\|_{\ell_2} + \frac{9\epsilon \beta^2 \gamma^2}{\alpha^4} \|\boldsymbol{z}_0\|_{\ell_2} \\
&\overset{(i)}{\leq} \gamma \frac{\beta^2}{\alpha^3} \|\boldsymbol{r}_0\|_{\ell_2} + \frac{9 \frac{\alpha^2}{4\gamma\beta} \beta^2 \gamma^2}{\alpha^4} \|\boldsymbol{r}_0\|_{\ell_2} \\
&= \gamma \|\boldsymbol{r}_0\|_{\ell_2} \left( \frac{\beta^2}{\alpha^3} + \frac{9}{4} \frac{\beta}{\alpha^2} \right) \\
&\overset{(ii)}{\leq} 20 \frac{\beta^2}{\alpha^3} \|\boldsymbol{r}_0\|_{\ell_2} \\
&= 20 \frac{\left( 3\sigma_v \sigma_z \left( \sqrt{k} + 2\sqrt{m} \right) \sqrt{\frac{d + \frac{n-1}{\pi}}{2n}} \right)^2}{\left( \sigma_v \sigma_z \left( \frac{1}{2}\sqrt{k-9} - 2\sqrt{m} \right) \sqrt{\frac{d + \frac{n-1}{\pi}}{2n}} \right)^3} \|\boldsymbol{r}_0\|_{\ell_2} \\
&\overset{(iii)}{\leq} C \cdot \frac{1}{\sigma_v \sigma_z \sqrt{k}} \cdot \left( \frac{2}{3} \sigma_v \sigma_w \sigma_z \sqrt{k \cdot d \cdot m} + \|\bar{\boldsymbol{x}}\|_{\ell_2} \right)
\end{aligned}
$$

where $(i)$ holds because $\left\| \mathcal{J}_0^\dagger \right\| \leq \frac{1}{\alpha}$ and $4\gamma\beta\epsilon = \alpha^2$, $(ii)$ holds as $1 \leq \frac{\beta}{\alpha}$ and as we substitute $\gamma = 5$ from Lemma A.3, and $(iii)$ follows from $k \geq C \cdot m$ and using $\delta = \frac{1}{3}$ in Lemma A.10. Now a sufficient condition for equation 36 to hold is that

$$
\frac{1}{\sigma_v \sigma_z \sqrt{k}} \cdot \left( \frac{2}{3} \sigma_v \sigma_w \sigma_z \sqrt{k \cdot d \cdot m} + \|\bar{\boldsymbol{x}}\|_{\ell_2} \right) \leq c \frac{\sigma_w \sqrt{k}}{(d \log d)^{\frac{3}{2}}},
$$

which is equivalent to

$$
\frac{2}{3} \sigma_v \sigma_w \sigma_z \cdot (d \log d)^{\frac{3}{2}} \sqrt{k \cdot d \cdot m} + (d \log d)^{\frac{3}{2}} \|\bar{\boldsymbol{x}}\|_{\ell_2} \leq c \cdot k \sigma_v \sigma_w \sigma_z.
$$

Finally, this inequality is satisfied by assuming $k \geq C \cdot md^4 \log{(d)}^3$ and setting $\sigma_v \sigma_w \sigma_z \geq \frac{\|\bar{\boldsymbol{x}}\|_{\ell_2}}{md^{\frac{5}{2}} \log d^{\frac{3}{2}}}$. This shows that assumption 3 holds with probability at least $1 - ne^{-\frac{m}{2}} - ne^{-c \cdot md^3 \log(d)^2} - k \cdot e^{-c \cdot n} - e^{-\frac{m}{1500}} - e^{-\frac{kd}{162}}$, concluding the proof of Theorem 2.1.

## A.7 PROOF OF THEOREM 2.2

Consider a nonlinear least-squares optimization problem of the form

$$
\min_{\theta \in \mathbb{R}^p} \mathcal{L}(\theta) := \frac{1}{2} \|f(\theta) - y\|_{\ell_2}^2
$$

with $f : \mathbb{R}^p \mapsto \mathbb{R}^m$ and $y \in \mathbb{R}^m$. Suppose the Jacobian mapping associated with $f$ satisfies the following three assumptions.

**Assumption 1** We assume $\sigma_{min}(\mathcal{J}(\theta_0)) \geq 2\alpha$ for a fixed point $\theta_0 \in \mathbb{R}^p$.

**Assumption 2** Let $\| \cdot \|$ be a norm dominated by the Frobenius norm i.e. $\|\theta\| \leq \|\theta\|_F$ holds for all $\theta \in \mathbb{R}^p$. Fix a point $\theta_0$ and a number $R > 0$. For any $\theta$ satisfying $\|\theta - \theta_0\| \leq R$, we have $\|\mathcal{J}(\theta) - \mathcal{J}(\theta_0)\| \leq \frac{\alpha}{3}$.

**Assumption 3** We assume for all $\theta \in \mathbb{R}^p$ obeying $\|\theta - \theta_0\| \leq R$, we have $\|\mathcal{J}(\theta)\| \leq \beta$.

With these assumptions in place we are now ready to state the following result from Oymak & Soltanolkotabi (2020):

**Theorem A.11** *Given $\theta_0 \in \mathbb{R}^p$, suppose assumptions 1, 2, and 3 hold with*

$$
R = \frac{3\|f(\theta_0) - y\|_{\ell_2}}{\alpha}. \tag{37}
$$

*Then, using a learning rate $\eta \leq \frac{1}{3\beta^2}$, all gradient descent updates obey*

$$\|f(\theta_\tau) - y\|_{\ell_2} \leq \left(1 - \eta\alpha^2\right)^\tau \|f(\theta_0) - y\|_{\ell_2} . \tag{38}$$

We are going to apply this theorem in our case where the parameter is $W$, the nonlinear mapping is $f(W) = \frac{1}{n}\sum_{i=1}^{n} V\phi(Wz_i)$ with $\phi = ReLU$, and the norm $\|\cdot\|$ set to the operator norm.

Similar to previous part, by using Lemma A.7 we conclude that with probability at least $1 - 3e^{-\frac{m}{72}} - 4k \cdot e^{-c \cdot n} - 2k \cdot e^{-\frac{d}{216}}$, assumption 1 is satisfied with

$$2\alpha := \sigma_v \sigma_z \left(\frac{1}{2}\sqrt{k-9} - 2\sqrt{m}\right) \sqrt{\frac{d + \frac{n-1}{\pi}}{2n}} .$$

Next we show that assumption 2 is valid for $\alpha$ as defined in the previous line and for radius $\widetilde{R}$ defined later. First we note that

$$\frac{\alpha}{3} \geq c \cdot \sigma_v \sigma_z \cdot \sqrt{k},$$

where the inequality holds by assuming $K \geq C \cdot m$ for a sufficiently large positive constant $C$. Now by using equation 35 assumption 2 holds if

$$\frac{5}{4}\sigma_v\sigma_z\sqrt{d}\left(2\sqrt{m} + \sqrt{6\left(\frac{2kR}{\sigma_w}\right)^{\frac{2}{3}}\log\left(\frac{k}{3\left(\frac{2kR}{\sigma_w}\right)^{\frac{2}{3}}}\right)}\right) \leq c \cdot \sigma_v\sigma_z \cdot \sqrt{k},$$

which is equivalent to

$$C\sqrt{md} + C\sqrt{d} \cdot \sqrt{6\left(\frac{2kR}{\sigma_w}\right)^{\frac{2}{3}}\log\left(\frac{k}{3\left(\frac{2kR}{\sigma_w}\right)^{\frac{2}{3}}}\right)} \leq \sqrt{k}.$$

The first term in the L.H.S. of the inequality above is upper bounded by $\frac{1}{2}\sqrt{k}$ if $k \geq C \cdot md$. For upper bounding the second term it is sufficient to show that

$$C\sqrt{d}\sqrt{6\left(\frac{2kR}{\sigma_w}\right)^{\frac{2}{3}}\log\left(\frac{k}{3 \cdot \left(\frac{2kR}{\sigma_w}\right)^{\frac{2}{3}}}\right)} \leq \sqrt{k},$$

which by defining $x = 3d\left(\frac{2R}{\sigma_w\sqrt{k}}\right)^{\frac{2}{3}}$ is equivalent to $x \cdot \log\left(\frac{d}{x}\right) \leq C$. Now this last inequality holds if we have $x \leq \frac{c}{\log(d)}$ for a sufficiently small constant $c$, which by rearranging terms amounts to showing that $R \leq c \cdot \frac{\sigma_w\sqrt{k}}{(d \cdot \log(d))^{\frac{3}{2}}}$. Hence up to this point, we have shown that assumption 2 holds with radius $\widetilde{R} := c \cdot \frac{\sigma_w\sqrt{k}}{(d \cdot \log(d))^{\frac{3}{2}}}$, and this implies that it holds for all values of $R$ less than $\widetilde{R}$.

Therefore, we work with the definition of $R$ in equation 37 to show that $R \leq \widetilde{R}$ as follows:

$$R = \frac{3\|f(\theta_0) - y\|_{\ell_2}}{\alpha}$$

$$\overset{(i)}{\leq} \frac{3}{\alpha}\left(\frac{2}{3}\sigma_v\sigma_w\sigma_z\sqrt{k \cdot d \cdot m} + \|\bar{x}\|_{\ell_2}\right)$$

$$= \frac{2\left(2\sigma_v\sigma_w\sigma_z\sqrt{k \cdot m \cdot d} + 3\|\bar{x}\|_{\ell_2}\right)}{\sigma_v\sigma_z\left(\frac{1}{2}\sqrt{k-9} - 2\sqrt{m}\right)\sqrt{\frac{d + \frac{n-1}{\pi}}{2n}}},$$

where in $(i)$ we used Lemma A.10 with $\delta = \frac{1}{3}$. Hence for showing $R \leq \widetilde{R}$ it suffices to show that

$$\frac{2\left(2\sigma_v\sigma_w\sigma_z\sqrt{k \cdot m \cdot d} + 3\|\bar{x}\|_{\ell_2}\right)}{\sigma_v\sigma_z\left(\frac{1}{2}\sqrt{k-9} - 2\sqrt{m}\right)\sqrt{\frac{d + \frac{n-1}{\pi}}{2n}}} \leq c \cdot \frac{\sigma_w\sqrt{k}}{(d \cdot \log(d))^{\frac{3}{2}}},$$

which by assuming $k \geq C \cdot m$ simplifies to

$$\sigma_v \sigma_w \sigma_z \left(d \cdot \log\left(d\right)\right)^{\frac{3}{2}} \sqrt{k \cdot m \cdot d} + \left(d \cdot \log\left(d\right)\right)^{\frac{3}{2}} \|\bar{\boldsymbol{x}}\|_{\ell_2} \leq C \cdot k \cdot \sigma_v \sigma_w \sigma_z.$$

Now this last inequality holds if $k \geq C \cdot md^4 \log\left(d\right)^3$ and by setting $\sigma_v \sigma_w \sigma_z \geq \frac{\|\bar{\boldsymbol{x}}\|_{\ell_2}}{md^{\frac{5}{2}} \log d^{\frac{3}{2}}}$. Therefore Assumption 2 holds for radius $R$ defined in equation 37 with probability at least $1 - ne^{-\frac{m}{2}} - ne^{-c \cdot md^3 \log(d)^2} - k \cdot e^{-c \cdot n} - e^{-\frac{m}{1500}} - e^{-\frac{kd}{162}}$.

Finally to show assumption 3 holds, we note that for all $\boldsymbol{W}$ satisfying $\|\boldsymbol{W} - \boldsymbol{W}_0\| \leq R$, where the value of $R$ is defined in equation 37, it holds that

$$\begin{aligned}
\|\mathcal{J}\left(\boldsymbol{W}\right)\| &\leq \|\mathcal{J}\left(\boldsymbol{W}_0\right)\| + \|\mathcal{J}\left(\boldsymbol{W}\right) - \mathcal{J}\left(\boldsymbol{W}_0\right)\| \\
&\leq \|\mathcal{J}\left(\boldsymbol{W}_0\right)\| + \frac{\alpha}{3} \\
&\leq \|\mathcal{J}\left(\boldsymbol{W}_0\right)\| + \frac{\sigma_{\min}\left(\mathcal{J}\left(\boldsymbol{W}_0\right)\right)}{6} \\
&\leq 2\|\mathcal{J}\left(\boldsymbol{W}_0\right)\| \\
&\leq 3\sigma_v \sigma_z \left(\sqrt{k} + 2\sqrt{m}\right) \sqrt{\frac{d + \frac{n-1}{\pi}}{2n}},
\end{aligned}$$

where the last inequality holds by using lemma A.8, hence establishing that assumption 3 holds with

$$\beta = 3\sigma_v \sigma_z \left(\sqrt{k} + 2\sqrt{m}\right) \sqrt{\frac{d + \frac{n-1}{\pi}}{2n}}$$

with probability at least $1 - e^{-\frac{m}{2}} - 2k \cdot e^{-c \cdot n} - k \cdot e^{-\frac{d}{216}}$, finishing the proof of Theorem 2.2.

# B PROOFS OF THE AUXILIARY LEMMAS

In this section, we first provide a proof of Lemma A.6 and next go over the proofs of the key lemmas stated in Section A.5.

## B.1 PROOF OF LEMMA A.6

Recall that

$$\begin{aligned}
\mathcal{J}(\boldsymbol{W})\mathcal{J}(\boldsymbol{W})^T &= \frac{1}{n^2} \sum_{i,j=1}^{n} \left(\boldsymbol{V} diag(\phi'(\boldsymbol{W}\boldsymbol{z}_i))diag(\phi'(\boldsymbol{W}\boldsymbol{z}_j)\boldsymbol{V}^T\right)\left(\boldsymbol{z}_i^T \boldsymbol{z}_j\right) \\
&= \frac{1}{n^2}\boldsymbol{V} \operatorname*{diag}_{\ell=1,\ldots,k}\left(\left\|\sum_{i=1}^{n} \boldsymbol{z}_i \phi'(\boldsymbol{w}_\ell^T \boldsymbol{z}_i)\right\|_{\ell_2}^2\right)\boldsymbol{V}^T = \frac{1}{n^2}\boldsymbol{V} \cdot \boldsymbol{D}^2 \cdot \boldsymbol{V}^T,
\end{aligned}$$

where $\boldsymbol{D}$ is a diagonal matrix with entries

$$D_{\ell\ell} = \left\|\sum_{i=1}^{n} \boldsymbol{z}_i \phi'(\boldsymbol{w}_\ell^T \boldsymbol{z}_i)\right\|_{\ell_2} = \left\|\boldsymbol{Z}^T \phi'(\boldsymbol{Z}\boldsymbol{w}_\ell)\right\|_{\ell_2}. \tag{39}$$

The matrix $\boldsymbol{Z} \in \mathbb{R}^{n \times d}$ contains the $\boldsymbol{z}_i$'s in its rows. In order to proceed we are going to analyze the entries of the diagonal matrix $\boldsymbol{D}^2$. We observe that

$$\left\|\boldsymbol{Z}^T \phi'(\boldsymbol{Z}\boldsymbol{w})\right\|_{\ell_2}^2 = \underbrace{\left\|(\boldsymbol{I} - \frac{\boldsymbol{w}\boldsymbol{w}^T}{\|\boldsymbol{w}\|^2})\boldsymbol{Z}^T \phi'(\boldsymbol{Z}\boldsymbol{w})\right\|_{\ell_2}^2}_{A} + \underbrace{\left\|\frac{\boldsymbol{w}\boldsymbol{w}^T}{\|\boldsymbol{w}\|^2}\boldsymbol{Z}^T \phi'(\boldsymbol{Z}\boldsymbol{w})\right\|_{\ell_2}^2}_{B}.$$

First, we compute the expectation of $A$. We observe that

$$A = \left\|\sum_{i=1}^{n}(\boldsymbol{I} - \frac{\boldsymbol{w}\boldsymbol{w}^T}{\|\boldsymbol{w}\|^2})\boldsymbol{z}_i \phi'(\boldsymbol{w}^T \boldsymbol{z}_i)\right\|_{\ell_2}^2.$$

Conditioned on $\boldsymbol{w}$, $\left(\boldsymbol{I} - \frac{\boldsymbol{w}\boldsymbol{w}^T}{\|\boldsymbol{w}\|^2}\right)\boldsymbol{z}_i$ is distributed as $\mathcal{N}\left(0, \sigma_z^2\left(\boldsymbol{I} - \frac{\boldsymbol{w}\boldsymbol{w}^T}{\|\boldsymbol{w}\|^2}\right)\right)$ and $\boldsymbol{w}^T\boldsymbol{z}_i$ has distribution $\mathcal{N}\left(0, \sigma_z^2\|\boldsymbol{w}\|^2\right)$. Moreover, these two random variables are independent, because $\boldsymbol{w}$ is in the null space of $\boldsymbol{I} - \frac{\boldsymbol{w}\boldsymbol{w}^T}{\|\boldsymbol{w}\|^2}$. This observation yields

$$
\begin{aligned}
\mathbb{E}(A) &= \mathbb{E}\left\|\sum_{i=1}^n (\boldsymbol{I} - \frac{\boldsymbol{w}\boldsymbol{w}^T}{\|\boldsymbol{w}\|^2})\boldsymbol{z}_i\phi'(\boldsymbol{w}^T\boldsymbol{z}_i)\right\|_{\ell_2}^2 \\
&= \sum_{i=1}^n\sum_{j=1}^n \mathbb{E}\left(\left\langle (\boldsymbol{I} - \frac{\boldsymbol{w}\boldsymbol{w}^T}{\|\boldsymbol{w}\|^2})\boldsymbol{z}_i, (\boldsymbol{I} - \frac{\boldsymbol{w}\boldsymbol{w}^T}{\|\boldsymbol{w}\|^2})\boldsymbol{z}_j\right\rangle\phi'(\boldsymbol{w}^T\boldsymbol{z}_i)\phi'(\boldsymbol{w}^T\boldsymbol{z}_j)\right) \\
&= \sum_{i=1}^n \mathbb{E}\left(\left\|(\boldsymbol{I} - \frac{\boldsymbol{w}\boldsymbol{w}^T}{\|\boldsymbol{w}\|^2})\boldsymbol{z}_i\right\|_{\ell_2}^2\right)\mathbb{E}\left(\left[\phi'(\boldsymbol{w}^T\boldsymbol{z}_i)\right]^2\right) \\
&= \sum_{i=1}^n \frac{1}{2}(d-1)\sigma_z^2 = \frac{n}{2}(d-1)\sigma_z^2.
\end{aligned}
$$

Next we show that $A$ is concentrated around its mean. Because $\left(\boldsymbol{I} - \frac{\boldsymbol{w}\boldsymbol{w}^T}{\|\boldsymbol{w}\|^2}\right)\boldsymbol{z}_i$ is independent from $\boldsymbol{w}^T\boldsymbol{z}_i$, we use $\boldsymbol{z}_i'$ as an independent copy of $\boldsymbol{z}_i$. Hence we can write

$$
\begin{aligned}
A &= \left\|(\boldsymbol{I} - \frac{\boldsymbol{w}\boldsymbol{w}^T}{\|\boldsymbol{w}\|^2})\boldsymbol{Z}^T\phi'(\boldsymbol{Z}\boldsymbol{w})\right\|_{\ell_2}^2 \\
&= \left\|(\boldsymbol{I} - \frac{\boldsymbol{w}\boldsymbol{w}^T}{\|\boldsymbol{w}\|^2})\boldsymbol{Z}^T\phi'(\boldsymbol{Z}'\boldsymbol{w})\right\|_{\ell_2}^2 \\
&= \left\|\sum_{i=1}^n (\boldsymbol{I} - \frac{\boldsymbol{w}\boldsymbol{w}^T}{\|\boldsymbol{w}\|^2})\boldsymbol{z}_i\phi'\left(\boldsymbol{w}^T\boldsymbol{z}_i'\right)\right\|_{\ell_2}^2 = \left\|\sum_{i=1}^n \boldsymbol{g}_i u_i\right\|_{\ell_2}^2,
\end{aligned}
$$

where $\boldsymbol{g}_i = \left(\boldsymbol{I} - \frac{\boldsymbol{w}\boldsymbol{w}^T}{\|\boldsymbol{w}\|^2}\right)\boldsymbol{z}_i \sim \mathcal{N}\left(0, \sigma_z^2\left(\boldsymbol{I} - \frac{\boldsymbol{w}\boldsymbol{w}^T}{\|\boldsymbol{w}\|^2}\right)\right)$ and $u_i = \phi'\left(\boldsymbol{w}^T\boldsymbol{z}_i'\right) \sim bern(\frac{1}{2})$,[4] and these are all independent from each other. Note that $\|\sum_{i=1}^n \boldsymbol{g}_i u_i\|_{\ell_2}^2$ has the same distribution as $\|\boldsymbol{g}\|_{\ell_2}^2 \cdot \|\boldsymbol{u}\|_{\ell_2}^2$, where $\boldsymbol{g} \sim \mathcal{N}\left(0, \sigma_z^2\left(\boldsymbol{I} - \frac{\boldsymbol{w}\boldsymbol{w}^T}{\|\boldsymbol{w}\|^2}\right)\right)$ and $\boldsymbol{u}$ is a vector with entries $u_i$. Note that for the norm of $\boldsymbol{u}$, the event

$$
\frac{n}{2}(1-\delta) \leq \|\boldsymbol{u}\|_{\ell_2}^2 \leq \frac{n}{2}(1+\delta)
$$

holds with probability at least $1 - 2e^{-\frac{n\delta^2}{2}}$. Recall that for $\boldsymbol{g} \sim \mathcal{N}(0, \sigma^2\boldsymbol{I}_d)$ and $0 < \delta \leq \frac{1}{2}$ we have

$$
\begin{aligned}
\mathbb{P}\left(\|\boldsymbol{g}\|_{\ell_2}^2 \geq (1+\delta)\mathbb{E}\left(\|\boldsymbol{g}\|_{\ell_2}^2\right)\right) &\leq e^{-\frac{d\delta^2}{6}}, \\
\mathbb{P}\left(\|\boldsymbol{g}\|_{\ell_2}^2 \leq (1-\delta)\mathbb{E}\left(\|\boldsymbol{g}\|_{\ell_2}^2\right)\right) &\leq e^{-\frac{d\delta^2}{4}}.
\end{aligned}
\tag{40}
$$

By applying the union bound and noting that $\mathbb{E}\left(\|\boldsymbol{g}\|_{\ell_2}^2\right) = (d-1)\sigma_z^2$, for $0 < \delta \leq \frac{3}{2}$, we obtain that the event

$$
\left|\|\boldsymbol{g}\|_{\ell_2}^2\|\boldsymbol{u}\|_{\ell_2}^2 - \frac{n}{2}(d-1)\sigma_z^2\right| \leq \delta\frac{n}{2}(d-1)\sigma_z^2
$$

---

[4]Here, $bern(\frac{1}{2})$ means that the random variable takes values 0 and 1 each with probability $1/2$.

holds with probability at least $1 - 2e^{-\frac{n\delta^2}{18}} - 2e^{-\frac{d\delta^2}{54}}$.

In order to analyze $B$, we first note that

$$
\begin{aligned}
B = \left\| \frac{\boldsymbol{w}\boldsymbol{w}^T}{||\boldsymbol{w}||^2} \boldsymbol{Z}^T \phi'(\boldsymbol{Z}\boldsymbol{w}) \right\|_{\ell_2}^2 &= \left| \frac{\boldsymbol{w}^T}{||\boldsymbol{w}||} \boldsymbol{Z}^T \phi'(\boldsymbol{Z}\boldsymbol{w}) \right|^2 \\
&= \left| \left\langle \boldsymbol{Z} \frac{\boldsymbol{w}}{||\boldsymbol{w}||}, \phi'(\boldsymbol{Z}\boldsymbol{w}) \right\rangle \right|^2 \\
&= \left| \langle \boldsymbol{g}, \phi'(||\boldsymbol{w}||\boldsymbol{g}) \rangle \right|^2 \\
&= \left| \sum_{i=1}^n \boldsymbol{g}_i \cdot \mathbb{1}_{(\boldsymbol{g}_i \geq 0)} \right|^2 = \left( \sum_{i=1}^n ReLU(\boldsymbol{g}_i) \right)^2,
\end{aligned}
$$

where $\boldsymbol{g}_i = \boldsymbol{z}_i^T \frac{\boldsymbol{w}}{||\boldsymbol{w}||} \sim \mathcal{N}\left(0, \sigma_z^2\right)$. It follows that

$$
\begin{aligned}
\mathbb{E}(B) = \mathbb{E}\left( \sum_{i=1}^n ReLU(\boldsymbol{g}_i) \right)^2 \\
= \sum_{i=1}^n \mathbb{E}\left( ReLU^2(\boldsymbol{g}_i) \right) + \sum_{i \neq j} \mathbb{E}\left( ReLU(\boldsymbol{g}_i) ReLU(\boldsymbol{g}_j) \right) \\
= \sigma_z^2 \left( \frac{n}{2} + \frac{n(n-1)}{2\pi} \right),
\end{aligned}
$$

which results in

$$
\mathbb{E}\left( \boldsymbol{D}_{\ell\ell}^2 \right) = \mathbb{E}(A) + \mathbb{E}(B) = \sigma_z^2 \left( \frac{nd}{2} + \frac{n(n-1)}{2\pi} \right), \quad 1 \leq \ell \leq k.
$$

Next, in order to show that $B$ concentrates around its mean, we note that $ReLU(\boldsymbol{g}_i)$ is a sub-Gaussian random variable with $\psi_2-$norm $C\sigma_z$, where $C$ is a fixed constant. Therefore $X = \sum_{i=1}^n ReLU(\boldsymbol{g}_i)$ is sub-Gaussian with $\psi_2-$norm $C\sqrt{n}\sigma_z$. By the sub-exponential tail bound for $X^2 - \mathbb{E}(X^2)$ we obtain

$$
\mathbb{P}\left( |B - \mathbb{E}(B)| \geq t \right) \leq 2e^{-c\frac{t}{n\sigma_z^2}}.
$$

Finally by putting these results together and using union bounds we have

$$
\mathbb{P}\left\{ \left| \boldsymbol{D}_{\ell\ell}^2 - \mathbb{E}\left( \boldsymbol{D}_{\ell\ell}^2 \right) \right| \geq \delta \mathbb{E}\left( \boldsymbol{D}_{\ell\ell}^2 \right) \right\} \leq 2e^{-\frac{n\delta^2}{18}} + 2e^{-\frac{d\delta^2}{54}} + 2e^{-c_1 n\delta}, \quad 0 \leq \delta \leq \frac{3}{2},
$$

finishing the proof of Lemma A.6.

## B.2 PROOF OF LEMMA A.7

Our main tool for bounding the minimum singular value of the Jacobian mapping will be the following lemma from Soltanolkotabi (2019):

**Lemma B.1** *Let $\boldsymbol{d} \in \mathbb{R}^k$ be a fixed vector with nonzero entries and $\boldsymbol{D} = diag(\boldsymbol{d})$. Also, let $\boldsymbol{A} \in \mathbb{R}^{k \times m}$ have i.i.d. $\mathcal{N}(0,1)$ entries and $\mathcal{T} \subseteq \mathbb{R}^m$. Define*

$$
b_k(\boldsymbol{d}) = \mathbb{E}\left[ \|\boldsymbol{D}\boldsymbol{g}\|_{\ell_2} \right],
$$

*where $\boldsymbol{g} \sim \mathcal{N}(\boldsymbol{0}, \boldsymbol{I}_k)$. Also define*

$$
\sigma(\mathcal{T}) := \max_{\boldsymbol{v} \in \mathcal{T}} \|\boldsymbol{v}\|_{\ell_2}.
$$

*Then for all $\boldsymbol{u} \in \mathcal{T}$ we have*

$$
\left| \|\boldsymbol{D}\boldsymbol{A}\boldsymbol{u}\|_{\ell_2} - b_k(\boldsymbol{d}) \|\boldsymbol{u}\|_{\ell_2} \right| \leq \|\boldsymbol{d}\|_{\ell_\infty} \omega(\mathcal{T}) + \eta
$$

*with probability at least $1 - 6e^{\frac{-\eta^2}{8\|\boldsymbol{d}\|_{\ell_\infty}^2 \sigma^2(\mathcal{T})}}$.*

In order to apply this lemma, we set the elements of $\boldsymbol{d}$ to be $D_{\ell\ell}$ as in equation 39 and choose $\mathcal{T} = S^{m-1}$ and $\boldsymbol{A} = \boldsymbol{V}^T \in \mathbb{R}^{k \times m}$ with $\mathcal{N}(0, \sigma_v^2)$ entries. It follows that

$$b_k(\boldsymbol{d}) = \mathbb{E} \|\boldsymbol{Dg}\|_{\ell_2} = \sqrt{\mathbb{E}\left(\|\boldsymbol{Dg}\|_{\ell_2}^2\right) - \text{Var}\left(\|\boldsymbol{Dg}\|_{\ell_2}\right)},$$

where

$$\mathbb{E}\left(\|\boldsymbol{Dg}\|_{\ell_2}^2\right) = \|\boldsymbol{d}\|_{\ell_2}^2 = \sum_{\ell=1}^{k} \boldsymbol{D}_{\ell\ell}^2.$$

We are going to use the fact that for a $B$-Lipschitz function $\phi$ and normal random variable $g \sim \mathcal{N}(0, 1)$, based on the Poincare inequality (Ledoux, 2001) we have $\text{Var}(\phi(g)) \leq B^2$. By noting that for a diagonal matrix $\boldsymbol{D}$

$$\left|\|\boldsymbol{D}x\|_{\ell_2} - \|\boldsymbol{D}y\|_{\ell_2}\right| \leq \|\boldsymbol{D}\boldsymbol{x} - \boldsymbol{D}\boldsymbol{y}\|_{\ell_2} \leq \|\boldsymbol{d}\|_{\ell_\infty} \|\boldsymbol{x} - \boldsymbol{y}\|_{\ell_2},$$

we get

$$\mathbb{E} \|\boldsymbol{Dg}\|_{\ell_2} = \sqrt{\mathbb{E}\left(\|\boldsymbol{Dg}\|_{\ell_2}^2\right) - \text{Var}(\|\boldsymbol{Dg}\|_{\ell_2})}$$

$$\geq \sqrt{\|\boldsymbol{d}\|_{\ell_2}^2 - \|\boldsymbol{d}\|_{\ell_\infty}^2}.$$

This combined with $\omega\left(S^{m-1}\right) \leq \sqrt{m}$ and Lemma B.1 yields that the event

$$\sigma_{min}(\boldsymbol{VD}) \geq \sigma_v \left(\sqrt{\|\boldsymbol{d}\|_{\ell_2}^2 - \|\boldsymbol{d}\|_{\ell_\infty}^2} - \|\boldsymbol{d}\|_{\ell_\infty} \sqrt{m} - \eta\right) \tag{41}$$

holds with probability at least $1 - 3e^{\frac{-\eta^2}{8\|\boldsymbol{d}\|_{\ell_\infty}^2}}$.

Next, using the concentration bound for $\boldsymbol{D}_{\ell\ell}^2$, which we obtained in Section B.1, we bound $\|\boldsymbol{d}\|_{\ell_2}^2$ and $\|\boldsymbol{d}\|_{\ell_\infty}$, where we have set $\boldsymbol{d}_i = \boldsymbol{D}_{ii}$ for $1 \leq i \leq k$. For $0 \leq \delta \leq \frac{3}{2}$ we compute that

$$\mathbb{P}\left(\max_{1 \leq i \leq k} \boldsymbol{d}_i \geq (1 + \delta)\sqrt{\mathbb{E}[\boldsymbol{d}_i^2]}\right) = \mathbb{P}\left(\bigcup_{i=1}^{k} \boldsymbol{d}_i^2 \geq (1 + \delta)^2 \mathbb{E}[\boldsymbol{d}_i^2]\right)$$

$$\leq k \cdot \mathbb{P}\left(\boldsymbol{d}_i^2 \geq (1 + \delta)^2 \mathbb{E}[\boldsymbol{d}_i^2]\right)$$

$$\leq k \cdot \mathbb{P}\left(\boldsymbol{d}_i^2 \geq (1 + \delta) \mathbb{E}[\boldsymbol{d}_i^2]\right) \leq k \cdot \left(e^{-\frac{n\delta^2}{18}} + e^{-\frac{d\delta^2}{54}} + e^{-c_1 n\delta}\right), \tag{42}$$

as well as

$$\mathbb{P}\left(\|\boldsymbol{d}\|_{\ell_2} \leq (1 - \delta)\sqrt{k}\sqrt{\mathbb{E}[\boldsymbol{d}_i^2]}\right) \leq \mathbb{P}\left(\bigcup_{i=1}^{k} \boldsymbol{d}_i^2 \leq (1 - \delta)^2 \mathbb{E}[\boldsymbol{d}_i^2]\right)$$

$$\leq k \cdot \mathbb{P}\left(\boldsymbol{d}_i^2 \leq (1 - \delta)^2 \mathbb{E}[\boldsymbol{d}_i^2]\right)$$

$$\leq k \cdot \mathbb{P}\left(\boldsymbol{d}_i^2 \leq (1 - \delta) \mathbb{E}[\boldsymbol{d}_i^2]\right) \leq k \cdot \left(e^{-\frac{n\delta^2}{18}} + e^{-\frac{d\delta^2}{54}} + e^{-c_1 n\delta}\right). \tag{43}$$

Finally by replacing $\eta$ with $\eta \|\boldsymbol{d}\|_{\ell_\infty} \sqrt{m}$ in equation 41, combined with equation 42 and equation 43, for a random $\boldsymbol{W}_0$ with i.i.d. $\mathcal{N}(0, \sigma_w^2)$ entries we have:

$$\sigma_{min}(\mathcal{J}(\boldsymbol{W}_0)) = \frac{1}{n}\sigma_{min}(\boldsymbol{VD})$$

$$\geq \frac{\sigma_v}{n}\left(\sqrt{(1 - \delta)^2 k - (1 + \delta)^2} - \sqrt{m}(1 + \eta)(1 + \delta)\right)\sqrt{\mathbb{E}[\boldsymbol{d}_i^2]}$$

$$= \left(\sqrt{(1 - \delta)^2 k - (1 + \delta)^2} - \sqrt{m}(1 + \eta)(1 + \delta)\right)\sigma_v \sigma_z \sqrt{\frac{d + \frac{n-1}{\pi}}{2n}}, \quad 0 \leq \delta \leq \frac{3}{2},$$

with probability at least $1 - 3e^{-\frac{\eta^2 m}{8}} - 2k \cdot \left(e^{-\frac{n\delta^2}{18}} + e^{-\frac{d\delta^2}{54}} + e^{-c_1 n\delta}\right)$. This completes the proof of Lemma A.7.

### B.3 PROOF OF LEMMA A.8

Recall that

$$\mathcal{J}(\boldsymbol{W})\mathcal{J}(\boldsymbol{W})^T = \frac{1}{n^2}\boldsymbol{V}\operatorname*{diag}_{\ell=1,\dots,k}\left(\left\|\sum_{i=1}^n \boldsymbol{z}_i\phi'(\boldsymbol{w}_\ell^T\boldsymbol{z}_i)\right\|_{\ell_2}^2\right)\boldsymbol{V}^T = \frac{1}{n^2}\boldsymbol{V}\cdot\boldsymbol{D}^2\cdot\boldsymbol{V}^T,$$

which implies that

$$\|\mathcal{J}(\boldsymbol{W}_0)\| = \frac{1}{n}\|\boldsymbol{V}\cdot\boldsymbol{D}\| \le \frac{1}{n}\|\boldsymbol{V}\|\,\|\boldsymbol{D}\|.$$

For matrix $\boldsymbol{V}\in\mathbb{R}^{m\times k}$ with i.i.d $\mathcal{N}(0,\sigma_v^2)$ the event

$$\|\boldsymbol{V}\| \le \sigma_v\left(\sqrt{k}+2\sqrt{m}\right)$$

holds with probability at least $1-e^{-\frac{m}{2}}$. Regarding matrix $\boldsymbol{D}$ by repeating equation 42 the following event

$$\|\boldsymbol{D}\| = \max_{1\le i\le k} D_{ii} \le (1+\delta)\sqrt{\mathbb{E}[D_{ii}^2]} = (1+\delta)\,\sigma_z\sqrt{\frac{nd}{2}+\frac{n(n-1)}{2\pi}}, \quad 0\le\delta\le\frac{3}{2}$$

holds with probability at least $1 - k\cdot\left(e^{-\frac{n\delta^2}{18}}+e^{-\frac{d\delta^2}{54}}+e^{-c_1 n\delta}\right)$. Putting these together it yields that the event

$$\|\mathcal{J}(\boldsymbol{W}_0)\| \le (1+\delta)\,\sigma_v\sigma_z\left(\sqrt{k}+2\sqrt{m}\right)\sqrt{\frac{d+\frac{n-1}{\pi}}{2n}}, \qquad 0\le\delta\le\frac{3}{2}$$

holds with probability at least $1 - e^{-\frac{m}{2}} - k\cdot\left(e^{-\frac{n\delta^2}{18}}+e^{-\frac{d\delta^2}{54}}+e^{-c_1 n\delta}\right)$, finishing the proof of Lemma A.8.

### B.4 PROOF OF LEMMA A.10

First, note that if $\boldsymbol{W}$ has i.i.d. $\mathcal{N}(0,\sigma_w^2)$ entries and $\boldsymbol{V},\boldsymbol{W},\boldsymbol{Z}$ are all independent, then $\|f(\boldsymbol{W})\|_{\ell_2} = \frac{1}{n}\left\|\boldsymbol{V}\phi(\boldsymbol{W}\boldsymbol{Z}^T)\mathbf{1}_{n\times 1}\right\|_{\ell_2}$ has the same distribution as $\|\boldsymbol{v}\|_{\ell_2}\|\boldsymbol{a}\|_{\ell_2}$, where $\boldsymbol{v}\sim\mathcal{N}(0,\sigma_v^2\boldsymbol{I}_m)$ and $\boldsymbol{a} = \frac{1}{n}\phi(\boldsymbol{W}\boldsymbol{Z}^T)\mathbf{1}$ has independent sub-Gaussian entries, so its $\ell_2$-norm is concentrated. Note that conditioned on $\boldsymbol{W}$, $a_i = \frac{1}{n}\sum_{j=1}^n ReLU(\boldsymbol{z}_j^T\boldsymbol{w}_i)$ is sub-Gaussian with $\|a_i\|_{\psi_2} = C\frac{\|\boldsymbol{w}_i\|_{\ell_2}\sigma_z}{\sqrt{n}}$, and it is concentrated around $\mathbb{E}a_i = \frac{1}{\sqrt{2\pi}}\|\boldsymbol{w}_i\|_{\ell_2}\sigma_z$. This gives

$$\mathbb{P}\{a_i\le(1+\delta)\mathbb{E}a_i\} \ge 1 - e^{-c\frac{\delta^2(\mathbb{E}a_i)^2}{\|a_i\|_{\psi_2}^2}} = 1 - e^{-cn\delta^2},$$

which implies that

$$\mathbb{P}\{a_i^2\le(1+3\delta)(\mathbb{E}a_i)^2\} \ge \mathbb{P}\{a_i^2\le(1+\delta)^2(\mathbb{E}a_i)^2\} \ge 1 - e^{-cn\delta^2}, \quad 0\le\delta\le 1.$$

Due to the union bound we get that

$$\mathbb{P}\left\{\|\boldsymbol{a}\|_{\ell_2}^2\ge(1+\delta)\sum_{i=1}^k(\mathbb{E}a_i)^2\right\} \le \mathbb{P}\left\{\bigcup_{i=1}^k a_i^2\ge(1+\delta)(\mathbb{E}a_i)^2\right\}$$

$$\le \sum_{i=1}^k\mathbb{P}\{a_i^2\ge(1+\delta)(\mathbb{E}a_i)^2\} \le k\cdot e^{-cn(\delta/3)^2}, \quad 0\le\delta\le 3.$$

By substituting $\sum_{i=1}^k(\mathbb{E}a_i)^2 = \frac{1}{2\pi}\sigma_z^2\|\boldsymbol{W}\|_F^2$ this shows

$$\mathbb{P}\left\{\|\boldsymbol{a}\|_{\ell_2}\le(1+\delta)\frac{1}{\sqrt{2\pi}}\sigma_z\|\boldsymbol{W}\|_F\right\} \ge \mathbb{P}\left\{\|\boldsymbol{a}\|_{\ell_2}^2\le(1+\delta)\frac{1}{2\pi}\sigma_z^2\|\boldsymbol{W}\|_F^2\right\} \ge 1 - k\cdot e^{-cn(\delta/3)^2}, \quad 0\le\delta\le 3.$$

We also have the following result for $\boldsymbol{v} \sim \mathcal{N}(0, \sigma_v^2 \boldsymbol{I}_m)$

$$\mathbb{P}\left\{\|\boldsymbol{v}\|_{\ell_2} \leq (1+\delta)\,\sigma_v\sqrt{m}\right\} \geq 1 - e^{-\frac{\delta^2 m}{2}}.$$

By combining the above results we obtain

$$\mathbb{P}\left\{\|\boldsymbol{a}\|_{\ell_2}\|\boldsymbol{v}\|_{\ell_2} \leq (1+\delta)\,\frac{1}{\sqrt{2\pi}}\sigma_v\sigma_z\sqrt{m}\,\|\boldsymbol{W}\|_F\right\} \geq \mathbb{P}\left\{\|\boldsymbol{a}\|_{\ell_2}\|\boldsymbol{v}\|_{\ell_2} \leq (1+\delta/3)^2\,\frac{1}{\sqrt{2\pi}}\sigma_v\sigma_z\sqrt{m}\,\|\boldsymbol{W}\|_F\right\}$$

$$\geq 1 - k \cdot e^{-cn(\delta/9)^2} - e^{-\frac{(\delta/3)^2 m}{2}}, \quad 0 \leq \delta \leq 3.$$

Furthermore, we can bound $\|\boldsymbol{W}\|_F$ by the tail inequality

$$\mathbb{P}\left\{\|\boldsymbol{W}\|_F \leq (1+\delta)\,\sigma_w\sqrt{kd}\right\} \geq 1 - e^{-\frac{\delta^2 kd}{2}}.$$

Hence, by combining the last two results we have that

$$\mathbb{P}\left\{\|\boldsymbol{a}\|_{\ell_2}\|\boldsymbol{v}\|_{\ell_2} \leq (1+\delta)\,\frac{1}{\sqrt{2\pi}}\sigma_v\sigma_z\sigma_w\sqrt{k\cdot d\cdot m}\right\} \geq \mathbb{P}\left\{\|\boldsymbol{a}\|_{\ell_2}\|\boldsymbol{v}\|_{\ell_2} \leq (1+\delta/3)^2\,\frac{1}{\sqrt{2\pi}}\sigma_v\sigma_z\sigma_w\sqrt{k\cdot d\cdot m}\right\}$$

$$\geq 1 - k \cdot e^{-cn(\delta/27)^2} - e^{-\frac{(\delta/9)^2 m}{2}} - e^{-\frac{(\delta/3)^2 kd}{2}}, \quad 0 \leq \delta \leq 3.$$

Therefore, due to the triangle inequality the event

$$\|f(\boldsymbol{W}_0) - \bar{\boldsymbol{x}}\|_{\ell_2} \leq (1+\delta)\frac{1}{\sqrt{2\pi}}\sigma_v\sigma_w\sigma_z\sqrt{k\cdot d\cdot m} + \|\bar{\boldsymbol{x}}\|_{\ell_2}, \quad 0 \leq \delta \leq 3$$

holds with probability at least $1 - k \cdot e^{-c_2 n(\delta/27)^2} - e^{-\frac{(\delta/9)^2 m}{2}} - e^{-\frac{(\delta/3)^2 kd}{2}}$ for some positive constant $c_2$, completing the proof of Lemma A.10.

## C    ADDITIONAL EXPERIMENTS

**Effect of single component overparameterization:** In Section 3 of the main paper, we performed experiments in the setting where the size of generator and discriminator are held roughly the same (both discriminator and generator uses the same value of $k$). In this part, we analyze single-component overparameterization where we study the effect of overparameterization when one of the components (generator / discriminator) has varying $k$, while the other component uses the standard value of $k$ (64 for DCGAN and 128 for Resnet GAN). The FID variation of single-component overparameterization are shown in Fig. 7. We observe similar trends as the previous case where increasing overparameterization leads to improved FID scores. Interestingly, increasing the value of $k$ beyond the default value used in the other component leads to a slight drop in performance. Hence, choosing comparable sizes of discriminator and generator models is recommended.

## D    EXPERIMENTAL DETAILS

The model architectures we use this in the experiments are shown in Figure 8. In both DCGAN and Resnet-based GANs, the papemeter $k$ controls the number of convolutional filters in each layer. The larger the value of $k$ is, the more overparameterized the models are.

**Optimization:**    Both DCGAN and Resnet-based GAN models are optimized using the commonly used hyper-parameters: Adam with learning rate $0.0001$ and betas $(0.5, 0.999)$ for DCGAN, gradient penalty of 10 and 5 critic iterations per generator's iteration for both DCGAN and Resnet-based GAN models. Models are trained for $300,000$ iterations with a batch size of 64.

## E    NEAREST NEIGHBOR VISUALIZATION

In this section, we visualize the nearest neighbors of samples generated using GAN models trained with different levels of overparameterization. More specifically, we trained a DCGAN model with $k = 8$ and $k = 128$, synthesize random samples from the trained model and query the nearest neighbors in the training set. The plot of obtained samples is shown in Figure. 10. We observe that overparameterized models generate samples with high diversity.

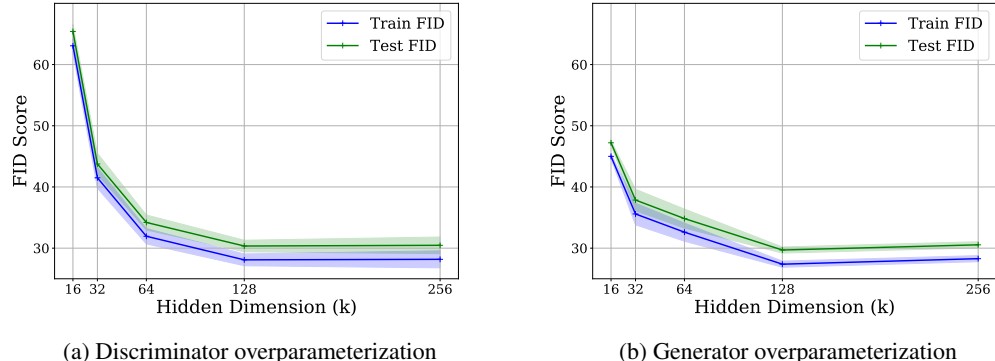

(a) Discriminator overparameterization

(b) Generator overparameterization

Figure 7: **Single Component Overparamterization Results:** We plot FID scores of Resnet GAN as the hidden dimension of one of the components are varied, while the hidden dimension of other component is held fixed. Even in this case, overparameterization improves model convergence.

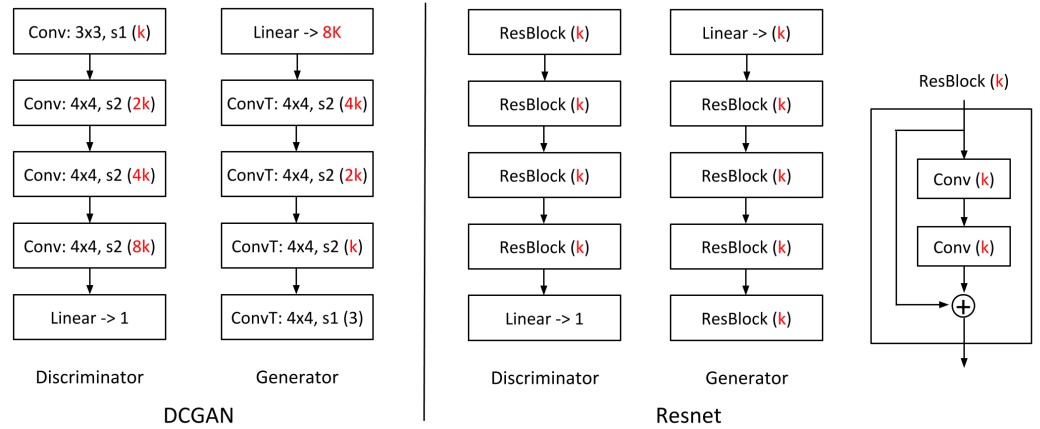

Figure 8: **Architectures used in over-parameterization experiments.** The number of out-channels in convolutional layers is indicated in red. Parameter $k$ controls the width of the architectures – larger the $k$, more over-parameterized the models are.

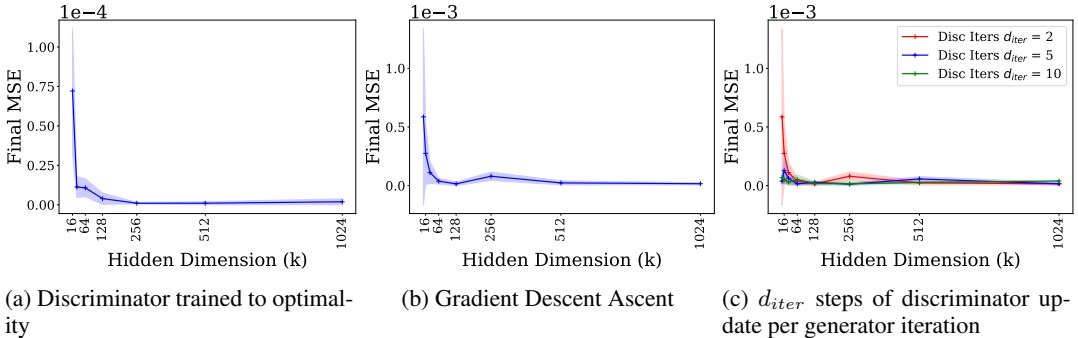

(a) Discriminator trained to optimality

(b) Gradient Descent Ascent

(c) $d_{iter}$ steps of discriminator update per generator iteration

Figure 9: **Convergence plot** GAN model trained on the Two-Moons dataset, with linear discriminator and 1-hidden layer generator as the hidden dimension ($k$) increases. Over-parameterized models show improved convergence

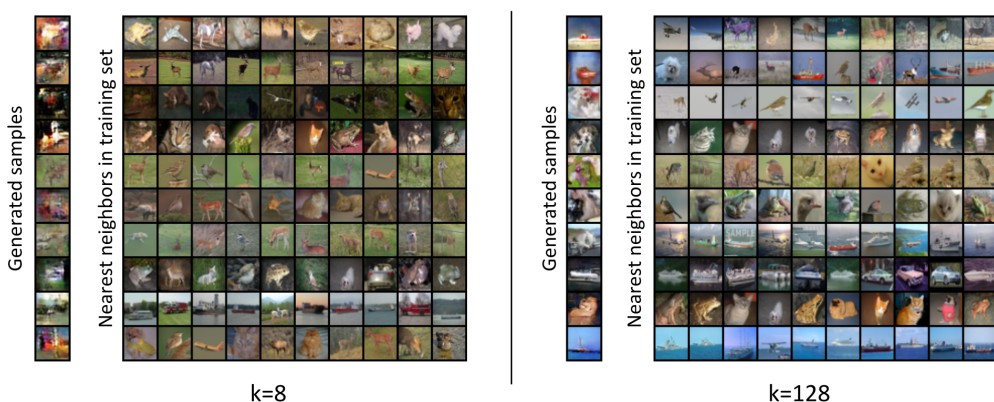

Figure 10: **Nearest neighbor visualization.** We visualize the nearest neighbor samples in training set for generations from DCGAN model trained on CIFAR-10 dataset. Left panel shows DCGAN trained with $k = 8$, while the right one shows the one with $k = 128$. We observe that overparameterized models generate samples with high diversity.

