# OpenReview forum: "Understanding Over-parameterization in Generative Adversarial Networks"
_ICLR.cc/2021/Conference — ICLR 2021 Poster_

### Official Review · AnonReviewer3 · 2020-10-13
**More details should be provided**

**Rating:** 4
**Confidence:** 5

**Review:**

In this paper, the authors proposed to analyze the over-parameterization in GANs optimized by the alternative gradient descent method. Specifically, considering a GAN with an over-parameterized G and a linear D, the authors proposed the theorem 2.1 to provide a theoretical convergence rate in GAN’s training.

However, this paper is problematic. The details are as follows:

1. The gap between the paper’s title and theoretical claims. To my best knowledge, the OVER-PARAMETERIZATION means that the model will overfit to the training data and cannot generalize to test data. Thus, the generalization bound between training error and the test error is the main concern in this topic. However, in this paper, the authors mainly focus on the convergence rate during the training. I think this topic is more correlated to the non-convex optimization problem, rather than the over-parameterization problem.

2. The theoretical claims in Sec 2 are not convincing. First, what is the data distribution? In the whole Sec 2, there is no detailed explanation about the data distribution, which is one of the most important parts of the analysis. Only the `Numerical Validation` part mentioned that the data distribution is a univariate Gaussian. Though we assume that the data is Gaussian, simply minimizing the distance between the mean of data and the ones of G is not enough: the variance in the data is not taken into consideration. I’m not sure that using a linear discriminator is enough for G to model the data distribution; generally, we assume the discriminator has infinity capacity.

3. The theoretical derivation is ambitious. In Theorem 2.1, why $V$ is not optimized in Eqn. (2) & (3)? In Theorem 2.3, why $f$ is a general mapping? In GANs, it should be a parametric mapping from z to x. Besides, Theorem 2.3 claims that it is a general minimax problem. However, it is still restricted to a linear discriminator. Clarity needs to be improved.

4. Finally, the experimental results cannot fully validate the authors’ claims. In Fig. 4, with smaller k, G’s capacity is not sufficient to capture the data distribution. In this case, comparing the convergence rate is unfair. Further, the paper does not provide any bound on the gap between training error and test error. I don’t know the purpose of Fig. 3 and Fig. 5.

Three related work:
[*1] talks about the existence of the equilibrium of GANs’ minimax problem. If the equilibrium does not exist, then the convergence rate cannot validate any claims as I mentioned above.
[*2] also uses control theory to understand GAN’s training and [*3] adopts control theory to improve the training dynamics of deep models. The relationship should be discussed.


[*1] Farnia, Farzan, and Asuman Ozdaglar. "GANs May Have No Nash Equilibria." arXiv preprint arXiv:2002.09124 (2020).

[*2] Xu, Kun, et al. "Understanding and Stabilizing GANs’ Training Dynamics using Control Theory."

[*3] An, Wangpeng, et al. "A PID controller approach for stochastic optimization of deep networks." Proceedings of the IEEE Conference on Computer Vision and Pattern Recognition. 2018.

---

> ### Author Response · Authors · 2020-11-20
> **Response to reviewer 3**
>
> $\textbf{Overparameterization}$:
>
> You mentioned that “the OVER-PARAMETERIZATION means that the model will overfit to the training data and cannot generalize to test data. Thus, the generalization bound between training error and the test error is the main concern in this topic.”. Two key recent observations in supervised learning are that (1) model overparameterization does NOT lead to overfitting while (2) it helps the convergence of optimization solvers “during the training.” (e.g. see references Soltanolkotabi et al., 2018; Allen-Zhu et al., 2019; Du et al., 2019; Oymak & Soltanolkotabi, 2019; Zou & Gu, 2019; Oymak et al., 2019 in the paper). In this paper, we extend these two key observations to the unsupervised learning problem of GANs: (1) We provide analysis of the convergence of GDA for non-convex concave min-max optimization during the training, and (2) we empirically show that overparameterized GANs do not overfit (memorize) to the training set. We are also optimistic that we can develop rigorous theoretical guarantees to further justify our empirical observations in (2) by utilizing related guarantees on generalization in supervised overparameterized learning e.g. see [4] and  [5]
> [4] Arora et al., “Fine-grained analysis of optimization and generalization for overparameterized two-layer neural networks”
> [5] Oymak et al., “Generalization Guarantees for Neural Networks via Harnessing the Low-rank Structure of the Jacobian”
>
> $\textbf{“The theoretical claims in Sec 2 are not convincing. what is the data distribution?”}$:
>
> Theorems 2.1 and 2.2 hold for “any” data points x_1, x_2,.... In numerical simulations of Figure 2 validating these results, we use Gaussian distributions to generate samples.  In addition to using a Gaussian dataset, we have added new numerical simulations using the two-moons dataset. The results are included in Figure 9 of the revised draft.
>
> $\textbf{“Generally, we assume the discriminator has infinity capacity.”}$:
>
> We respectfully disagree. For example, in Wasserstein GAN, if the discriminator family is the set of all Lipschitz functions, [6,7] show that WGAN would require exponentially-many samples to reliably learn from the training set. Restricting GAN’s discriminator to a parametric class is a common practice. In our theoretical results of Theorems 2.1 and 2.2, to simplify analysis, the discriminator is limited to linear functions, meaning that GANs match the mean of observed and generated samples. Although this is a simple GAN, in Section 2.4, we explain how this result can be used to analyze a GAN with a deep random feature discriminator model. Thus, while these results do not cover all forms of discriminators it can in fact handle a broad class of functions due to the universal approximation property of random feature models. Moreover, we provide empirical study of overparameterization for practical GANs with deep network discriminators in Figures 3 and 4.
>
> [6] Arora et al., “Generalization and Equilibrium in Generative Adversarial Nets”
> [7] Feizi et al., “Understanding GANs in the LQG setting”
>
> $\textbf{“Why V is not optimized in Eqn. (2) & (3)?”}$:
>
> The proof technique can in fact be used to analyze optimization over V as well. However, it requires more messy/tedious calculations without providing any new insights. Therefore, we have opted to do an analysis with fixed V. We also note that this is a standard practice in the theory of overparameterized neural network training literature in the supervised setting e.g. see [8] and [9]
>
> [8] Du et al., “Gradient descent provably optimizes over-parameterized neural networks”.
> [9] Oymak et al., “Towards moderate overparameterization: global convergence guarantees for training shallow neural networks”
>
> $\textbf{“Theorem 2.3 claims general minimax problem. However, it is restricted to a linear discriminator”}$:
>
> Theorem 2.3 as stated only states a general nonlinear function not a general GAN. Perhaps the reviewer is referring to the question in the title of the section. To avoid any confusion we have added changes to “more general GANs”. We would also like to note that as mentioned in the discussion right after the theorem this result can be used beyond linear discriminators such as random feature models which are known to be universal approximators. So while this new result does not cover all forms of discriminators it can in fact handle a broad class of functions due to the universal approximation property of random feature models.

---

> > ### Author Response · Authors · 2020-11-20
> > **Response to reviewer 3 (2)**
> >
> > $\textbf{“In Fig. 4, with smaller k, G’s capacity is not sufficient to capture the data distribution.”}$:
> >
> > We do not agree with this statement. Please refer to Figure 8 in the Appendix where we discuss the details on architectures corresponding to the value of k. We argue that even with k=8, the ConvNets are sufficiently complex and can model the data distribution. In fact, it has been shown in [10] that pruned networks with similar model sizes as what has been considered in this work can model the data distribution. The issue with using smaller model size is the poor optimization landscape which leads to poor convergence of GDA. Using overparameterized models significantly help convergence in GANs, which is the main claim of this paper.
> > [10] Kalibhat et al., “Winning Lottery Tickets in Deep Generative Models”
> >
> > $\textbf{“The paper does not provide any bound on the gap between training error and test error.”}$
> >
> > We show the gap between training error and test error in Figure 4 and Figure 7 empirically.
> >
> > $\textbf{“I don’t know the purpose of Fig. 3 and Fig. 5.”}$:
> >
> >  In Figure 3 and Figure 5 of the original draft (Fig 4 and 7 of the revised draft), we show the FID scores of trained GAN models improve as the number of parameters in the GAN (k) increases. This supports the theoretical results showing overparameterized GANs (for simpler models) demonstrate better GDA convergence. We also show the gap between test and training FID scores in Figure 3, as you asked in your previous comment.
> >
> > $\textbf{“Three related work”}$:
> >
> > Thanks for these references. We added a discussion about them to introduction and Section 2.3 of the revised draft.

---

> > > ### Comment · AnonReviewer3 · 2020-11-25
> > > **Need further response**
> > >
> > > First, thanks for your clarity in terms of the definition of over-parameterizations. After carefully reading your response, I still have the concern that cannot raise my score to 6 or higher: The significance of your method.
> > >
> > > Specifically, the main claims in this paper simply focus on the mean of the data distribution. However, for most data distribution, even as simple as a gaussian distribution, simply match the mean cannot obtain any meaningful results. Therefore, the convergence rate between the data means and the model means is far from understanding the converge performance of GANs.
> > >
> > > Below are my minor concerns:
> > > The theoretical analysis does not follow the basic assumptions of generative models, i.e., both the training data and the test data come from a certain underlying distribution. In this case, a generative model is used to capture the underlying data distribution, rather than simply capturing the data mean. Please choose the proper evaluation metric, such as KDE or KL divergence, instead of the MSE between the data mean and the model mean.
> > > Could you provide the generated samples of the numerical validation part, i.e., MNIST and gaussian, as well as the two-moon dataset, to show that your generator leans meaningful results, instead of simply captures the data mean?
> > > Figures. Since you want to show the convergence results, the proper should be the learning curves, rather than the converged results w.r.t. model width. If the learning curves show that model with different width finally converges to the same results, but the models with more parameters provide faster converge results, then the results can validate your claims. Otherwise, it simply provides a counterpart of your claims: smaller models actually do not have the ability to model the data distribution.
> > > Bounds between the train and test. First, simply provides empirical results are not enough for a paper named "understanding the over-parameterization". Second, it seems that the gap between the train and test is growing w.r.t. width of the model, indicating that the model is indeed overfitting as the model becomes larger. More explanations are needed.
> > >
> > > Overall, I don't think this paper provides any significant contributions to the GAN community.

---

### Official Review · AnonReviewer4 · 2020-10-27
**Many questions**

**Rating:** 6
**Confidence:** 2

**Review:**

This paper studies how over-parameterization plays a role in GAN training. Theoretically, it shows that a GAN with over-parameterized 1-layer neural network generator and a linear discriminator can converge to global saddle point via stochastic optimisation. Similar results are obtained for nonlinear generators and discriminators under some conditions. It also provides empirical results to support its findings.

## Pros

- The paper is easy to read.
- The result can be significant as it suggests we simply need larger GANs in practice.

## Cons / Questions

- I'm not sure if the term over-parameterization is well-defined in the paper. For training neural classifiers, the word "over" is clear because of the transition point of the double descending curve. But here we only see a descending curve (for performance) within the regime of parameter space the paper studies. To be more specific, is $k=256$, which is only 2x or 4x of common choices of feature numbers, considered as over-parameterized? Similar for Figure 2, is $k=1024$ for linear layers over-parameterized?
- The paper says, "One of the key factors that has contributed to the successful training of GANs is model over-parameterization". But in fact, a lot of works have been showing that a careful balance of generator and discriminator capacity is in fact important, which is also agreed by the authors in the end of Section 3. So, I'm not sure if this opening sentence is a valid claim.
- The empirical results are based on DCGAN and Resnet that use convolutions. Does similar result hold for MLPs? Testing it on MNIST may be useful.
- Can you explain how step sizes and other parameters are set while varying the capacity of GANs?
- The study on generalization gap seems to be incomplete. In particular, we also need to check the GANs simply memorizes the data while increasing the capacity, right? It's unclear if a dropped FID in both train and test (i.e. Figure 5) can rule out this issue.

---

> ### Author Response · Authors · 2020-11-20
> **Response to reviewer 4**
>
> $\textbf{“if the term over-parameterization is well-defined in the paper”}$:
>
> We define overparameterization based on the model parameter count. That is a model with more parameters is more over-parameterized compared to the one with fewer number of parameters. We added this definition in the beginning of Section 2 in the revised draft. In our theoretical results, we show that for certain GANs, when the number of parameters in the model is sufficiently large, GDA converges to the global solution of the min-max optimization problem. Notably, the number of model parameters depends on the input and latent sample dimensionalities (see Theorems 2.1 and 2.2 for the exact bounds). In our empirical results (e.g. Figures 3 and 4), we use standard datasets such as CIFAR-10 and Celeb-A and evaluate GANs by progressively increasing their model parameter counts. For example, a DCGAN with k=256 is more overparameterized than that with k=128. In these cases, we observe a similar trend to what our theory explains (for simpler GANs).
>
> Note that in supervised learning, observing the double descent phenomenon is just a potential consequence of the model overparameterization. Even in supervised learning, the double descent phenomenon does not always occur in the overparameterized regime, thus cannot be used to define the overparameterization even in supervised learning.
>
> $\textbf{“a lot of works have been showing that a careful balance of generator and discriminator ..."}$:
>
> To be clear, in the paper we show that overparameterization is a factor contributing to the success of GANs. Of course there are many other considerations in a successful GAN design. For example, as we explain in Section 3, balancing the number of parameters of generator and discriminator is critical for the success of GAN’s training in practice.  This is why in our empirical studies of Figures 3 and 4, we increase the model size of both generator and discriminator (while still maintaining the balance). This claim is also supported in BigGAN[1], where authors observe that using large models (while maintaining the balance in generator and discriminator model sizes) is critical to improve the quality of synthesized samples which we systematically study in the paper. We have included this discussion in the revised draft (see intro).
> [1] Brock et al., “Large scale GAN training for high-fidelity image synthesis”, ICLR 2019.
>
> $\textbf{“Does similar result hold for MLPs?”}$:
>
> We thank the reviewer for suggesting this experiment. Upon your suggestion, we conducted this experiment. In Figure 3 of the revised draft, we included an experiment where we train a 3-layer MLP GAN model on MNIST dataset by varying the hidden dimension in MLP. We plot the training and test FIDs of the trained models. We observe a similar behavior as ConvNets, where the FID scores improve as large k is used.
>
> $\textbf{“Can you explain how step sizes and other parameters are set?”}$:
>
> We use the standard hyper-parameter configurations (which are mentioned in Section C of the Appendix). We observed that these standard hyper-parameters converge well in practice.

---

> > ### Author Response · Authors · 2020-11-20
> > **Response to reviewer 4 (2)**
> >
> > $\textbf{“The study on generalization gap seems to be incomplete ...”}$:
> >
> > We thank the reviewer for this suggestion. We agree with the reviewer that the FID score has an issue that it may assign low error scores to the memorized samples, hence it doesn’t completely capture the notion of generalization. In fact, this issue with the FID score has been observed in some of the recent papers (e.g. [2,3]). In [2] and [3], authors claim that one of the reasons why FID fails at memorization is because the FID scores have low sample complexity, which means that with few empirical samples, one can closely approximate the FID score of the full distribution. For divergences that satisfy the low sample complexity property, the divergence value between the true distribution and the empirical distribution will hence be small. So, they assign low error scores for memorized samples.
> >
> > To fix this issue, [2] and [3] use Neural Net Divergence (NND) as an evaluation measure for GANs. NND scores have high sample complexity. The authors in [2] show that by carefully training a model to measure NND, NND scores of memorized samples can be higher than that of GANs. To the best of our knowledge, this is the best existing evaluation metric to empirically measure generalization in GANs.
> >
> > In Figure 6 of the revised draft, we report overparameterization results using NND scores for DCGAN and Resnet models trained on CIFAR-10 dataset. We observe that for large values of $k$, NND scores of GAN are higher than the memorized samples (which are the training set samples). Similar to the FID curve, increasing $k$ does not increase the NND scores, it merely flattens the curve. This means that overparameterized models have not been memorizing training samples.
> >
> > [2] Gulrajani et al., “Towards  GAN  benchmarks  which  require generalization.”
> > [3] Arora et al., “Generalization and equilibrium in generative adversarial nets (gans)”

---

> > > ### Comment · AnonReviewer4 · 2020-11-23
> > > **Thanks for the response**
> > >
> > > Re. over-parameterization
> > >
> > > I'm still confused by the statement "a model with more parameters is more over-parameterized compared to the one with fewer number of parameters" because there is no role for the term "over". Is the statement equal to "a model with more parameters is more parameterized compared to the one with fewer number of parameters" (i.e. without the term "over")? I checked a few papers referenced in the draft as you pointed to R2. For example, [1] uses the phrase "the number of parameters is *sufficiently* large", where the word "sufficiently" justified the prefix "over". The reason I think this is important is because this really determines which regime of parameters the study focuses on. If "over-parameterization" has to be defined on a "sufficiently" large number of parameters space, how large is sufficient? Is empirical study on small regime is enough? I still agree with R2 that the work "seems to me to rather be about model complexity than about over-parametrization of the model."
> > >
> > > Re. MLP results
> > >
> > > Can you clarify what do you mean by "We observe a similar behavior as ConvNets, where the FID scores improve as large k is used."? Clearly the FID number goes up when k varies from 1024 to 2048, which is not the case of "similar behavior". This is again related to what is the sufficiently large number of parameters you want to study. Would it be the case the the current regime of the ConvNets experiment is also too narrowed so that we have not observed the increase of the FID? I would be convinced to see figures with a larger number of parameters for both the MLP and ConvNets study.
> > >
> > > Re. generalization gap
> > >
> > > Thanks for the new results. Can you explain what does the current NND number (around 13.8) mean? What's the typical value of a GAN which memorizes the set and what's the typical value of a GAN with reasonable generalization? It's very weird as (1) small models which I know they don't memorize has higher values (2) the value with training set samples is around 14.6 in the middle (3) value for models that saturate is lower than 14.6. It's clear that higher NND values don't indicate memorization, so how does this plot help? I think giving such explanations plus a nearest-neighbour check (i.e. generating some samples and show them together with 10 nearest-neighbours of each in the training set) as qualitative study is necessary to make readers understand what's going on.
> > >
> > > [1] Samet Oymak and Mahdi Soltanolkotabi. Towards moderate overparameterization: global convergence guarantees for training shallow neural networks. IEEE Journal on Selected Areas in Information Theory, 2020.

---

> > > > ### Author Response · Authors · 2020-11-24
> > > > **Response to reviewer 4 (3)**
> > > >
> > > > **“confused by the statement "a model with more parameters is more over-parameterized”**
> > > >
> > > > Yes, we also mean "the number of parameters is sufficiently large" (now more explicitly in the paper). To be clear, in Theorem 2.1, we require $	k\geq C\cdot md^9\log\left(d\right)^3$ where $k$ is the number of hidden neurons in the generator, $m$ and $d$ are dimensions of the input and latent spaces. This precisely defines, at least in the setting of this Theorem, what we mean by the number of parameters to be sufficiently large. In the setup of Theorem 2.2., we need  $k\geq C\cdot md^4\log\left(d\right)^3$. In more general GANs, we don’t have such specific thresholds. Thus, we progressively increase the number of model parameters and observe the effects in the loss (measured by FID) and the convergence behavior of GDA.
> > > >
> > > >
> > > > **“Is empirical study on small regime is enough?”.**
> > > >
> > > > Please note that the models we have tested empirically are not in a small regime. $k$ in DCGAN and Resnet-based models denote a multiplicative factor to the number of filters (mentioned in Section 3 of the revised draft). Please refer to Figure 8 in the Appendix for obtaining the exact model descriptions corresponding to any value of $k$. We would like to point out that even with $k=8$ (the smallest model in our experiments), the number of parameters used in DCGAN, Resnet GAN and MLP GAN are 109,000, 12,155 and 10,392. Even the smallest models we have considered have sufficient capacity to represent complex data distributions (as shown in [1])
> > > > [1] Kalibhat et al., “Winning Lottery Tickets in Deep Generative Models”
> > > >
> > > >
> > > > **MLP results: “Can you clarify what do you mean by "We observe a similar behavior as ConvNets”**
> > > >
> > > > As we show in Figure 3, using larger MLP GANs helps decrease the FID (till $k=1024$). Also note that the FID difference between $k=1024$ and $2048$ is insignificant (compared to the FID decreases in the previous steps). This is potentially due to an increased generalization gap in this regime where it offsets potential benefits of over-parameterization.  We have now clarified this in the revised draft.
> > > >
> > > > As mentioned before, we would like to re-emphasize that the number of parameters in convnet models are fairly large. For instance, the number of parameters in DCGAN and Resnet GANs corresponding to different values of k are given below:
> > > >
> > > > | Hidden dimension (k) | DCGAN | Resnet GAN |
> > > > | ----------- | ----------- | ----------- |
> > > > | 8  | 109,000  | 12,155 |
> > > > | 16 | 218,000 | 24,310 |
> > > > | 32 | 436,000 | 48,620 |
> > > > | 64 | 872,000 | 97,240 |
> > > > | 128 | 174,4000 | 194,480 |
> > > > | 256 | 348,8000 | 388,960 |
> > > >
> > > > We were unable to train models beyond this width due to hardware limitations as models could not fit in the memory.
> > > >
> > > > **Generalization gap and NND**
> > > >
> > > > NND computes the neural net divergence between the real data distribution and the generated data distribution. Similar to FID, for NND scores, **lower values are better**. A higher value of NND scores could mean two things: (1) generated samples are poor, or (2) samples do not have diversity (i.e. they are memorized). Models with smaller $k$ falls into the first category, hence, their NND scores are higher. Similarly, memorized samples do not have diversity, hence their NND scores are high as well. On the other hand, GAN models which are over-parameterized have better sample quality and at the same time have high diversity (i.e. they haven’t memorized training samples). Thus, they have lower NND scores. Please refer to Gulrajani et al., “Towards GAN benchmarks which require generalization.” for more details.
> > > >
> > > > As per Gulrajani et al., “Towards GAN benchmarks which require generalization.”, the following are the NND scores of some standard generative models in comparison with memorized samples:
> > > >
> > > > | Method| NND |
> > > > | ----------- | ----------- |
> > > > | Pixel CNN++  | 16.17  |
> > > > | IAF VAE | 18.11 |
> > > > | WGAN-GP | 12.97 |
> > > > | Training set (memorized) | 14.62 |
> > > >
> > > > So, their best performing method (WGAN-GP) got a NND score of 12.97, which is close to the best NND score we get. This indicates that our over-parameterized GANs have not memorized training samples.
> > > >
> > > >
> > > > Regarding your proposed *nearest neighbor check*, we have generated samples from a DCGAN model trained on CIFAR-10 dataset with k=8 and k=128, and queried the nearest neighbor samples in the training set. We included these results in Appendix section D of the revised draft as your proposed qualitative. We observe that an over-parameterized model generates samples with better quality and high diversity. Also, assessing nearest samples qualitatively re-emphasizes that over-parameterized models do not suffer from memorization of training samples.
> > > > We would appreciate it if you take these responses into account in re-evaluating our paper. Thanks.

---

### Official Review · AnonReviewer1 · 2020-10-28
**A good first step toward a theoretical understanding of over-parametrization in GANs.**

**Rating:** 7
**Confidence:** 3

**Review:**

# Summary
This paper studies the effect of model over-parametrization in GANs. While there is a lot of work on this in the supervised learning setting of classification/regression there is not much in the GAN framework where the minimax objective function complicates such an analysis. This paper considers two types of training of the GAN model, one with the simultaneous gradient descent ascent and one where the discriminator is trained to optimality for every generator update. It provides global convergence results under both algorithms in the case of a generator network with one hidden layer that is large enough and a linear discriminator.

# Contributions:
Given the importance of over-parametrization of neural networks for better performance and low generalization error this work is a step toward a theoretical understanding of this phenomena in the GAN setting. Even though the global convergence results are provided in a simple setting, the proof techniques using dynamical systems and control theory might be useful for extending these results to a more general framework.

The comparison between convergence rates of the two training algorithms is also interesting.

The experimental evidence seems compelling, although it would be nice to consider examples other than simple gaussians for a more comprehensive empirical analysis.

# Comments:
1. It would be nice to compare the two types algorithms considered in terms of empirical evidence. For example a combination of the two plots in Figure 2 would be nice in order to get a sense of the effects of the two theorems 2.1 and 2.2 empirically.
2. Just out of curiosity can we say anything about the training of the discriminator for a finite number of steps and how that affects convergence? Sort of as an interpolation between the two algorithms.
3. A little more variety in the toy empirical examples considered would be nice but not crucial. While the experiments on the real data are compelling it would be nice to have empirical evaluations where we can compute the true generalization error or know the ground truth which is not simply Gaussian.
4. It would also be nice if the authors could provide a summary of the proof of the main theorems either in the main paper or at least at the beginning of the appendix jus to provide a rough sketch of what pieces are required to show such a result.

---

> ### Author Response · Authors · 2020-11-20
> **Response to reviewer 1**
>
> $\textbf{Comments 1 and 2}$:
>
> We thank the reviewer for these comments. In the revised draft, we have included a new experiment, where we perform numerical simulations by training discriminator for $d_{iter}$ steps per generator update, with $d_{iter}$ is in the range {2, 5, 10} (Fig 2). This training algorithm stands as an interpolation between the regimes considered in Theorems 2.1 and 2.2. We observe similar convergence behavior in all cases. See updated Figure 2 in the paper.
>
> $\textbf{Comment 3}$:
>
> In addition to using a Gaussian dataset, we have added new numerical simulations using the two-moons dataset. The results are included in the appendix (Figure 9) of the revised draft. We observe similar convergence behavior as the one shown for the Gaussian experiments.
>
> $\textbf{Comment 4}$:
>
> Thank you for the suggestion. We included a proof sketch in Section A.2 of the Appendix, where we give an outline of the proof of Theorem 2.1. As we will explain briefly, this result is derived from the more general Theorem A.4, which is applicable to a larger class of min-max problems. Moreover, in Section A.2 we give an outline of the following four key steps, which are necessary to prove Theorem A.4,
>
> *Step 1*: Rewrite the GDA updates as a linear time-varying system
> *Step 2*: Approximate this system by a linear time-invariant system
> *Step 3*: Analyse this time-invariant linear dynamical system
> *Step 4*: Complete the proof via a perturbation argument

---

### Official Review · AnonReviewer2 · 2020-10-28
**Interesting problem and results, but unclear narrative**

**Rating:** 6
**Confidence:** 3

**Review:**

Summary:

While much work has been devoted to understanding the role of the discriminator in GAN training, comparably little is known about the role of the generator.
In this work, the author address this important question, more exactly the role that generator overparametrization plays in GAN training.
Empirically, the authors show that when applying DCGAN and RESNET architectures to CIFAR10 and CELEBA, increasing the number of parameters in the hidden layers decreases the Frechet inception distance and improves image quality.
Theoretically, they show that RELU networks with a single hidden layer converge to the global minimum under simultaneous gradient descent with high probabiliy, provided that the dimension of the hidden layer is large enough.

Recommendation:
I think that this work makes a contribution to an important topic that has not been studied very much, so far. My main concern is that I find the presentation somewhat lacking at this point. In particular, as detailed below, I would appreciate a discussion of overparametrization vs model complexity. I would also suggest to the authors to consider beginning by presenting their experiments on real GANs to describe the empirical phenomenon that their theorem is supposed to illuminate.
Finally, I believe that some additional guidance in interpreting the results of the theorem would be helpful. In particular, the comparison of the setting of Theorems 2.1 and 2.1 should be more prominent since the difference between these two result is what really captures the relationship to GANs, as opposed to arbitrary RELU models.

Questions/Suggestions
- How do the authors define "overparametrization"? In particular, how is it different from model complexity. It is not clear to me why Figure 1 should concern "overparametrization" as opposed to model complexity? I understand that in the deep learning setting these concepts might not be easy to differentiate, in either way I believe some more background on this topic would be warranted. Similarly, a key element the lower bound on $k$ in the theorem seems to be that ensures that minimization objective has optimal value of zero, that is the "data" can be reproduced exactly.
This again seems to me to rather be about model complexity than about overparametrization of the model. Looking at the proof it seems that the concentration argument leading to a well-conditioned Jacobian does not depend on model being perfectly able to represent the "training data". This might be worthwhile to explain in more detail in the paper.

- "The networks are optimized using Gradient Descent/Ascent (GDA) to reach a saddle-point of the min-max optimization problem."
There have been multiple recent works such as [Berard et al.](https://arxiv.org/abs/1906.04848) [Farnia and Ozdaglar et al, 2020](https://arxiv.org/abs/2002.09124), [Schafer et al.](https://arxiv.org/abs/1910.05852) that call the role of nash-equlibria/saddle points into question. I think the authors should engage, at least briefly, with this line of work.

- " One of the key factors that has contributed to the successful training of GANs is model overparameterization. By increasing the complexity of discriminator and generator networks, both in
depth and width, recent papers show that GANs can achieve photo-realistic image and video synthesis (Brock et al., 2019; Clark et al., 2019; Karras et al., 2019)."
This seems a bit misleading since at least the Karras et al. work suggests much more specific modifications than just increasing the complexity. I don't think any of the references listed support the claims about the importance of overgenearlization as much as the way in which they are cited suggests.

- "In particular, it has been empirically observed (as we also demonstratein this paper) that when the generator/discriminator contain a large number of parameters (i.e. are sufficiently over-parameterized) GDA does indeed find (near) globally optimal solutions. In this section we wish to demystify this phenomenon from a theoretical perspective."
This statement should be provided with some evidence. it is also not clear what a (near) globally optimal solution means. In the context of GANs, [Arjovsky and Bottou 2017](https://arxiv.org/abs/1701.04862), [Berard et al.](https://arxiv.org/abs/1906.04848), [Schafer et al.](https://arxiv.org/abs/1910.05852) show that GANs do not even converge to (nearly) locally optimal points, despite producing good images, contradicting the statement.

minor points:
- "very simpler" -> much simpler

================================================================================================

After reading the author's response and the other reviews I still lean slightly towards acceptance and have therefore left my rating unchanged.
While not being an expert on the subject, I find the work interesting. In case the paper gets rejected, I recommend to the authors to the feedback provided by the referees to clarify the narrative of the paper.

---

> ### Author Response · Authors · 2020-11-20
> **Response to reviewer 2**
>
> Thank you for your insightful review.
>
> $\textbf{How do the authors define "overparametrization"?}$:
>
> We define overparameterization based on the model parameter count. That is, a model with more parameters is more over-parameterized compared to the one with less number of parameters. We added this definition in the beginning of Section 2 in the revised draft. In our theoretical results, we show that for certain GANs, when the number of parameters in the model is sufficiently large, GDA converges to the global solution of the min-max optimization problem. Notably, the number of model parameters depends on the input and latent sample dimensionalities (see Theorems 2.1 and 2.2 for the exact bounds). As you point out, for the GANs considered in these Theorems, the loss approaches zero.  In our empirical results (e.g. Figures 3 and 4), we use standard datasets such as CIFAR-10 and Celeb-A and evaluate GANs by progressively increasing their model parameter counts. In these cases, the error (measures by FID) does not approach zero but we observe a similar trend to what our theory explains (for simpler GANs).
>
> We agree with you that defining overparameterization based on parameter counts for deep models is somehow related but distinct from the complexity of the hypothesis class. We have clarified this definition in our revised paper. We also note that the parameter count-based definition has extensively been used in studying overparameterization in supervised learning (e.g. references Soltanolkotabi et al., 2018; Allen-Zhu et al., 2019; Du et al., 2019; Oymak & Soltanolkotabi, 2019; Zou & Gu, 2019; Oymak et al., 2019 in the paper).
>
> $\textbf{“There have been multiple recent works that call the role of nash-equlibria/saddle points into question.”}$:
>
> Thank you for pointing us to these references. We agree that for a “general” GAN, (local) saddle points may not even exist. This is also true for a general non-convex non-concave min-max (see ref [Daskalakis et al. 2020] in the paper). Our theoretical results (Theorems 2.1 and 2.2) focus on a specific family of GANs with a two-layer neural net as its generator and a linear discriminator. In that case, we not only prove the global saddle point exists, but the GDA converges to it exponentially fast.
> For a general min-max problem, [Daskalakis et al. 2020] has recently shown that “approximate” local saddle points exist under some general conditions on the Lipschitzness of the objective and are equivalent to the GDA’s fixed points. Our empirical results for a general GAN indicate that over-parameterization helps GDA find better solutions compared to small models. Of course, understanding GDA dynamics for a general GAN remains an important open question for the community but we believe this work is a good step towards that goal. We have added this discussion as well as citations you mentioned  to the paper (see Section 2.3).
>
> $\textbf{“Karras et al. work suggests much more specific modifications than just increasing the complexity. ”}$:
>
> To be clear, in the paper we empirically show that overparameterization is a factor contributing to the success of GANs. Of course there are many other considerations in a successful GAN design (e.g. the architectures used as generator and discriminator networks, the regularizations, etc).  For instance, the following quote is taken from the BigGAN paper [1]: “We demonstrate that GANs benefit dramatically from scaling, and train models with two to four times as many parameters and eight times the batch size compared to prior art”. Hence, one of the key factors in improving the performance of GANs is the model parameter growth which we systematically study in the paper.  We made this point more clear in the revised draft (Intro).
>
> [1] Brock et al., “Large scale GAN training for high-fidelity image synthesis”, ICLR 2019.
>
> $\textbf{“It is also not clear what a (near) globally optimal solution means. In the context of GANs, ...”}$:
>
> Thanks for pointing this out. In our theoretical results (Theorems 2.1 and 2.2), we have defined the convergence error of GDA to the global saddle point of the min-max optimization precisely. For a general GAN, as you point out, local saddle points may not even exist and GDA may converge to approximate local saddle points. For a general GAN, we empirically show that overparameterization helps GDA to find better solutions. It remains an open problem whether these solutions have better approximation gaps compared to the ones of smaller networks. We have added a discussion about this to Section 2.3 of the paper.

---

### Decision · Program_Chairs · 2021-01-07
**Final Decision**

**Decision:**

Accept (Poster)

**Comment:**

### Paper summary
This paper investigates theoretically and empirically the effect of increasing the number of parameters ("overparameterization") in GAN training. By analogy to what happens in supervised learning with neural networks, overparameterization does help to stabilize the training dynamics (and improve performance empirically). This paper provides an explicit threshold for the width of a 1-layer ReLU network generator so that gradient-ascent training with a linear discriminator yields a linear rate of convergence to the global saddle point (which corresponds to the empirical mean of the generator matching the mean of the data). The authors also provides a more general theorem that generalizes this result to deeper networks.

### Evaluation
The reviewers had several questions and concerns which were well addressed in the rebuttal and following discussion, in particular in terms of clarifying the meaning of "overparameterization". After discussing the paper, R1, R2 and R4 recommend acceptance while R3 recommends rejection. The main concern of R3 is that the GAN formulation analyzed in the paper is mainly doing moment matching between the generator  distribution (produced from a *fixed* set of latent variables z_i) and the empirical mean of the data. R3 argues that this is not sufficient to "understanding the training of GANs". At least two aspects are missing: how the distribution induced by the generator converges according to other notion of divergence (like KL, Wasserstein, etc.); and what about the true generator distribution (not just its empirical version from a fixed finite set of samples z_i)? While agreeing these are problematic, the other reviewers judged that the manuscript was useful first step in understanding the role of overparameterization in GANs and thus still recommend acceptance. And importantly, this paper is the first to study this question theoretically.

I also read the paper in more details. I have a feeling that some aspects of this work were already developed in the supervised learning literature; but the gradient descent-ascent dynamic aspect appears novel to me and the important question of the role of overparameterization here is both timely, novel and quite interesting. I side with R1, R2 and R4: this paper is an interesting first step, and thus I recommend acceptance. See below for additional comments to be taken in consideration for the camera ready version.

### Some detailed comments
- Beginning of section 2.3: please be clearer early on that you will keep V fixed to a random initialization rather than learning it. The fact that this is standard in some other papers is not a reason to not be clear about it.
- Theorem 2.2: in the closed form of the objective when $d$ is explicitly optimized, we are back to a more standard supervised learning formulation, for example (5) could look like regression. The authors should be more clear about this, and also mention in the main text that the core technical part used to prove Theorem 2.2 is from Oymak & Soltanolkotabi 2020 (which considers supervised learning). This should also a bit more clear in the introduction -- it seems to me that the main novelty of the work is to look at the gradient-descent dynamic, which is a bit different than the supervised learning setup, even though some parts are quite related (like the full maximization with respect to $d$).
- p.6 equation (8): typo -- the  $-\mu d_t$ term is redundant and should be removed as already included from $\nabla_d h(d,\theta)$.
- p.7 "numerical validations" paragraph: Please describe more clearly what is the meaning of "final MSE". Is this a global saddle point (and thus shows the limit of the generator to match the empirical mean), or is this coming from slowness of convergence of the method (e.g. after a fixed number of iterations, or according to some stopping criterion?). Please clarify.